# Luminal epithelial cells integrate variable responses to aging into stereotypical changes that underlie breast cancer susceptibility

Rosalyn W Sayaman[1,2,3,4]*[†‡], Masaru Miyano[1,2†], Eric G Carlson[1,5], Parijat Senapati[6§], Arrianna Zirbes[1,5], Sundus F Shalabi[1,5#], Michael E Todhunter[1,2], Victoria E Seewaldt[1], Susan L Neuhausen[1], Martha R Stampfer[4], Dustin E Schones[6], Mark A LaBarge[1,2,7]*

[1]City of Hope, Department of Population Sciences, Beckman Research Institute, Duarte, United States; [2]City of Hope, Center for Cancer and Aging, Beckman Research Institute, Duarte, United States; [3]City of Hope, Cancer Metabolism Training Program, Beckman Research Institute, Duarte, United States; [4]Lawrence Berkeley National Lab, Biological Sciences and Engineering, Berkeley, United States; [5]City of Hope, Irell and Manella Graduate School of Biological Sciences, Duarte, United States; [6]City of Hope, Department of Diabetes Complications and Metabolism, Beckman Research Institute, Duarte, United States; [7]Center for Cancer Biomarkers Research, University of Bergen, Bergen, Norway

*For correspondence:
rwsayaman@gmail.com (RWS);
mlabarge@coh.org (MALaB)

[†]These authors contributed
equally to this work

Present address: [‡]University
of California, San Francisco,
Department of Laboratory
Medicine, Helen Diller Family
Comprehensive Cancer Center,
San Francisco, United States;
[§]Rajiv Gandhi Centre for
Biotechnology, Thycaud, India;
[#]Arab American University,
Department of Biology and
Biochemistry, Faculty of
Medicine, Ramallah, Palestine

Competing interest: The authors
declare that no competing
interests exist.

Reviewing Editor: Richard M
White, University of Oxford,
United Kingdom

**Abstract** Effects from aging in single cells are heterogenous, whereas at the organ- and tissue-levels aging phenotypes tend to appear as stereotypical changes. The mammary epithelium is a bilayer of two major phenotypically and functionally distinct cell lineages: luminal epithelial and myoepithelial cells. Mammary luminal epithelia exhibit substantial stereotypical changes with age that merit attention because these cells are the putative cells-of-origin for breast cancers. We hypothesize that effects from aging that impinge upon maintenance of lineage fidelity increase susceptibility to cancer initiation. We generated and analyzed transcriptomes from primary luminal epithelial and myoepithelial cells from younger <30 (y)ears old and older >55 y women. In addition to age-dependent directional changes in gene expression, we observed increased transcriptional variance with age that contributed to genome-wide loss of lineage fidelity. Age-dependent variant responses were common to both lineages, whereas directional changes were almost exclusively detected in luminal epithelia and involved altered regulation of chromatin and genome organizers such as *SATB1*. Epithelial expression variance of gap junction protein *GJB6* increased with age, and modulation of *GJB6* expression in heterochronous co-cultures revealed that it provided a communication conduit from myoepithelial cells that drove directional change in luminal cells. Age-dependent luminal transcriptomes comprised a prominent signal that could be detected in bulk tissue during aging and transition into cancers. A machine learning classifier based on luminal-specific aging distinguished normal from cancer tissue and was highly predictive of breast cancer subtype. We speculate that luminal epithelia are the ultimate site of integration of the variant responses to aging in their surrounding tissue, and that their emergent phenotype both endows cells with the ability to become cancer-cells-of-origin and represents a biosensor that presages cancer susceptibility.

## Editor's evaluation

In this important study, the authors investigate the relationship between aging and breast cancer development. They use a large panel of early passage ex vivo isolates from breast tissues of young and old donors to interrogate global transcriptome underpinning of age-related reduction in lineage fidelity. This yielded compelling evidence that luminal epithelial cells undergo specific changes in linage fidelity with age and that these changes are associated with increased risk of cancer. The rigor of the analysis is convincing and will provide new areas for functional exploration by other investigators.

## Introduction

Organisms, organs, and tissues exhibit stereotyped aging phenotypes that represent the integration of accumulated, stochastically incurred damages to individual cells that result in commonly understood hallmarks of aging (*López-Otín et al., 2013*; *Todhunter et al., 2018*). Stereotypical changes appear directional to the observer and are apparent at different physiologic scales, for example phenotypically though wrinkling, graying hairs, and increasing frailty; cellularly through increasing organ dysfunction, and loss of bone density, muscle mass and fat pads; and molecularly through decreasing levels of androgens and estrogens, and the upregulation or downregulation of gene or protein levels. Age-associated directional changes in transcriptomes of whole tissues are well documented (*de Magalhães et al., 2009*; *Glass et al., 2013*; *Peters et al., 2015*; *Volkova et al., 2005*). These directional molecular changes explain, at least in part, the noticeable phenotypic changes that accompany aging. However, although increased susceptibility to a plethora of diseases, including cancers, is a prominent consequence of aging, the emergence and onset of diseases vary among same-aged individuals. Indeed, variance in gene expression between individuals arises in the contexts of tumors, diet, and aging (*Bashkeel et al., 2019*; *Brinkmeyer-Langford et al., 2016*; *de Jong et al., 2019*; *Sharma et al., 2018*; *Viñuela et al., 2018*). We propose that this variability among individuals may itself be an important molecular phenotype of aging, and individuals with outlier expression profiles provide an avenue for understanding biological processes that explain the differences in cancer susceptibility between individuals as they age.

The breast is an excellent model system for examining aging at the cellular and molecular levels because normal tissue from individuals spanning the adult lifespan is available from common cosmetic and prophylactic surgeries. Cultured pre-stasis human mammary epithelial cells (HMEC) support growth of breast epithelial cell lineages from women across the lifespan (*Garbe et al., 2009*; *Labarge et al., 2013*) and enable detailed, reproducible molecular studies of cancer progression (*Stampfer et al., 2013*). Moreover, there are well-established lineage-specific markers and cell-sorting protocols that facilitate experimentation at lineage-specific resolution. Furthermore, breast tissue provides an ideal model for studying aging-associated cancer susceptibility as 82% of new breast cancers are diagnosed in women $\geq$ 50 y (*DeSantis et al., 2019*). Directional changes in gene expression with age have been reported in whole breast tissue, including changes associated with biological processes involved in breast cancer (*Lee and Lee, 2017*; *Yau et al., 2007*). However, aging is also associated with significant shifts in proportions of breast cell lineages, including epithelial and stromal populations (*Benz, 2008*; *Garbe et al., 2012*). Thus, it is unclear how tissue-level molecular changes during normal aging reflect changes in cell-intrinsic and microenvironment states. Lineage-specific analyses are needed to unravel such mechanisms.

The mammary epithelium, the origin of breast carcinomas, is a bilayer of two major phenotypically and functionally distinct cell lineages—myoepithelial cells (MEPs) and luminal epithelial cells (LEPs). MEPs are basally located and contractile and have tumor suppressive properties (*Pandey et al., 2010*). Luminal epithelial cells (LEPs) are apically located and include subpopulations of estrogen receptor negative (ER-) secretory cells, which represent 80–95% of luminal cells in breast tissue, and ER positive cells (ER+) (*Booth and Smith, 2006*). We previously demonstrated loss of lineage fidelity in breast epithelia as an aging phenomenon—in which the faithfulness of expression of established lineage-specific markers diminishes with age without loss of the lineage-specificity of other canonical markers nor of the gross phenotypic and histological differences between LEPs and MEPs (*Miyano et al., 2017*). Although our earlier work focused on a few select LEP-specific and MEP-specific markers,

**eLife digest** One of the main risk factors for breast cancers is aging. But how exactly aging contributes to breast cancer is not fully understood. The mammary gland is the part of the breast that can produce milk, and is composed of two major cell types: luminal epithelial cells and myoepithelial cells. Most age-related breast cancers are thought to originate from luminal epithelial cells, but some age-related changes in myoepithelial cells may also cause cancer. These changes in breast epithelial cells vary across individuals, which may explain why some individuals are at a higher risk for breast cancer than others.

Studying age-related changes in luminal and myoepithelial cells may help scientists pinpoint what causes age-related breast cancers. It may also help scientists to identify those at high risk for breast cancer, or develop alternative treatments. It could also help prevent breast cancers from occuring, getting worse or coming back. Studying age-related changes in gene expression in these two types of breast epithelial cells is the first step.

Sayaman et al. showed that aging leads to widespread changes in the genes expressed by luminal and myoepithelial cells. The researchers compared luminal and myoepithelial cells from breast tissue samples taken from postmenopausal women older than 55 years and premenopausal women younger than 30 years. Their experiments and analyses revealed that age-related gene expression changes reduced the cells' ability to maintain their identity and function over time. Many of the age-related expression changes occurred in cancer-linked genes.

Sayaman et al. found luminal and myoepithelial cells had increasingly varied gene expression among older women compared to younger women. Specifically, they saw changes in genes that helped these cells communicate, essentially changing the message relayed from myoepithelial cells to the neighboring luminal cells. The experiments further revealed large increases and decreases in gene expression with age in luminal cells but not in myoepithelial cells. Most of these age-related changes in luminal cells were linked to genes that play a role in cancer development.

These findings suggest that age-related changes make luminal cells more prone to becoming cancerous. Sayaman et al. also developed machine learning algorithms to identify age-related gene expression patterns that may distinguish tumor cells from normal breast tissue. More research in larger populations will help confirm the results. But if these efforts are successful, they may one day help clinicians identify women at risk of age-related breast cancers, detect cancers early, or create personalized prevention or treatment approaches.

we hypothesized that the aging mechanisms we observed could impinge upon genome-wide maintenance of lineage fidelity and could thus be potential drivers of susceptibility to cancer initiation in breast tissue.

Here, we demonstrate how age-dependent directional and variant transcriptional responses integrate in breast epithelia and explain how these changes could lead to increased susceptibility to cancer initiation. Directional responses reflect stereotyped changes associated with upregulation or downregulation of gene expression between younger and older cohorts; variant responses reflect increases in gene expression variance within a cohort associated with the heterogeneity of individuals within a group. Through transcriptomic profiling of primary LEPs and MEPs, we found that loss of lineage fidelity in gene expression with age was a genome-wide phenomenon. We identified two models mediating loss of lineage fidelity in breast epithelia with age: (i) via directional changes identified through differential expression (DE) analysis; and (ii) via an increase in variances identified through differential variability (DV) analysis. Age-dependent DE explained part of the observed loss of lineage fidelity, while our model of the overall increase in variances with age also accounted for a comparable fraction of this loss. Directional changes in expression with age strikingly occurred almost exclusively in luminal cells, whereas changes in variance were found in both lineages. Genome-wide directional changes in gene expression in LEPs involved dysregulation of chromatin and genome organizers such as *SATB1* with age. We also detected this dysregulation in bulk tissue that consist of all lineages of the stroma and epithelia. Loss of lineage fidelity led to enrichment of genes and biological processes commonly dysregulated in cancers, and altered the LEP-MEP interactome that was significantly modulated by apical cell-cell junction proteins, such as *GJB6*. Modulating *GJB6* expression via

shRNA in MEPs was sufficient to reduce the rate of molecular aging of adjacent LEPs as determined with a breast-specific biological clock. Using machine learning, we showed that genes that had age-dependent directional and variable changes in normal LEPs had predictive value in distinguishing normal breast tissue from breast cancers and in classifying breast cancer PAM50 subtypes. Age-dependent changes in LEPs reflected dysregulation of biological processes that are convergent with breast cancer. The degree and variability of age-dependent changes across individuals may explain the differential susceptibility between individuals to breast cancer initiation, and to the development of specific breast cancer subtypes.

## Results

### Genome-wide loss of lineage-specific expression in breast epithelia with age

Mammary glands are bilayer structures consisting of Keratin 19 (KRT19)-positive LEPs surrounded by KRT14-positive contractile MEPs (*Figure 1A*). The luminal lineage is composed of ER+ hormone sensing and ER- secretory populations with ER- LEPs accounting for the vast majority of the luminal lineage independent of age (*Pelissier Vatter et al., 2018*). To address the mechanism of age-associated loss of lineage fidelity, we used HMEC primary cultures from reduction mammoplasties (RM) that maintained the ER- LEP and MEP lineages from two age cohorts: younger <30 y women considered to be premenopausal ($m_{LEP}$ = 16, $m_{MEP}$ = 16 samples; n=11 subjects; age range 16-29y) and older >55 y women considered to be postmenopausal ($m_{LEP}$ = 11, $m_{MEP}$ = 11; n=8; age range 56-72y) (*Figure 1—source data 1A*). LEPs and MEPs were enriched by fluorescence-activated cell sorting (FACS) with anti-CD227 (*MUC1*) or anti-CD133 (*PROM1*) and anti-CD10 (*MME*) or anti-CD271 (*NGFR*), respectively (*Figure 1B*, *Figure 1—figure supplement 1A, B and C*). Unsupervised hierarchical clustering based on transcriptome-wide gene expression profiles revealed no detected bias between the two FACS-enrichment strategies (approximately unbiased, AU p≥0.95) (*Figure 1—figure supplement 1D*). Key lineage markers, including *EPCAM* and *ERBB2* for LEPs, and *TP63* and *EGFR* for MEPs, were differentially expressed in FACS-enriched LEPs and MEPs from HMEC primary cultures as expected (*Figure 1C*, *Figure 1—figure supplement 1E*; *Del Toro et al., 2024*; *Pandey et al., 2010*; *Thi et al., 2024*). We confirmed lineage-specific KRT protein expression by intracellular staining flow cytometry; KRT19 and KRT14 were exclusively expressed in CD133$^+$/CD271$^-$ LEPs and CD133$^-$/CD271$^+$ MEPs, respectively (*Figure 1—figure supplement 1F*). LEPs and MEPs from HMEC primary cultures were comparable to those from in vivo breast tissues based on gene and surface marker protein expression. Enriched LEPs and MEPs from dissociated uncultured breast epithelial organoids showed *PROM1* and *NGFR* gene expression in a manner anticipated by our sorting strategy (*Figure 1—figure supplement 1G and H*). RNA-seq analysis identified 17,328 genes with comparable ranges of expression levels and consistent lineage-specific expression between primary organoid and 4th passage LEPs and MEPs in both age cohorts (linear regression R$^2$=0.88–0.91, p<0.0001) (*Figure 1—figure supplement 2A–D*). Thus, early passage primary HMECs in culture retained lineage specificity when compared to uncultured primary breast epithelial organoids.

To understand how lineage fidelity of the two epithelial cell types changes with age, we performed DE analysis comparing LEP and MEP expression in younger and older women. DE genes between LEPs and MEPs decreased with age (adjusted p<0.05, <0.01, <0.001). Restricting analysis to genes that had strong lineage-specific bias (DE adj. p<0.001, absolute log$_2$ fold change (abs(lfc)) ≥1), we found 4,040 genes (23% of all genes analyzed) with highly significant lineage-specific DE in younger women (*Figure 1—source data 2A*) and (*Figure 1—source data 3A*). Of these genes, 59% were LEP-specific and 41% were MEP-specific. In contrast, 3345 genes had highly lineage-specific DE in older women (*Figure 1—source data 2B*) and (*Figure 1—source data 3B*), of which 56% were LEP-specific and 44% were MEP-specific. Global shifts in lineage-specific expression associated with age were visualized by strata-plot (*Figure 1D*). Loss of lineage-specific expression with age occurred genome-wide and was detected in 1022 genes, a majority of which (65%) were LEP-specific genes.

Loss of lineage fidelity is the loss of faithful expression of lineage-specific markers with age. Statistically, we described this loss as a phenomenon whereby the magnitude of gene expression differences that distinguish LEPs from MEPs decreased with age, which is seen as shifts in distributions of fold changes between lineages to smaller values in the older cohort (Kolmogorov-Smirnov two-sample

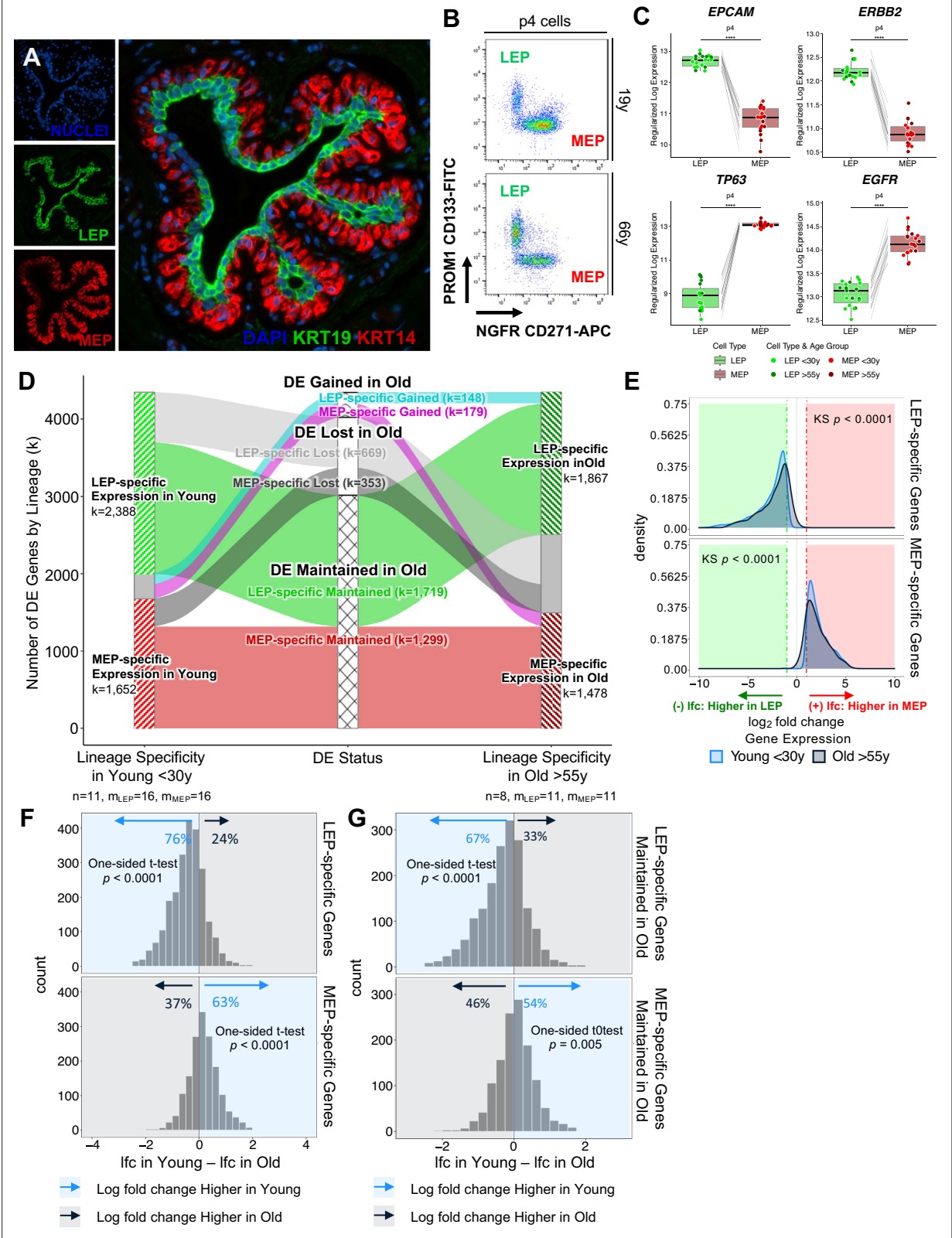

**Figure 1.** Genome-wide loss of lineage-specific expression with age. (**A**) Immunofluorescence staining of normal breast tissue showing the mammary epithelium with an apical LEPs (KRT19) surrounded by basal MEPs (KRT14). (**B**) Representative FACS enrichment plot of HMECs stained with LEP-specific CD133 (PROM1) and MEP-specific CD271 (NGFR). (**C**) Boxplots of subject-level gene expression rlog values of canonical LEP-specific markers *EPCAM* and *ERBB2* (top), and MEP-specific markers *TP63* and *EGFR* (bottom) in FACS isolated LEPs and MEPs in all age groups. Lineage-specific DE adj. p-

*Figure 1 continued on next page*

*Figure 1 continued*

values annotated (*<0.05, **<0.01, ***<0.001, ****<0.0001). (**D**) DE LEP-specific and MEP-specific genes (adj. p<0.001, abs(lfc) ≥1) in younger <30 y (left) and older >55 y (right) women. Strata plot shows changes in lineage-specific DE with age, showing the number of LEP- and MEP-specific genes gained (cyan and magenta), lost (light and dark gray), and maintained (green and red) in older women. Number of subjects (n) and sample replicates (m) in each DE analysis annotated; number of DE genes (k) in each age group and DE status indicated. (**E**) Distribution of lfc in expression between LEPs and MEPs in younger and older subjects for either DE LEP-specific (top panel) or MEP-specific (bottom panel) genes. KS p-values for equality of distributions of lfc between younger and older women annotated. (**F–G**) Histogram of pairwise differences in lfc in expression between LEPs and MEPs in younger vs. older women for (**F**) all genes with lineage-specific expression in younger women or (**G**) only genes that maintain lineage-specific expression in older women. Genes with LEP-specific and MEP-specific expression are shown in the top and bottom panels respectively. The percent of genes with higher lfc in younger women (light blue) or higher lfc in older women (blue gray) are indicated. One-sided t-test p-values annotated.

The online version of this article includes the following source data and figure supplement(s) for figure 1:

**Source data 1.** RNA-sequencing sample list.

**Source data 2.** Lineage-specific DE summary.

**Source data 3.** Genome-wide loss of lineage-specific expression with age.

**Figure supplement 1.** Isolation of luminal and myoepithelial lineages.

**Figure supplement 2.** Luminal and myoepithelial lineages in organoids and human mammary epithelial cell cultures.

test, KS p<0.0001) (*Figure 1E*). We found that 76% of LEP-specific genes and 63% of MEP-specific genes had higher fold changes between lineages in younger cells compared to older cells (*Figure 1F*). These percentages indicated loss of lineage fidelity was not restricted to genes that lost lineage-specific expression. Indeed, within the subset of genes for which lineage-specific DE was maintained with age by significance threshold, the majority—67% of LEP-specific genes and 54% of MEP-specific genes—still showed larger fold differences between LEPs and MEPs in younger women (*Figure 1G*). These data expand on our earlier findings that demonstrated loss of lineage fidelity in a limited set of lineage-specific probes (*Miyano et al., 2017*). Here, we establish a statistical definition of loss of lineage fidelity and we underscore the genome-wide nature of this phenomenon whereby gene expression differences that distinguish the major epithelial lineages of the breast decrease with age.

## Loss of lineage fidelity with age leads to disrupted inter-lineage signaling

Because loss of lineage-specific expression could upset the relative balance of ligands and receptors in each lineage, we explored how loss of lineage fidelity could lead to disrupted or dysregulated cell-cell communication between neighboring cell types (*Figure 2A*). We defined the breast interactome as a set of possible ligand-receptor interactions between epithelial cell populations based on the DE of cell-specific ligands and their cognate receptors in younger women, and further defined ligand and receptor pathways mediated by cell-cell signaling through functional enrichment analysis (*Figure 2Ai*).

Using published ligand-receptor pairs (LRPs) (*Ramilowski et al., 2015*), we identified 224 candidate lineage-specific LRPs in young LEPs and MEPs (*Figure 2—source data 1A*) based on the DE of 62 LEP-specific and 66 MEP-specific ligands, and 45 LEP-specific and 47 MEP-specific cognate receptors (*Figure 2—figure supplement 1A*). Protein-protein interaction (PPI) functional enrichment of lineage-specific LRPs identified top KEGG canonical biological pathways (FDR p<0.001) (*Figure 2—figure supplement 1B and C*), including ligands and receptors related to cytokine-cytokine receptor interaction, PI3K-Akt, MAPK and Rap1 signaling commonly enriched in LEPs and MEPs. Enrichment of cytokine, immune, and infection-related pathways further suggested lineage-specific interactions between epithelial and immune cells. LEP-specific LRPs were enriched for cell adhesion molecules (CAMs) involved in cell-cell and cell-extracellular matrix (ECM) interactions and axon guidance molecules (AGMs), while MEP-specific LRPs were enriched for ECM-receptor interaction and focal adhesion LRPs.

Next, we assessed age-dependent dysregulation of cell-cell communication through loss of lineage-specific expression of a cognate receptor (*Figure 2Aii*) or its respective ligand (*Figure 2Aiii*). Loss of lineage fidelity with age led to disruption of 74 LRPs based on the loss of lineage-specific expression of ligands and/or their cognate receptors (*Figure 2B*, *Figure 2—source data 1B*). For each lineage, we considered KEGG canonical biological pathways that were likely to exhibit dysregulated signaling (FDR p<0.01) (*Figure 2—figure supplement 1D and E*). Using our functional enrichment

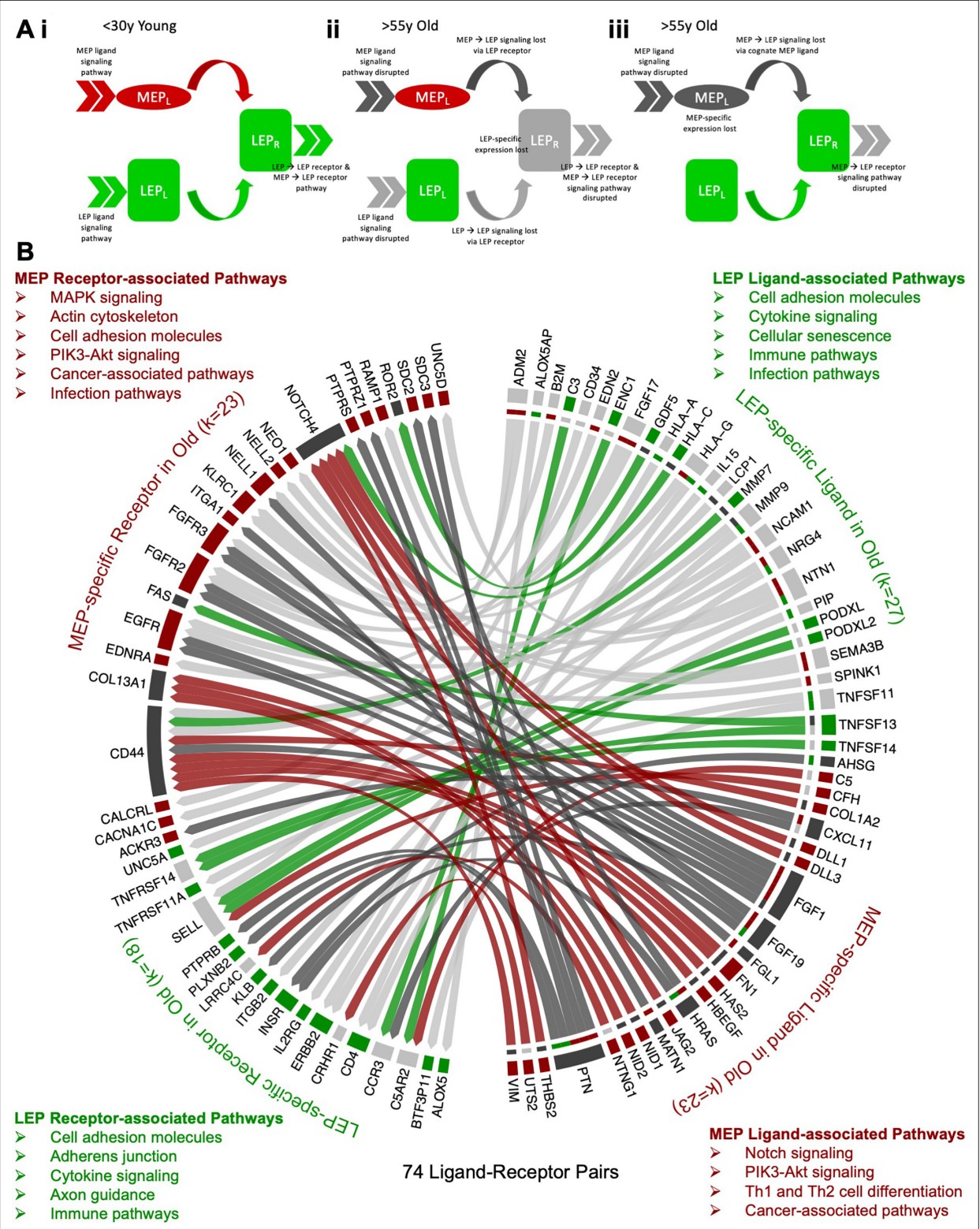

**Figure 2.** Loss of lineage fidelity with age leads to disrupted lineage-specific signaling. (**A**) Schematic illustrating cell-cell communication and dysregulated lineage-specific signaling with age either through loss of lineage-specific expression of the ligand (L) or its cognate receptor (R). (**i**) MEP_L→ LEP_R and LEP_L→ LEP_R lineage-specific signaling and active pathways (>>) associated with both ligands and receptors in <30 y young cells; (**ii**) LEP-specific expression of the receptor is lost with age leading to disrupted MEP_L→ LEP_R and LEP_L→ LEP_R lineage-specific signaling. LEP receptor pathway, as well as MEP ligand and LEP ligand pathways are detected as dysregulated by functional enrichment analysis methods; and (**iii**) Lineage-specific

*Figure 2 continued on next page*

*Figure 2 continued*

expression of the MEP ligand is lost with age and MEP$_L$→ LEP$_R$ lineage-specific signaling is disrupted. MEP ligand pathway is dysregulated. MEP-directed LEP receptor pathway is also dysregulated as cell-cell signaling homeostasis is shifted and only LEP → LEP signaling is driving the LEP receptor pathway. (**B**) Interactome map of DE lineage-specific ligand-receptor pairs (LRPs) (adj. p<0.001, abs(lfc) ≥1) that show loss of lineage-specific expression of either ligands and/or their cognate receptors in older LEPs (light gray) or MEPs (dark gray). LRPs are connected by chord diagrams from the cell type expressing the ligand (L) to the cell type expressing the cognate receptor (R). Number of LRPs, and genes (k) in each category annotated. Summary of functionally enriched KEGG pathways (FDR p<0.001) associated with loss of lineage-specific DE in ligands and receptors are shown.

The online version of this article includes the following source data and figure supplement(s) for figure 2:

**Source data 1.** Loss of lineage fidelity with age leads to disrupted lineage-specific signaling.

**Figure supplement 1.** Loss of lineage fidelity with age leads to disrupted lineage-specific signaling.

approach, we took into account not only direct disruption of respective receptor-associated pathways (e.g., *Figure 2Aii*) or ligand-associated pathways (e.g., *Figure 2Aiii*) due to loss of lineage-specific expression, but also the effect on corresponding pathways of its cognate pair through loss of its signaling partner (e.g., *Figure 2Aii*) or loss of cell-cell signaling homeostasis via dysregulation in the balance of LRPs (e.g., *Figure 2Aiii*). Loss of lineage-specific expression of LEP LRPs with age was enriched for canonical pathways involved in (i) cell-cell and cell-ECM interactions, including CAMs, AGMs, and adherens junctions, and (ii) cytokine, immune, and infection-related pathways. Loss of lineage-specific expression of MEP LRPs with age was associated with (i) pathways involved in cancer; (ii) pathways involved with MAPK, EGFR, NOTCH, and PI3K-AKT signaling; and (iii) MEP contractility. Loss of lineage fidelity with age has the potential to affect a wide range of biological processes that regulate lineage-specific function and signaling, including potential dysregulation of cancer-related processes and immune-specific signaling by the epithelia.

## Models of loss of lineage fidelity in breast epithelia

To understand the changes within each cell population that contributed to the observed loss of lineage fidelity, we explored two models that could explain the decrease in DE between LEPs and MEPs with age. The first model took into account age-dependent directional changes either through stereotypic upregulation or downregulation of gene expression that led to a loss of lineage-specific expression—for example LEPs acquire MEP-like expression patterns and/or MEPs acquire LEP-like expression patterns in the older cohort (*Figure 3Ai*). The second model considered aging-associated increases in variances in the expression of lineage-specific genes in LEPs and/or MEPs from older women that led to a loss of detection of DE between lineages (*Figure 3Aii*). We describe the contributions of each in the following sections.

## The luminal lineage is a hotspot for age-dependent directional changes

There was an extreme lineage bias in the numbers of DE genes between younger and older cells, with the majority of age-dependent changes occurring in LEPs (*Figure 3—source data 1A and B*). In LEPs, 471 genes were DE as a function of age, in contrast to only 29 DE genes in MEPs (adj. p<0.05) (*Figure 3B*). Only five genes showed age-dependent changes that were lineage independent, showing DE associated with age in both LEPs and MEPs—*LRRC4, PSORS1C1,* and *SCNN1B* upregulated in older epithelia, and *ZNF518B* and *ZNF521* downregulated in older epithelia (*Figure 3—figure supplement 1A*), thus leaving only 24 genes that changed with age exclusively in MEPs. That stereotypic directional changes associated with aging were almost exclusively found in LEPs suggests that this lineage could serve as a primary indicator of aging—a proverbial canary in the coalmine.

Age-dependent differential upregulation (251 genes) and downregulation (220 genes) of LEP gene expression (adj. p<0.05) occurred at comparable frequencies (*Figure 3C*). In LEPs, 82% of the genes that were DE changed in a direction towards acquiring MEP-like patterns with age (*Figure 3C*). Although changes in MEPs were far fewer, we note that shifts in expression in older MEPs led to more LEP-like patterns (*Figure 3—figure supplement 1B*). Validation in primary organoids showed that expression of these age-dependent DE genes in LEPs and MEPs (k=495 genes) robustly clustered primary FACS enriched epithelial cells by lineage and age group (AU p≥0.95) (*Figure 3—figure supplement 1C*).

Because dysregulation of regulatory factors like transcription factors (TFs) could lead to further dysregulation of downstream targets, we compared TF expression between younger and older cells

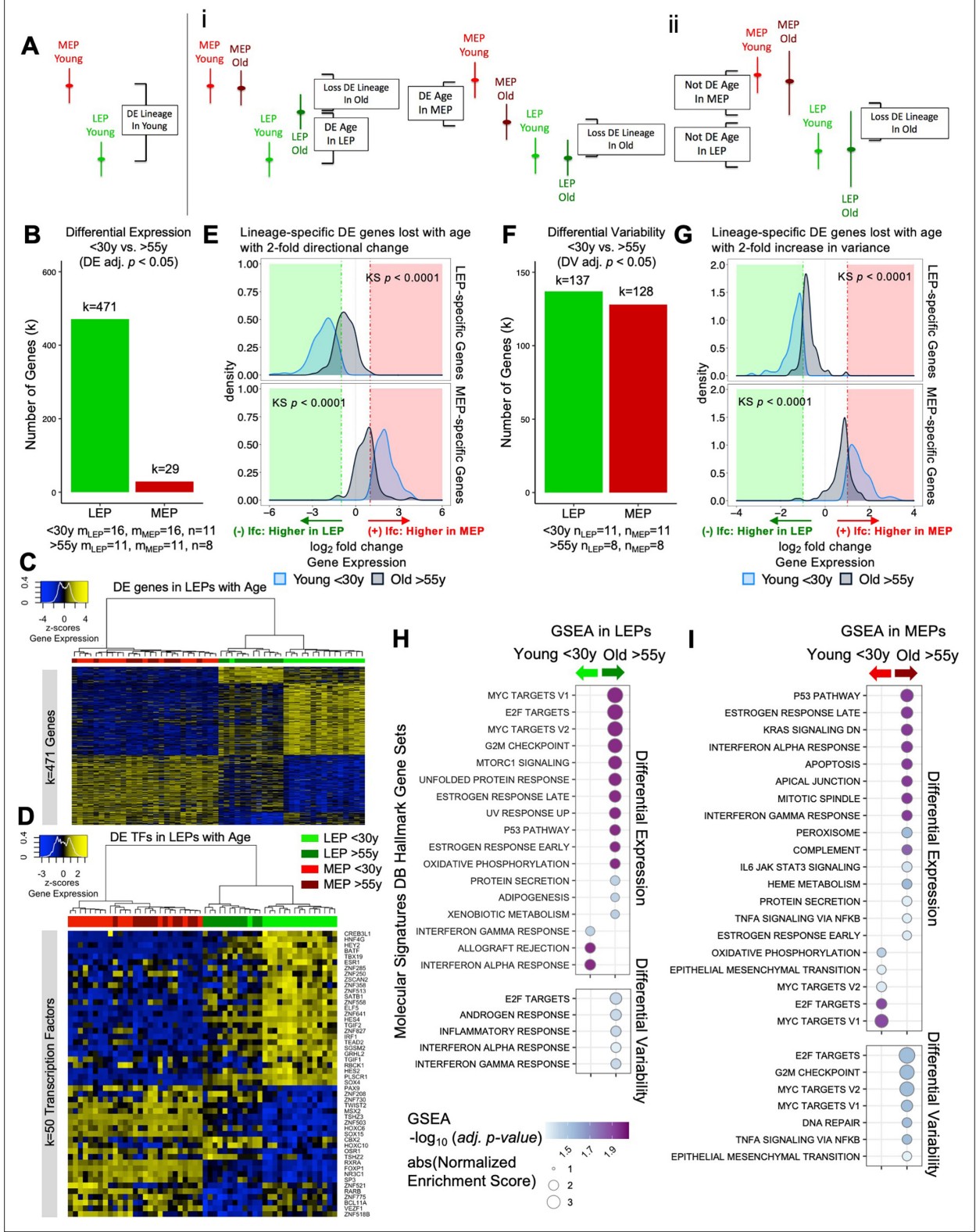

**Figure 3.** The luminal lineage is a hotspot for age-dependent directional changes. (**A**) Models of loss of lineage fidelity illustrate hypothesized mechanisms leading to loss of lineage fidelity: (**i**) Age-dependent DE shifts in gene expression in LEPs and/or MEPs of older relative to younger cells; or (**ii**) An increase in gene expression variance in older LEPs and/or MEPs that lead to loss of detection of lineage-specific DE between LEPs and MEPs with age. (**B**) Number of DE genes (adj. p<0.05) between younger and older LEPs or MEPs. Number of subjects (n) and sample replicates (m) in DE analysis annotated. (**C–D**) Hierarchical clustering of all LEP and MEP samples based on sample-level expression of age-dependent (**C**) DE genes in LEPs (adj.

*Figure 3 continued on next page*

*Figure 3 continued*

p<0.05) and (**D**) DE transcription factors in LEPs (adj. p<0.05). Number of DE genes (k) indicated. Gene expression scaled regularized log (rlog) values are represented in the heatmap; clustering performed using Euclidean distances and Ward agglomerative method. (**E**) Distribution of lfc in expression between LEPs and MEPs in younger and older women for LEP-specific (top panel) or MEP-specific (bottom panel) genes that are lost with age (DE adj. p<0.001, abs(lfc) ≥1) and that have at least a two-fold age-dependent directional change in the older cohort. KS p-values annotated. (**F**) Number of DV genes (adj. p<0.05) between younger and older LEPs or MEPs. Number of subjects (n) in DV analysis annotated. (**G**) Distribution of lfc in expression between LEPs and MEPs in younger and older women for LEP-specific (top panel) or MEP-specific (bottom panel) genes that are lost with age (DE adj. p<0.001, abs(lfc) ≥1) and that have at least a two-fold age-dependent increase in variance in the older cohort. KS p-values annotated. (**H–I**) MSigDB Hallmark gene sets identified by GSEA to be enriched (adj. *p<0.05*) in younger and older (**H**) LEPs and (**I**) MEPs based on age-dependent DE (top) or DV (bottom).

The online version of this article includes the following source data and figure supplement(s) for figure 3:

**Source data 1.** Age-dependent directional and variant responses in the luminal and myoepithelial lineage.

**Source data 2.** Enriched pathways associated with age-dependent directional and variant responses in the luminal and myoepithelial lineages.

**Figure supplement 1.** The luminal lineage is a hotspot for age-dependent directional changes.

**Figure supplement 2.** Comparisons of age-dependent directional and variant responses in the luminal and myoepithelial lineages with obesity and parity signatures.

**Figure supplement 3.** Aging-associated increase in variance contributes to loss of lineage fidelity.

---

in each lineage. Expression of key TFs (*Lambert et al., 2018*) was significantly altered in older cells. Fifty TFs showed age-dependent DE in LEPs and 4 TFs showed DE in MEPs (adj. p<0.05); the majority of these TFs have known roles in breast cancer progression. Of the DE TFs in LEPs, 88% changed expression towards MEP-like expression in older LEPs (*Figure 3D*). These included TFs that were highly expressed in younger LEPs and were downregulated with age, such as the LEP-specific TF *ELF5* (*Miyano et al., 2021*), as well as *GRHL2*, *SGSM2*, *HES4*, *ZNF827*, and the genome organizer *SATB1* (*Figure 3—figure supplement 1Di-v*). Loss of *GRHL2* and *SGSM2* was associated with downregulation of E-cadherin and epithelial-to-mesenchymal transition (EMT) in mammary epithelial cells (*Lin et al., 2019*; *Xiang et al., 2012*). *HES4* is a canonical target gene of Notch1 that plays an important role in normal breast epithelial differentiation and cancer development (*Kontomanolis et al., 2018*). *ZNF827* mediates telomere homeostasis through recruitment of DNA repair proteins (*Vilas et al., 2018*), and *SATB1* has genome organizing functions in stem cells and tumor progression (*Kohwi-Shigematsu et al., 2013*).

Several TFs also gained expression in older LEPs, including: *SP3* and *ZNF503* (*Figure 3—figure supplement 1Dvi-vii*). *SP3* silencing inhibits Akt signaling and breast cancer cell migration and invasion (*Mansour, 2021*). *ZNF503* inhibits *GATA3* expression, a key regulator of mammary LEP differentiation, and its downregulation is associated with aggressive breast cancers (*Kouros-Mehr et al., 2006*; *Shahi et al., 2017*). Age-dependent dysregulation of TFs in LEPs may drive larger-scale changes through TF binding of gene regulatory regions of downstream targets in older LEPs.

To investigate how age-related changes affect lineage fidelity in LEPs (*Figure 3Ai*), we analyzed the overlap between age-dependent DE genes and genes that showed loss of lineage-specific expression. We found that only 9% of the lineage-specific DE loss was attributable to age-dependent DE in either LEPs or MEPs (adj. p<0.05). Expanding our criteria to include genes with at least a two-fold change in DE due to age, we found that these age-related changes accounted for only 21% of the observed loss in lineage-specific expression. This led to a notable reduction in the difference in expression levels between LEP- and MEP-specific genes in older cells (*Figure 3E*). Further analysis showed that the age-related DE genes in LEPs and MEPs did not overlap with genes associated with obesity, parity, or time since full-term pregnancy in breast tissue (*Burkholder et al., 2020*; *Santucci-Pereira et al., 2019*; *Figure 3—figure supplement 2A and B*). Collectively, although these results indicate that age-dependent DE changes do contribute to the loss of lineage fidelity, they do not fully account for it.

## Aging-associated increase in variance contributes to loss of lineage fidelity

Next, we explored the alternate model that incorporated measures of variance as an explanation for the loss of lineage-specific expression in older epithelia (*Figure 3Aii*). Gene expression means and variances of LEPs and MEPs from younger cells were categorized into quantiles, and corresponding

categories in older cells were then assessed (*Figure 3—figure supplement 3A–D*). Gene expression means shifted minimally between younger and older cells, whereas shifts in variances occurred at a higher frequency. Although the dynamic ranges of gene expression in LEPs and MEPs changed as a function of age, these changes were not stereotyped across individuals—that is different aged individuals had different sets of genes that deviated from the range of expression seen in younger samples.

Differential variability (DV) analysis identified 137 genes in LEPs and 128 genes in MEPs that had significant age-dependent DV (adj. p<0.05) (*Figure 3F, Figure 3—source data 1C and D*). Twelve regulatory TFs in either LEPs or MEPs that had tuned windows of expression in younger cells were dysregulated in older cells through a significant increase in variance (adj. p<0.05) (*Figure 3—figure supplement 3E and F*). These TFs included *EHF, KDM2B, HES4, MYCL, GLI1,* and *DMRTA1* in LEPs and *HES6* in MEPs (*Figure 3—figure supplement 3Gi-vii*). The luminal-specific TF *EHF* is a target of *GRHL2* that is conserved in luminal breast cancer cell lines (*Wang et al., 2023*). *KDM2B (FBXL10)* is a histone demethylase ZF-CxxC protein that binds unmethylated CpG-rich DNA. *HES4*, which was also DE with age, is a known Notch target. The proto-oncogene *MYCL* promotes progression of triple negative breast cancers through activation of JAK/STAT3 pathway (*Jiang et al., 2022*). *GLI1* activates the hedgehog pathway in mammary stem cells (*Bhateja et al., 2019*). *DMRTA1* enables sequence-specific double-stranded DNA binding activity. Last, estrogen-regulated *HES6* enhances proliferation of breast cancer cells (*Hartman et al., 2009*). As with DE, genes that were DV with age in LEPs and MEPs were distinct from genes reported to be associated with obesity, parity, and time since full-term pregnancy in breast tissue (*Burkholder et al., 2020; Santucci-Pereira et al., 2019; Figure 3—figure supplement 2C and D*).

These analyses suggested that age-dependent variability in expression across individuals can lead to differential outcomes as different downstream targets could be modulated in different individuals.

To determine how our model of age-dependent variability affected lineage specificity, we focused on genes that showed at least a two-fold increase in variance in the older cohort and that lost lineage-specific expression with age (*Figure 3G*). Genes that had two-fold increases in variances with age explained 27% of the observed loss of lineage-specific expression events, on a par with the proportion (21%) explained by genes that had two-fold changes in DE. Both of our models of directional and variant changes with age led to a significant decrease in the differential magnitude of LEP- and MEP-specific expression in the older cells (*Figure 3E and G*).

Together, these analyses show that increased variances in transcription are considerable drivers of the loss of lineage fidelity in breast epithelia. The observed variances across the older cohort may underlie the age-dependent dysregulation of susceptibility-associated biological processes in specific individuals.

## Hallmark pathways associated with cancer are dysregulated with age in luminal and myoepithelial lineages

Gene set enrichment analysis (GSEA) identified hallmark gene sets that were dysregulated with age, including gene sets known to be dysregulated in breast cancers that were enriched in older LEPs and MEPs (*Figure 3H–I, Figure 3—source data 2A–D*).

Seventeen hallmark gene sets were significantly modulated in LEPs (adj. p<0.05) based on DE (*Figure 3H*, top). Three immune-related gene sets were enriched in younger LEPs and included genes upregulated in response to interferon IFN-alpha and -gamma, and during allograft rejection. In contrast, 14 gene sets were enriched in older LEPs, which included genes regulated by MYC, genes encoding cell-cycle-related targets of E2F TFs and involved in the G2/M checkpoint, genes upregulated by mTORC1 complex activation and during unfolded protein response, and genes involved in the p53 and protein secretion pathways.

Twenty hallmark gene sets were significantly modulated in MEPs (adj. p<0.05) based on DE (*Figure 3I*, top). Five gene sets were enriched in younger MEPs, including MYC and E2F targets and genes defining EMT. In contrast, 15 gene sets were enriched in older MEPs, including genes involved in p53 pathways; genes downregulated by KRAS activation; genes mediating programmed cell death by caspase activation (apoptosis); immune-related gene sets upregulated in response to IFN-alpha and IFN-gamma, and by IL-6 via STAT3; genes regulated by NF-kB in response to TNF and genes encoding components of the innate complement system; and genes encoding components of apical junction complexes.

Five gene sets were significantly modulated (adj. p<0.05) based on DV and were enriched in older LEPs (*Figure 3H*, bottom). These included E2F targets that were similarly enriched via DE, genes defining responses to inflammation, and genes upregulated in response to IFN-alpha and IFN-gamma—gene sets that in contrast were enriched via DE in younger LEPs. Seven gene sets were significantly modulated (adj. p<0.05) based on DV and enriched in older MEPs (*Figure 3I*, bottom). These included genes involved in DNA repair and G2/M checkpoint; genes regulated by NF-kB in response to TNF (a gene set similarly enriched via DE); as well as MYC targets, E2F targets, and genes defining EMT—gene sets that in contrast were enriched via DE in younger MEPs.

Several enriched gene sets were involved in processes that were disrupted with age either via DE or DV, and such overlaps likely suggest integration of directional and variant responses and reflect their convergent impact in common biological processes. Furthermore, the divergence in the age-dependent DE and DV enrichment of cellular processes, such as MYC gene targets and genes involved in immunomodulatory signaling, suggests the genes that become variable with age are associated with pathways that are otherwise important in maintaining lineage-specificity and -function in younger cells.

## Age-dependent gene expression changes detected at the single cell level

To further validate our findings, three publicly available scRNA-seq datasets from non-cancer primary human mammary gland organoids (*Murrow et al., 2022*; *Nee et al., 2023*; *Pal et al., 2021*) were preprocessed to select for identified epithelial cell types and subjected to dimensionality reduction (*Figure 4A–C*, *Figure 4—figure supplement 1A–C*). These studies independently described three epithelial cell types that were consistent with the well-established epithelial lineages in literature identified from histological and functional assays, and sequencing studies of FACS isolated cell populations from normal breast tissue. To discuss the epithelial lineages across the three different scRNA-seq datasets, we adopted the Luminal1 (inclusive of luminal progenitors), Luminal2 (luminal hormone-sensing), and Basal (basal-myoepithelial) terminology from Nee et al. (*Figure 4A–C*, top left panel). The canonical LEP marker *MUC1* (CD227) was highly expressed in Luminal1 and Luminal2 cell types, with the Luminal1 cells also expressing LEP marker *PROM1* (CD133) and Luminal2 cells expressing androgen receptor (*AR*), while MEP markers *NGFR* (CD271) and *MME* (CD10) were highly expressed in the Basal cell type (*Figure 4A–C*). Single-cell GSEA (scGSEA) performed using UCell (*Andreatta and Carmona, 2021*) further showed corresponding cell type-enrichment of our LEP-specific and MEP-specific gene sets (*Figure 1—source data 3A*, *Figure 4—figure supplement 1D–F*). Distributions of cell states were highly associated with specific samples as was described in Murrow et al. and in our reanalysis of the Nee et al. dataset (*Figure 4—figure supplement 1G and H*).

To further validate the observation that the DE genes in older LEPs represent a loss of lineage fidelity, genes DE in LEPs with age (*Figure 3—source data 1A*) were split into two custom gene lists: old LEP signature defined as genes that were upregulated in older LEPs (p<0.05, lfc >0) and young LEP signature defined as genes that were downregulated in older LEPs (p<0.05, lfc <0). scGSEA allowed for each epithelial cell from each dataset to be scored for enrichment of either age-dependent DE gene list (*Figure 4D–F*). The standardized mean difference (*Andrade, 2020*) was used to compare mean signatures between cell types and biological groups and to indicate significance levels.

Basal cells had the greatest magnitude of enrichment for genes upregulated in older LEPs, and luminal cells (Luminal1 and Luminal2) were most enriched for genes downregulated in older LEPs (*Figure 4D–F*). These results confirmed that LEPs acquired gene expression patterns associated with basal cells with age. Enrichment patterns between datasets were consistent, indicating that the signatures were robust enough for more granular analyses.

Because the basal-like nature of older LEPs was highlighted by scGSEA analysis, we further explored whether the old LEP (upregulated in older LEP) and young LEP (downregulated in older LEP) signatures were enriched in clinically defined patient populations. The Pal et al. dataset contained metadata indicating menopausal status, and we found that the premenopausal/postmenopausal dichotomy in scRNA-seq was concordant with the young/old dichotomy evaluated via our lineage-specific RNA-sequencing methodology. Indeed, genes upregulated in older LEPs were most enriched in Luminal1 cells from women who had undergone menopause (*Figure 4D*, left panel). Furthermore,

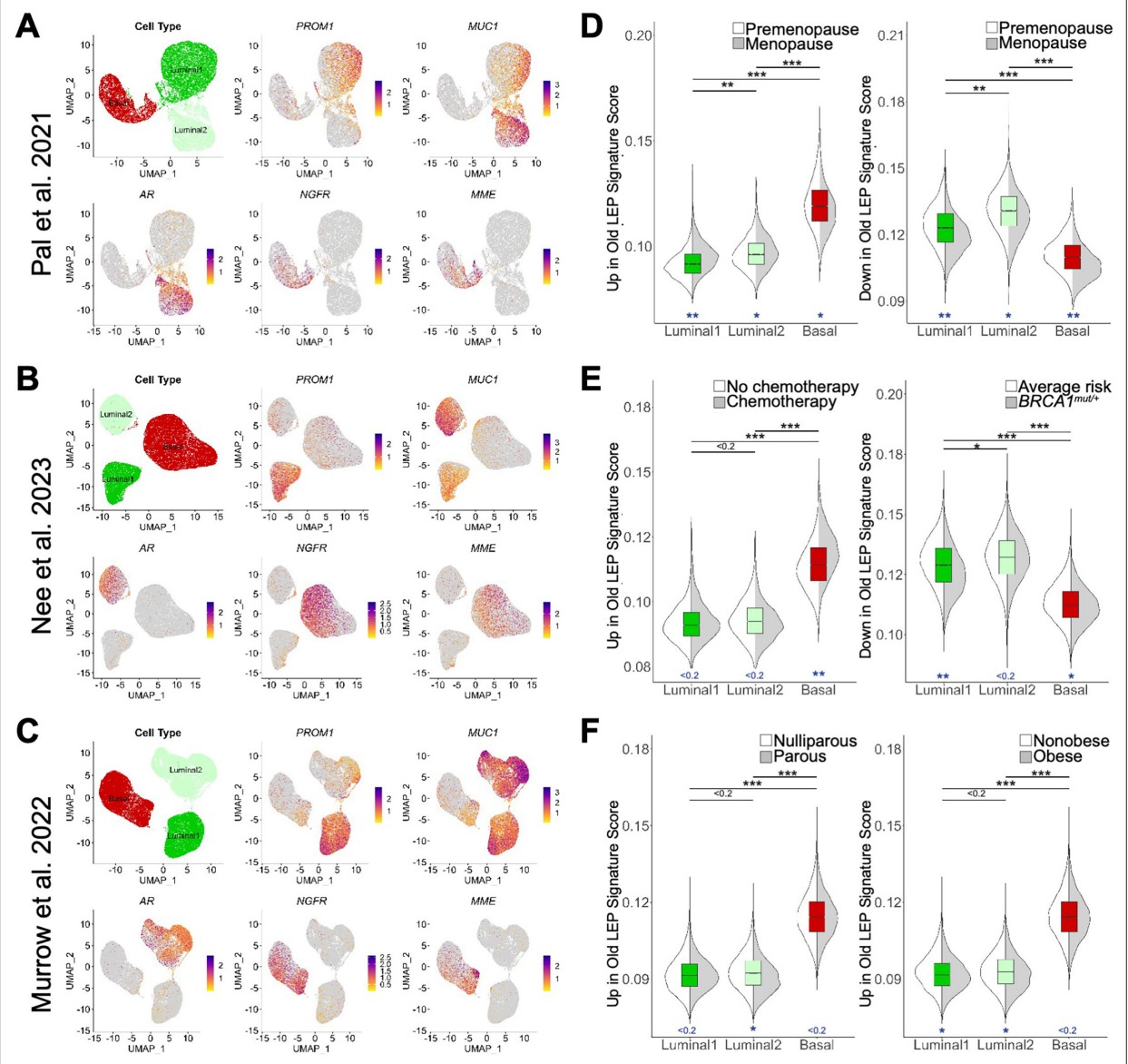

**Figure 4.** Age-dependent gene expression changes detected at the single cell level. (**A–C**) UMAP projections of scRNA-seq data from (**A**) 13 non-tumorigenic breast tissue samples (19-69y) from reduction mammoplasties via (**Pal et al., 2021**), (**B**) 11 non-tumorigenic non-carrier (24-50y) breast tissue samples from reduction mammoplasties, prophylactic mastectomies, and contralateral to DCIS/tumor via (**Nee et al., 2023**), and (**C**) 28 healthy reduction mammoplasty tissue samples (19-39y) via (**Murrow et al., 2022**). Reanalysis of scRNA-seq data focused on three identified epithelial cells types, with UMAP projections showing single cells identified as Luminal1 (luminal adaptive secretory precursors), Luminal2 (luminal hormone-sensing), or Basal (basal-myoepithelial) (top left panel), and normalized expression levels of LEP-specific markers *PROM1* and *MUC1* (top middle and right panels), ER + LEP-specific marker *AR* (bottom left panel), and MEP-specific markers *NGFR* and *MME* (bottom middle and right panels). (**D–F**) Distribution of scGSEA signature scores capturing enrichment of age-dependent DE gene sets across cell types (boxplots) and clinical conditions (split violin plots). Distribution of signature scores showing enrichment of age-dependent DE genes (**D**) upregulated (left) or downregulated in old LEPs across cell types and between premenopausal and menopausal women via (**Pal et al., 2021**); (**E**) upregulated in old LEPs between women with cancer who received chemotherapy and those that did not (left), or downregulated in old LEPs between non-tumorigenic BRCA1 mutation carriers and non-carriers (right) via (**Nee et al., 2023**); and (**F**) upregulated in old LEPs between parous and nulliparous (left), or obese and nonobese (right) individuals via (**Murrow et al., 2022**). Standardized mean difference was used to compare mean signatures between cell types and biological groups, and to indicate significance levels.

The online version of this article includes the following figure supplement(s) for figure 4:

**Figure supplement 1.** Lineage-specific gene expression changes detected at the single cell level.

genes downregulated in older LEPs (upregulated in younger LEPs) were downregulated in Luminal1 cells and Basal cells (*Figure 4D*, right panel) from postmenopausal women.

Accelerated aging in cancer patients who receive chemotherapy and individuals who carry high-risk germline mutations is becoming increasingly appreciated (*Shalabi et al., 2021*; *Siddique et al., 2021*). We further interrogated the Nee et al. dataset which included women who have undergone chemotherapy as well as women who carry *BRCA1* mutations. We found chemotherapy increased the magnitude of enrichment of the old LEP signature in Basal cells (*Figure 4E*, left panel). Interpreting this shift was challenging until pathways related to aged luminal cells were considered (*Figure 3H*). Basal cells appeared to be most sensitive to the effects of chemotherapy or to potentially have the most durable perturbations (time since treatment was not reported) to pathways related to MYC, G2/M checkpoints, and p53. Our findings lend further credence to loss of lineage fidelity and accelerated aging in women with *BRCA1* mutations as Luminal1 cells from these women did not look as young as their age-matched average risk counterparts (*Figure 4E*, right panel) and resembled menopausal cells (*Figure 4D*, right panel).

Parity and obesity are often discussed in the context of breast cancer susceptibility. To address how they may affect the aging signature, we used the Murrow et al. dataset. Our scRNA-seq analysis revealed that enrichment of the aging LEP gene expression signature in Luminal and Basal compartments was minimally affected by parity or obesity (*Figure 4F*), reinforcing our findings in bulk tissue (*Figure 3—figure supplement 2A*).

Thus, the aging signature we developed from enriched LEPs extends to single-cell studies of epithelial cell types and provides valuable insight towards analysis of these datasets. Moreover, we showed that our aged luminal signature was enriched in women who received chemotherapy and those with increased risk of developing breast cancer, possibly capturing the accelerated aging phenotypes in these groups.

## Age-dependent directional changes in the luminal lineage are indicators of aging breast tissue

Because LEPs dominated the age-specific signal amongst epithelia, we examined if the age-dependent DE contribution of the luminal lineage was detectable in bulk normal primary breast tissue (GSE102088, n=114) (*Song et al., 2017*). Genome-wide analysis identified 97 genes that were DE between younger <30 y and older >55 y tissues (adj. p<0.05) (*Figure 5—source data 1A*); the relatively smaller number of genes compared to age-dependent DE observed in LEPs was likely due the cellular heterogeneity found in bulk tissue. To characterize the contribution of the LEP lineage to aging biology of the breast, we performed GSEA to assess enrichment of LEP-specific age-dependent DE genes at the tissue level. We found significant enrichment of differentially upregulated genes identified in young LEPs in tissue from younger women (adj. p=0.012) (*Figure 5A*) and differentially upregulated genes identified in old LEPs in tissue from older women (adj. p=0.006) (*Figure 5B*). These GSEA results indicate that although age-dependent changes in other cell populations may confound detection of the LEP-specific signal, age-dependent changes in LEPs were still prominent in bulk tissue.

We then explored the GSEA leading-edge genes—genes that made the largest contribution to the significant enrichment of the LEP-specific age-dependent genes in bulk tissue. Of the leading-edge genes, genome organizer *SATB1*, which showed significant LEP-specific expression relative MEPs in young and old women (adj. p<0.0001, lfc ≤-2) (*Figure 5Ci*), had the strongest age-dependent signal in bulk tissue (*Figure 5Cii*). *SATB1* was significantly downregulated in both LEPs and breast tissue of older relative to younger women (adj. p<0.05) (*Figure 5Ci–ii*). We also detected this decrease in *SATB1* expression with age in normal breast tissue of women with cancer in The Cancer Genome Atlas (TCGA) cohort (n=111, Wilcoxon adj. p<0.001) (*Figure 5—figure supplement 1A*). In TCGA breast cancers (n=1,089), PAM50 Luminal A (LumA), Luminal B (LumB), and Her2-enriched (Her2) breast cancer subtypes had the lowest expression of *SATB1* relative to PAM50 Basal-like (Basal) and Normal-like (Normal) intrinsic subtypes (post hoc Wilcoxon adj. p<0.0001) (*Figure 5Ciii*). Moreover, we found that for primary tumors that had matched normal tissue in TCGA (n=114 tumor and n=109 normal), *SATB1* was significantly downregulated in PAM50 LumA, LumB and Her2 breast cancers relative to their matched normal tissue (*Figure 5—figure supplement 1B*), suggesting further *SATB1* dysregulation during cancer initiation. To understand the role of *SATB1* in regulating the aging biology of the

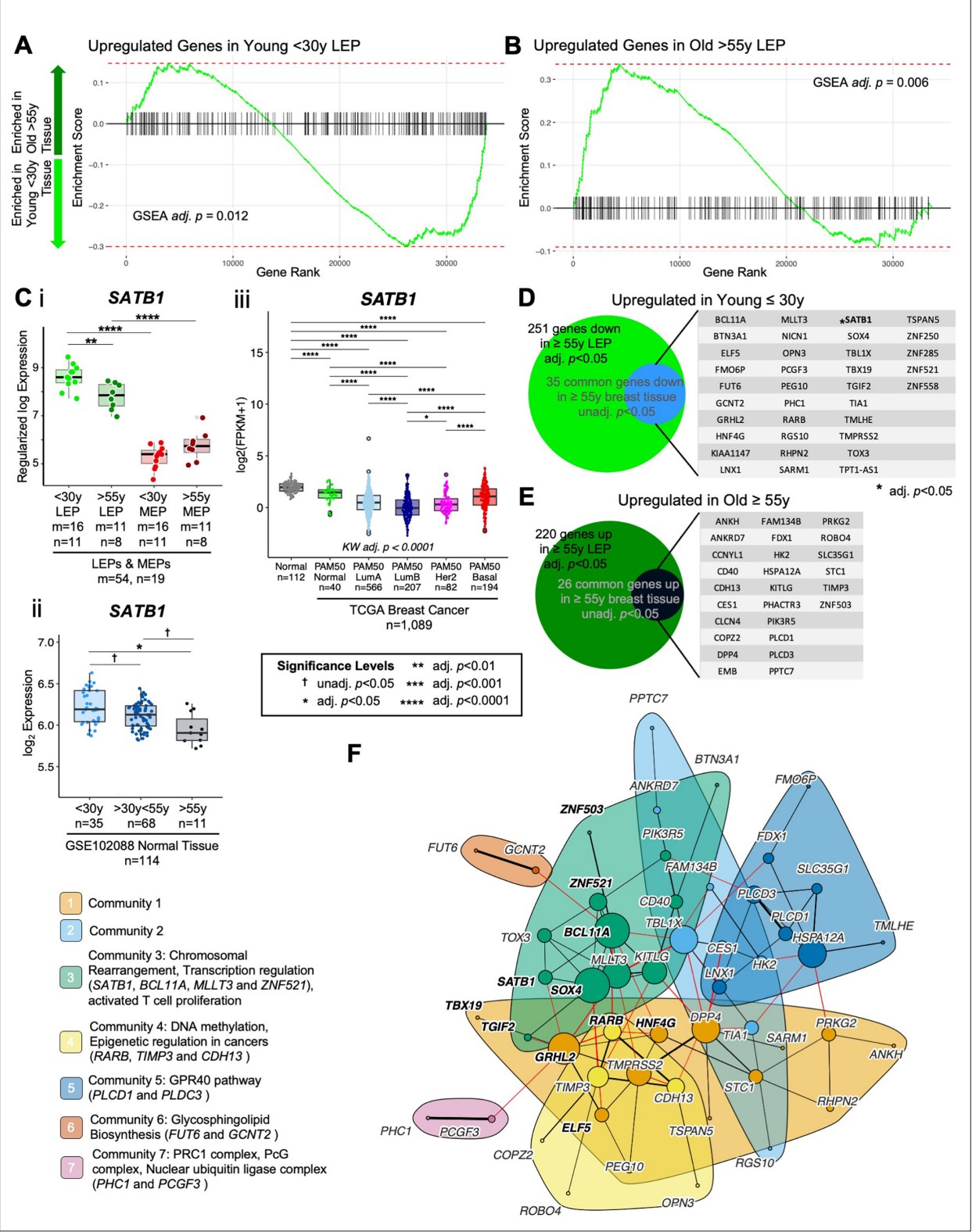

**Figure 5.** Age-dependent directional changes in the luminal lineage are indicators of aging breast tissue. (**A–B**) GSEA enrichments plots from age-dependent DE analysis of bulk tissue showing gene ranks based on DE test statistics and gene set enrichment scores. Age-dependent enrichment of two gene sets composed of (**A**) differentially upregulated genes in younger <30 y LEPs and (**B**) differentially upregulated genes in older >55 y LEPs in bulk tissue are shown. Negative enrichment scores indicate upregulation of specified gene set in tissue from younger <30 y women, while positive

*Figure 5 continued on next page*

*Figure 5 continued*

enrichment scores indicated upregulation in tissue from older >55 y women. GSEA enrichment BH adj. p-values are annotated. (**C**) Boxplots of *SATB1* gene expression: (**i**) subject-level rlog values in LEPs and MEPs of younger and older women; (**ii**) log$_2$ values in normal breast tissue (GSE102088); and (**iii**) log$_2$ FPKM values in the TCGA breast cancer cohort by PAM50 subtype in cancers and in matched normal tissue. Age-dependent DE adj. p-values in LEPs and normal breast tissue, and lineage-specific DE adj. p-values in LEPs and MEPs are indicated (**i–ii**). Kruskal-Wallis (KW) test performed across PAM50 breast cancer subtypes as well as matched normal tissue; p-values adjusted across all age-dependent DE and DV genes identified in LEPs found in the TCGA dataset (**iii**). Post hoc pairwise Wilcoxon test adj. p-value significance levels annotated (*<0.05, **<0.01, ***<0.001, ****<0.0001); p-values adjusted across all pairwise comparisons and across all DE and DV genes identified in LEPs found in the TCGA dataset (**iii**). Number of subjects (n) and sample replicates (m) in each analysis annotated. (**D–E**) Venn diagram of genes with age-dependent DE in LEPs (adj. p<0.05) and at least nominal DE (unadj. p<0.05) in normal primary breast tissue. Genes commonly (**D**) upregulated and (**E**) downregulated in LEPs and bulk tissue with age are listed. (**F**) PPI network of common age-dependent DE genes in LEPs (adj. p<0.05) and bulk tissue (unadj. p<0.05) with TFs annotated in bold. Seven gene communities identified; corresponding network functional enrichment (FDR p<0.05) of selected processes annotated.

The online version of this article includes the following source data and figure supplement(s) for figure 5:

**Source data 1.** Age-dependent directional changes in the luminal lineage are indicators of aging breast tissue.

**Figure supplement 1.** Age-dependent directional changes in the luminal lineage are indicators of aging breast tissue.

**Figure supplement 2.** Age-dependent directional changes in the luminal lineage across breast cancer subtypes and matched normal tissue in TCGA.

luminal lineage, we explored 778 unique gene targets previously identified to be SATB1-activated and/or repressed in the MDA-MB-231 breast cancer cell line (*Han et al., 2008*). Of these, we identified 515 *SATB1*-target genes in our dataset and 64% had correlated expression (|*R*|≥0.5) in LEP and MEP samples (*Figure 5—figure supplement 1C*). Furthermore, 26 SATB1-target genes showed specific age-dependent DE in the LEP lineage (*Figure 5—figure supplement 1D*), suggesting that *SATB1* regulates in part the aging phenotype observed in LEPs (*Figure 5—figure supplement 1D*). Single-sample GSEA (ssGSEA) of SATB1-activated and SATB1-repressed genes that were DE in LEPs showed enrichment of age-dependent DE SATB1-activated genes in young LEPs compared to old LEPs (adj. p=002) (*Figure 5—figure supplement 1E*). Together, these results suggest that *SATB1*-mediated genome organization may play a regulatory role in the maintenance of the luminal lineage and in the observed genome-wide dysregulation with age and breast cancer.

Because we expected the signal in bulk tissue to be muted due to cellular heterogeneity, we also evaluated leading edge genes that showed nominally significant DE between younger and older tissue (unadj. p<0.05) (*Figure 5—source data 1A*). Of the 251 genes upregulated in younger LEPs, 35 genes (14%) showed nominally significant differential upregulation in young tissue (*Figure 5D*). These genes included EMT-associated *GRHL2* and the LEP-specific TF *ELF5,* which we had previously shown to be predictive of accelerated aging in genetically high risk LEPs (*Miyano et al., 2021*), as well as *ZNF521*, which showed age-dependent downregulation in both LEP and MEP lineages. Of the 220 genes upregulated in older LEPs, 26 genes (12%) showed nominally significant differential upregulation in older tissue (*Figure 5E*), including the GATA3 inhibitor *ZNF503*. Of the 61 genes we identified to be commonly dysregulated between younger and older LEPs and breast tissue, 17 were LEP-specific and 14 were MEP-specific in our lineage-specific DE analysis.

Common age-dependent DE genes between LEPs and bulk tissue showed significant PPI network enrichment (PPI enrichment p=0.014), including a 51-gene network that involved 11 DE TFs (*Figure 5F*) and 10 genes with high connectivity in the network (degree >10) that are potential nodes of integration. These included genes downregulated in the older group: TF *BCL11A*—a subunit of the BAF (SWI/SNF) chromatin remodeling complex (*Kadoch et al., 2013*), TF *SOX4*—involved in determination of cell fate, TF *GRHL2*, and *MLLT3*—a chromatin reader component of the super elongation complex (*Moustakim et al., 2018*); and genes upregulated in the older group: *DPP4 (CD26)*—a cell surface receptor involved in the costimulatory signal essential T-cell activation (*Ikushima et al., 2000*), *HSPA12A*—a heat shock protein associated with cellular senescence, and *KITLG*—a ligand for the luminal progenitor marker c-KIT in breast (*Kim and Villadsen, 2018*).

An optimal community structure detection algorithm identified seven gene communities with maximal modularity (*Figure 5F*). Functional network enrichment (FDR <0.05) identified three communities with transcriptional regulatory activity. Community 3 anchored by the TFs *BCL11A* and *SOX4* was enriched for genes associated with transcriptional regulation. *SATB1*, *BCL11A*, *MLLT3*, and *ZNF521* were linked to chromosomal rearrangement and were downregulated in LEPs and breast tissues of older women. These genes also showed breast cancer subtype-specific expression, and *BCL11A* and

*ZNF521*, like *SATB1*, were downregulated in PAM50 LumA, LumB, and Her2 tumors relative to PAM50 Basal tumors and compared to their matched normal tissue (*Figure 5—figure supplement 2i-iii and Bi-iii*). Community 4 members *RARB*, *TIMP3* and *CDH13* have been implicated as tumor suppressor gene targets of DNA methylation and epigenetic regulation in cancers. Community 7 members *PHC1* and *PCGF3* are components of the Polycomb group (PcG) multiprotein polycomb repressor complex (PRC)- PRC1-like complex that is required for maintenance of the transcriptionally repressive state of many genes throughout development. *PHC1* and *PCGF3* were downregulated in LEPs and breast tissues of older women; these genes showed breast cancer subtype-specific expression, and *PHC1* was downregulated in PAM50 LumB and Her2 tumors relative to their matched normal tissue (*Figure 5—figure supplement 2Aiv-v and Biv-v*).

Taken together, genes commonly DE in younger and older LEPs and breast tissue either reflect stereotypic aging-associated molecular changes across different breast cell populations or are driven by LEP-specific changes. This suggests that age-dependent molecular changes in LEPs contribute to essential processes involved in the aging biology of the entire breast and that are dysregulated in cancers.

## Genes encoding for cell-cell junction proteins are dysregulated in aging epithelia

We showed previously that MEPs can impose aging phenotypes on LEPs—LEPs from younger women acquiring expression patterns of older LEPs when co-cultured on apical surfaces of MEPs from older women (*Miyano et al., 2017*). This non-cell autonomous mechanism of aging requires direct cell-cell contact between LEPs and MEPs, suggesting that cell-cell junction proteins play a role in age-dependent dysregulation in LEP-MEP signaling. Indeed, we found that apical junction-associated genes were significantly enriched with age in MEPs (*Figure 3I* top).

We explored age-dependent dysregulation of a curated set of genes encoding for membrane components of adherens junctions, tight junctions, gap junctions, desmosomes, and CAMs in LEPs and MEPs to identify candidate genes that may regulate communication between the lineages (*Figure 6—source data 1A*). Because age-dependent changes involve both DE and DV, we used the non-parametric Lepage test to jointly monitor the central tendency and variability of expression of 198 genes encoding for cell-surface junction proteins between the younger and older cohorts. We found 42 genes were modulated in LEPs and/or MEPs with age (Lepage test p<0.05) (*Figure 6A*, *Figure 6—source data 1B*). These included genes that were modulated via a significant directional change with age such as the desmosomal cadherins genes *DSG3* (desmoglein) and *DSC3* (desmocollin), which are expressed in both LEPs and MEPs (*Garrod and Chidgey, 2008*; *Figure 6—figure supplement 1Ai-ii*), and the genes encoding for essential tight junction components, *CLDN10* and *CLDN11* (*Figure 6—figure supplement 1Aiii-iv*).

We were particularly interested in gene expression of the gap junction protein GJB6 (Connexin-30), which is expressed by both LEPs and MEPs in the normal mammary gland and forms homo- (LEP-LEP) and hetero-cellular (LEP-MEP) channels (*Teleki et al., 2014*), because it showed modulation via an increase in variance in older MEPs (p=0.02) and nominal increase in variance in older LEPs (p=0.06) (*Figure 6B*). Moreover, GJB6 expression was positively associated across isogenic samples of LEPs and MEPs as indicated by linear regression (R=0.5, p=0.008) (*Figure 6—figure supplement 1B*). As such, modulation of *GJB6* provided an avenue for exploring changes that could occur in both lineages. To assess transcriptional regulation of the GJB6 protein, we analyzed ChIP-seq (Cistromics) mammary gland data from The Signaling Pathways Project Ominer database (*Figure 6—figure supplement 1C*). Nine TFs had binding signals within +/-10 kb of the TSS of *GJB6*, including progesterone receptor *PGR*, *MYC* and the LEP-specific TF *ELF5*, which we previously showed to be regulated via direct LEP-MEP interactions in our co-culture studies (*Miyano et al., 2017*).

## Gap Junction protein GJB6 is a mediator of the non-cell autonomous mechanism of aging in breast epithelia co-cultures

Because changes in MEPs were predominantly associated with DV rather than DE, we hypothesized that MEPs from different individuals could exert aging phenotypes on LEPs via different gene regulatory mechanisms that may implicate the DV genes observed in LEPs (*Figure 6C*). As LEPs exhibited the vast majority of age-dependent DE changes, we further hypothesized that LEPs

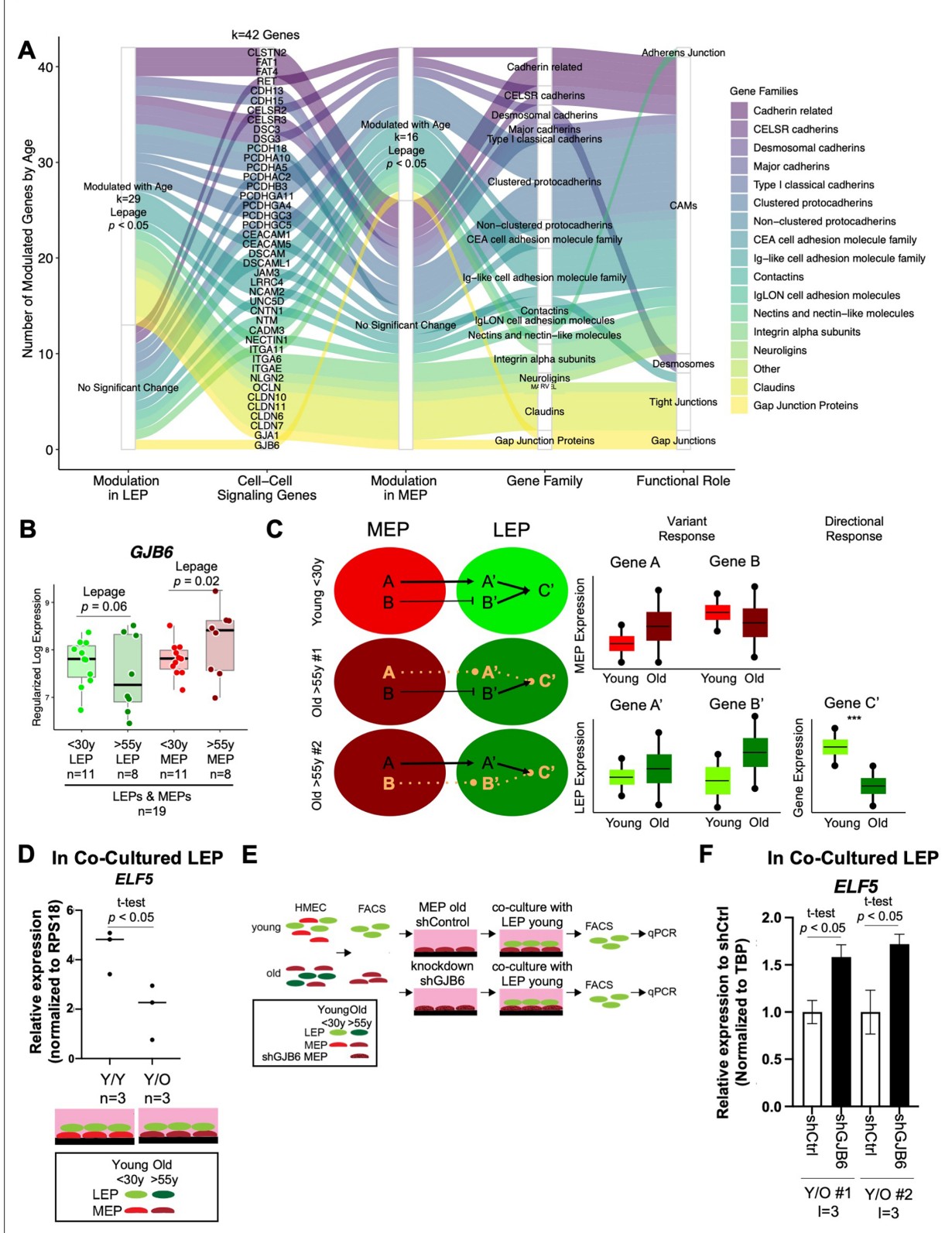

**Figure 6.** Gap Junction protein GJB6 is a mediator of the non-cell autonomous mechanism of aging in breast epithelia co-cultures. (**A**) Age-dependent modulation of apical junction-associated genes in LEPs and MEPs (Lepage test p<*0.05*). Number of age-modulated genes (k) indicated. Genes annotated with their respective HUGO Gene Nomenclature Committee (HGNC) gene family and functional role in adherens junctions, cell adhesion molecules, desmosomes, tight junctions, or gap junctions. (**B**) Boxplot of *GJB6* subject-level rlog expression values in LEPs and MEPs in younger and

*Figure 6 continued on next page*

*Figure 6 continued*

older women. Lepage test p-values are indicated. Number of subjects (n) in analysis annotated. (**C**) Schematic illustrating integration of directional and variant responses in older epithelial cells. Different genes are dysregulated in LEPs and MEPs of older individuals leading to an increase in variance in expression across aged cells. Through cell-cell signaling, variant responses in MEPs (gene A or gene B) lead to variant responses in LEPs (gene A' or gene B'). Where these variant responses integrate and affect common downstream genes in LEPs (gene C') lead to detectable age-dependent directional changes (***) that are seen as stereotyped responses in the lineage. (**D**) Relative expression of *ELF5* in younger LEPs co-cultured with either younger (Y/Y) or older (Y/O) MEPs. Two-tailed t-test p-value indicated. (**E**) Schematic of co-culture methodology with HMEC cells from younger and older women enriched by FACS for LEPs and MEPs; *GJB6* knock-down in older MEP feeder layer by shRNA; younger LEPs are co-cultured on top of the older MEP feeder layer for 10 days; LEPs separated from MEPs by FACS; and LEP expression levels measured by qPCR. (**F**) Relative expression of *ELF5* in younger LEPs co-cultured with either shControl or shGJB6 older MEPs. Two-tailed t-test p-value indicated. Number of subjects (n) and technical replicates (l) annotated.

The online version of this article includes the following source data and figure supplement(s) for figure 6:

**Source data 1.** Genes encoding for cell-cell junction proteins are dysregulated in aging epithelia.

**Figure supplement 1.** Genes encoding for cell-cell junction proteins are dysregulated in aging epithelia.

serve as integration nodes for dysregulation in MEPs where variant changes converge via common pathways that lead to directional changes in genes downstream of these pathways (*Figure 6C*). We identified *ELF5* to be one such target. *ELF5* is a highly LEP-specific TF (*Figure 6—figure supplement 1D*) and is a key TF of alveologenesis that regulates differentiation of progenitor cells towards the luminal secretory lineage (*Lee et al., 2013*). Indeed, in our prior work, we showed that *ELF5* expression was dynamic and responsive to age-dependent microenvironment changes and expression of *ELF5* and published *ELF5*-target genes showed age-dependent correlation in LEPs (*Miyano et al., 2017*), and that *ELF5* served as an independent biological clock in breast (*Miyano et al., 2021*). *ELF5* was downregulated in younger LEPs when co-cultured on apical surfaces of older MEPs for 10 days (*Figure 6D*), concordant with the observed phenomenon of *ELF5* down-regulation in LEPs with age.

A key part of our variant responses hypothesis is that tuned windows of expression is essential for proper function, and deviations either up or down from this range in older individuals could lead to aging-dependent dysregulation. Thus, selective KD of upregulated genes or overexpression of downregulated genes in specific MEP samples that show outlier expression would be ideal; however, as tuning windows of expression are difficult to achieve experimentally by overexpression, our co-culture experiments focused on KD of DV genes in specific old MEP samples where we see upregulated expression above the 75% quantile distribution in young MEP samples. Thus, we asked whether knockdown or inhibition of *GJB6* expression in the older MEPs with the highest expression, relative to younger MEPs, could restore proper signaling between LEPs and MEPs. To test this, we used our established heterochronous co-culture system and measured recovery of LEP expression of *ELF5* as a readout of biological age (*Figure 6E*). MEP cell strains used in KD experiments were specifically selected to be the samples from older women with ~two-fold increase expression of GJB6 that were above the 75% quantile distribution of GJB6 expression in young MEP.

If bringing variant *GJB6* under tighter control prevents chronologically older MEPs from imposing older biological ages in younger LEPs, then *ELF5* levels should not decrease in co-culture. LEPs from younger women were co-cultured for 10 days on older MEPs treated with either shGJB6 or scramble shRNA (shCtrl) (*Figure 6—figure supplement 1E*). When co-cultured on top of older MEP-shGJB6 relative to older MEP-shCtrl, LEPs maintained expression of *ELF5* at higher levels (*Figure 6F*), consistent with the higher expression levels in younger women. LEP-expression of *ELF5* likewise showed a stepwise (although non-significant) increase when older MEP feeder layers were pre-treated with increasing concentrations of a non-specific gap junction inhibitor 18 alpha-glycyrrhetinic acid (18αGA) (*Figure 6—figure supplement 1F*). Thus, reducing *GJB6* expression in older MEPs that overexpressed *GJB6* relative to younger MEPs prevented these older MEPs from imposing an older biological age on co-cultured younger LEPs as determined by *ELF5* expression. Based on these results, we propose that variance is a driver of stereotypical aging phenotypes at the tissue level, and that constraining specific changes caused by an increase in molecular noise during aging, such as in cell-cell communication nodes, may prevent the spread of age-related cues within epithelia.

## Age-dependent dysregulation in LEPs shape predictors of normal breast tissue and PAM50 subtypes

We propose that aging mechanisms operate through at least two distinct pathways: (i) General dysregulatory mechanisms, which are reflected in directional age-dependent changes that share common features with known cancer mechanisms. These changes contribute to an increased overall susceptibility to cancer as individuals age. (ii) Individual-specific dysregulatory mechanisms, which are identified by variations in aged populations. These variations may help explain why certain individuals have greater vulnerability to cancer (not all aged individuals develop cancer) and why those who do develop cancer develop specific subtypes.

Hierarchical clustering and machine learning (ML) algorithms can effectively capture the influence of DV genes. It is worth noting that even DE genes inherently exhibit variability. We suggest that by integrating the variability in aging responses, we can gain further insights that not only enable the identification of key branching points in clustering and the most relevant age-dependent biomarkers in classification but also help explain the varying susceptibility of individuals to specific breast cancer subtypes. Indeed, we found that 98% of age-dependent DE and DV genes in LEPs (580 of 589 genes in the TCGA cohort) were differentially expressed between PAM50 intrinsic subtypes (KW BH adj. p<0.05, post hoc Wilcoxon BH adj. p<0.05) (*Figure 7—source data 1A and B*). When we further analyzed the median expression of DE and DV genes across PAM50 LumA, LumB, Her2, and Basal subtypes, we found differential distribution of median expression across subtypes (Fisher's exact test Bonferroni adj. p<0.05). (i) The largest fraction of DE genes upregulated in younger LEPs had the highest expression in Basal and lowest expression in LumB subtypes. (ii) The largest fraction of DE genes upregulated in older LEPs had highest expression in LumA and lowest expression in Basal subtypes. And (iii), the largest fraction of DV genes with higher variance in older LEP had highest expression in Basal and lowest expression in LumB subtypes. In contrast (iv), DV genes that had higher variance in younger LEPs showed no association between subtype and median expression levels (*Figure 7—figure supplement 1A*).

Our GSEA and literature review of genes that showed age-dependent changes in LEPs revealed enrichment for pathways and genes commonly dysregulated in breast cancers. Unsupervised hierarchical clustering of TCGA primary tumor and matched normal samples (n=1201) based on expression of LEP-derived age-dependent DE and DV genes (k=589 genes) in the TCGA cohort identified four main sample clusters (*Figure 7A*): (i) cluster 1 represented predominantly by PAM50 LumA and Her2 breast cancer subtypes; (ii) cluster 2 by PAM50 LumB and LumA subtypes; (iii) cluster 3 by PAM50 Basal subtype; and (iv) cluster 4 by matched normal samples. This suggests that age-dependent changes in LEPs may reflect dysregulation of biological processes that play a role in tumor initiation in normal tissue and in the etiology of breast cancer subtypes. We therefore assessed whether DE and DV genes that change in LEPs with age can be used as biomarkers that can classify normal tissue from cancer and predict breast cancer subtypes.

Using 75% of TCGA data for training and cross-validation (n=873) (*Figure 7B*), we built an elastic net ML classifier of normal breast tissue and PAM50 breast cancer subtypes based on the expression of age-dependent DE and DV genes identified in LEPs that were represented in our three ML breast tissue datasets (k=536 genes in TCGA, GSE81540, and GTEx). The best performing model selected during cross-validation had a mean balanced accuracy of 0.91, mean sensitivity of 0.86, and mean specificity of 0.96. Our ML classifier proved predictive in the remaining 25% of TCGA test data, which the model had not seen (n=288, mean balanced accuracy = 0.93, mean sensitivity = 0.88, mean specificity = 0.97), and in an independent test set composed of normal tissues from GTEx and breast cancer tissues from GSE81540 (n=3364,, mean balanced accuracy = 0.87, mean sensitivity = 0.77, mean specificity = 0.94) (*Figure 7B–D*). We further assessed performance of our ML model in the two test sets using three measures of the area under the receiver operating characteric curve (AUC) for multi-class prediction: (i) AUC of each group vs. the rest; (ii) micro-average AUC calculated by stacking all groups together; and (iii) macro-average AUC calculated as the average of all group results (*Wei and Wang, 2020*). We found all per group AUCs vs. the rest to be >0.9, and micro-average and macro-average AUC >0.95 in both the TCGA (*Figure 7C*) and GTEx/GSE81540 test sets (*Figure 7D*). In addition to accurately classifying PAM50 subtypes, LEP-specific aging biomarkers distinguished normal from cancer tissue 100% and 93.3% of the time, respectively, in the TCGA and GTEx/GSE81540 test sets.

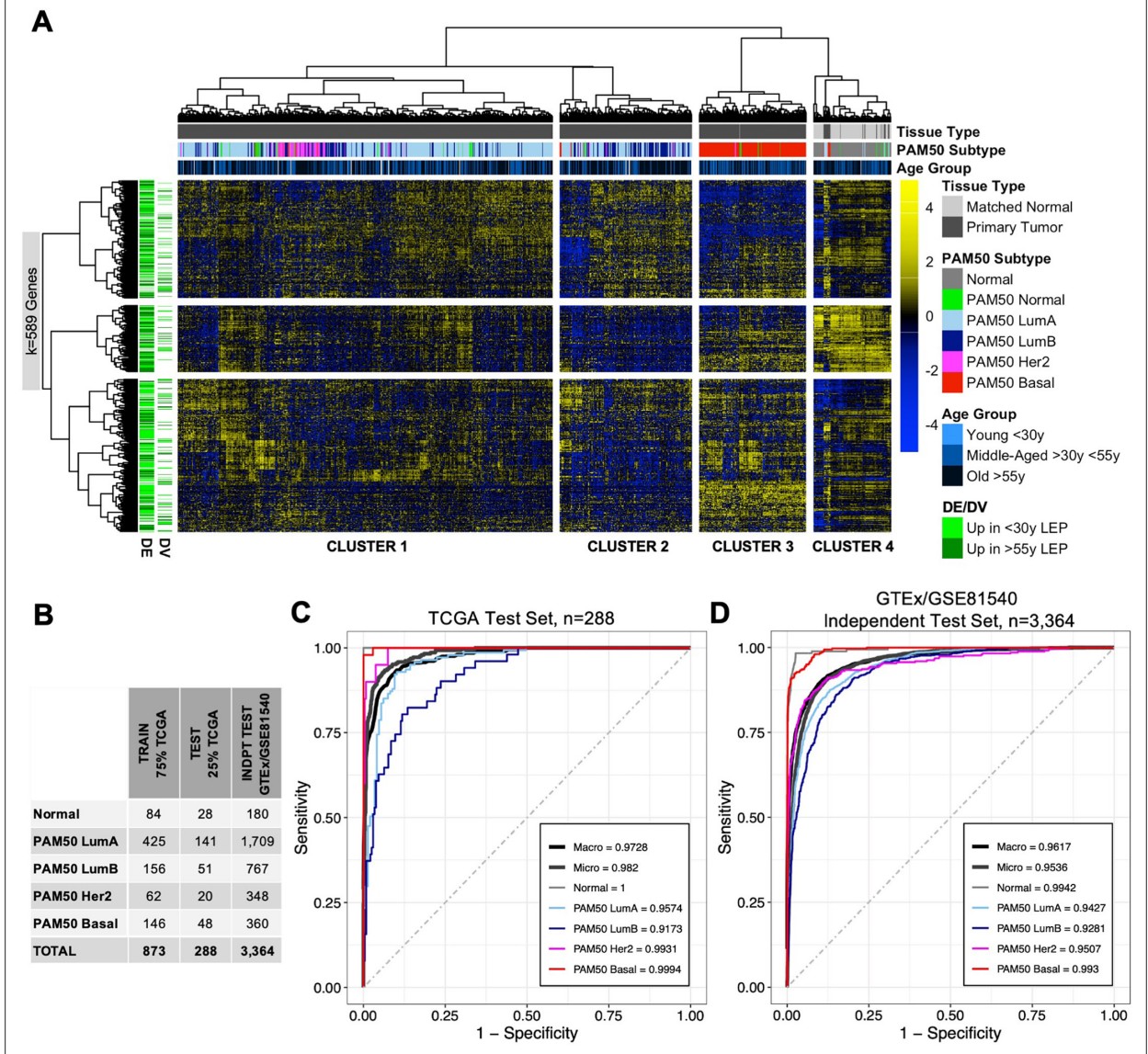

**Figure 7.** Age-dependent dysregulation in LEPs shape predictors of normal breast tissue and PAM50 subtypes. (**A**) Unsupervised hierarchical clustering of TCGA samples from matched normal and primary tumor tissue based on expression of age-dependent DE and DV genes identified in LEPs (adj. p<0.05). PAM50 intrinsic subtypes and patient age at diagnosis are annotated. Gene expression scaled log₂ FPKM values are represented in the heatmap; clustering performed using Euclidean distances and Ward agglomerative method. (Note: extreme outlier values are set to either the minimum or maximum value of the scale bar). Number of age-dependent DE and DV genes (k) included in analysis annotated. (**B**) Number of individuals in each ML class in the training, test and independent test sets. (**C–D**) Multi-class classification model performance in predicting normal tissue and breast cancer breast cancer subtypes in the (**C**) TCGA test set and (**D**) GTEx/GSE81540 independent test set. Macro AUC, micro AUC, and AUC of each group vs. rest are shown.

The online version of this article includes the following source data and figure supplement(s) for figure 7:

**Source data 1.** Age-dependent dysregulation in LEPs shape predictors of normal breast tissue and PAM50 subtypes.

**Figure supplement 1.** Breast cancer expression levels of aging biomarkers identified in normal luminal epithelia are associated with PAM50 subtypes in TCGA.

We next identified the genes that contributed most to the predictive ML model. We identified 127 genes with scaled variable importance contribution ≥25% in predicting at least one class (*Figure 8A*, *Figure 8—source data 1A*); 18% of predictors derived from DV analysis. Of these, estrogen receptor *ESR1* (downregulated in older LEPs) and transmembrane protein *TMEM45B* (upregulated in older LEPs) were part of the 50-gene PAM50 subtype predictors that had prognostic significance (*Parker*

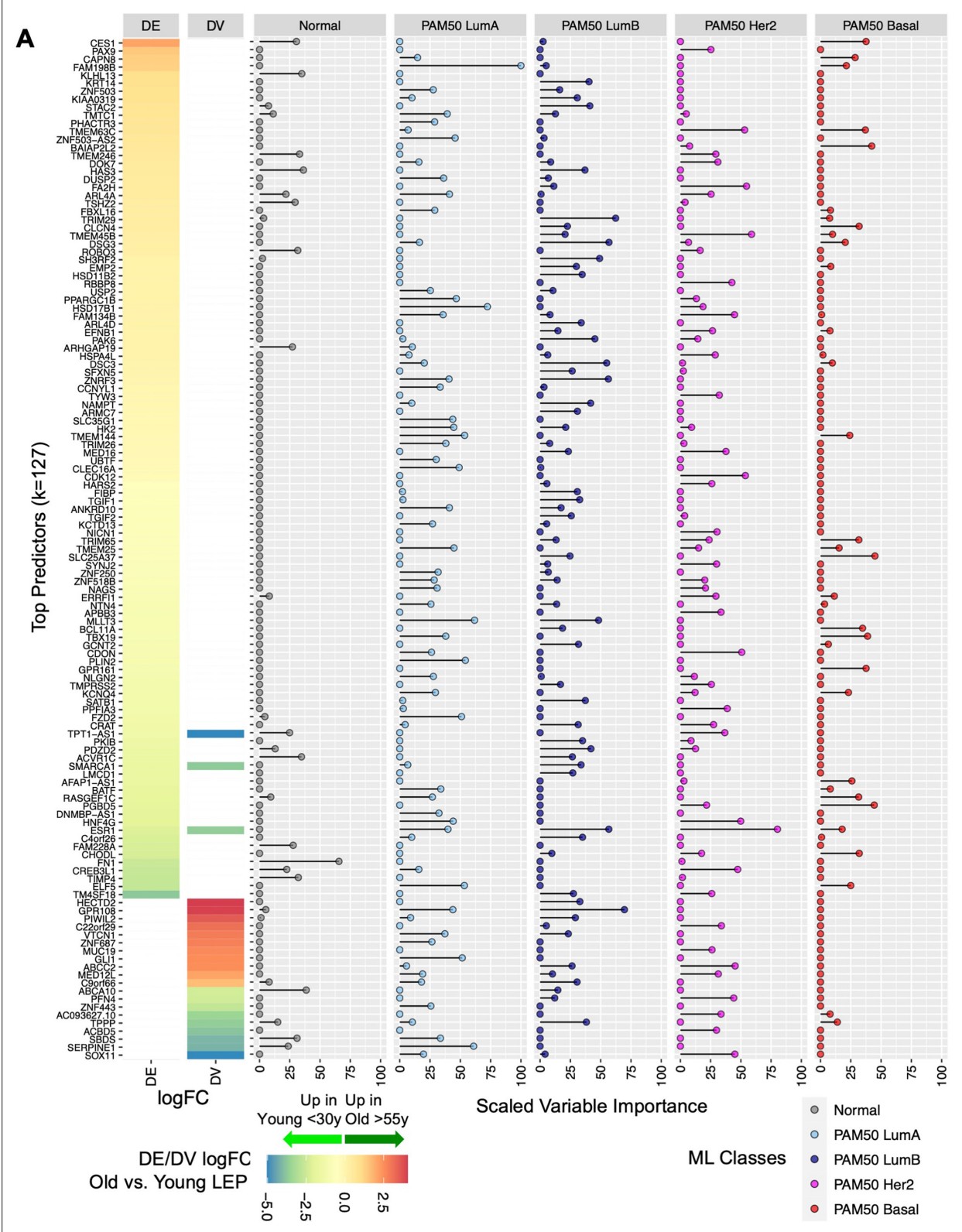

**Figure 8.** Relative contribution of top aging-associated biomarkers in predictive models of normal breast tissue and PAM50 subtypes in TCGA. (**A**) Gene predictors with scaled variable importance ≥25% in prediction of at least one class: normal breast tissue, PAM50 LumA, PAM50 LumB, PAM50 Her2 or PAM50 Basal, in TCGA. Rank ordered heatmap of DE and DV lfc in LEPs with (+) lfc higher in older and (-) lfc higher in younger LEPs (left); scaled variable importance of each gene in each TCGA class (right). Number of gene predictors (k) annotated.

*Figure 8 continued on next page*

*Figure 8 continued*

The online version of this article includes the following source data and figure supplement(s) for figure 8:

**Source data 1.** Relative contribution of top aging-associated biomarkers in predictive models of normal breast tissue and PAM50 subtypes in TCGA.

**Figure supplement 1.** Breast cancer expression levels of top aging-associated predictors of normal breast tissue and PAM50 subtypes in TCGA.

**Figure supplement 2.** Association of top aging-associated predictors of PAM50 subtypes with survival outcomes in TCGA.

*et al., 2009*). Of interest, our analysis of the predictive power of DE and DV genes at identifying PAM50 subtypes (*Figure 8A*) showed that DV genes were strong predictors (scaled variable contribution ≥25%) of PAM50 LumA, LumB, and Her2 subtypes and bore no strong predictive value for PAM50 Basal subtypes.

We highlight the top five genes that showed highest variable importance for each class: (i) Normal tissue—*FN1, ABCA10, HAS3, KLHL13,* and *ACVR1C*; (ii) PAM50 LumA—*FAM198B, HSD17B1, MLLT3, SERPINE1,* and *PLIN2*; (iii) PAM50 LumB—*GPR108, TRIM29, DSG3, ESR1,* and *ZNRF3*; (iv) PAM50 Her2—*ESR1, TMEM45B, FA2H, CDK12,* and *TMEM63C*; and (iv) PAM50 Basal—*SLC25A37, PGBD5, BAIAP2L2, TBX19,* and *GPR161* (*Figure 8—figure supplement 1A–E*). Top predictors of normal tissue and of PAM50 Her2 and PAM50 Basal subtypes showed larger differences in median expression relative to other groups. In contrast, PAM50 LumA and LumB top predictors showed large differences in median expression compared to non-luminal subtypes but exhibited relatively smaller, yet still significant, differences in median expression relative to each other (*Figure 8—figure supplement 1A–E*). Although expression of these predictive genes may not be specific to the LEP lineage alone, our findings suggest that age-dependent dysregulation of these genes in LEPs could disrupt lineage-specific signaling and the homeostatic control mechanisms of key biological processes that have been implicated in breast cancers.

Finally, using available TCGA survival data on PAM50 LumA, LumB, and Basal cancers (*Liu et al., 2018*), we assessed the contribution of the top predictors (scaled variable importance ≥25%) in each subtype to overall survival (OS) and progression-free interval (PFI) (*Hudis et al., 2007*; *Punt et al., 2007*). We used multivariate Cox proportional hazards regression models (*Harrell, 2015*) to test the simultaneous effect of all top predictors on survival time, adjusting for age at diagnosis and cancer stage (early vs. late). For PAM50 LumA cancers, we found 13% and 17% of top predictors ($k_{LumA\_top\_pred}=47$) were associated with OS and PFI, respectively; for PAM50 LumB cancers, 5% and 16% of top predictors ($k_{LumB\_top\_pred}=37$) were associated with OS and PFI; and for PAM50 Basal cancers, 0% and 3% of top predictors ($k_{LumB\_top\_pred}=14$) were associated with OS and PFI (Wald p<0.05;) (*Figure 8—figure supplement 2A–C*). These results suggest that subtype-specific expression of genes dysregulated with age in the LEP lineage also affect biology underlying survival outcomes specifically in patients with luminal type cancers.

Our results illustrate that age-dependent changes in LEPs embody biology that is relevant to and contributes to tissue-level biology predictive of breast cancer subtypes. These changes may reflect age-related dysregulation convergent with development of frank tumors. The degree and variability of these age-dependent changes across older individuals may explain the differential susceptibility between certain individuals to breast cancer initiation as they age and the development of specific breast cancer subtypes.

## Discussion

We have shown that aging phenotypes in the mammary gland result from the integration of directional and variant responses that reshape the transcriptomic landscape of the two main epithelial lineages of the breast, the LEPs and MEPs. Transcriptomic changes lead to a loss in lineage fidelity with age where faithfulness of lineage-specific expression is diminished. This is seen via the genome-wide loss of tuned windows of expression of lineage-specific genes and decrease in the magnitude of differential expression between the genes that define LEPs and MEPs. Our approach delineated the contribution of each epithelial lineage and identified two models mediating loss of lineage fidelity in breast epithelia with age—either via directional changes, as measured by differential gene expression, or via an increase in variance, as measured by differential variability analysis. Aging-dependent expression variances occurred in both lineages, whereas the overwhelming majority of directional

expression changes with age occurred in LEPs. This is a striking finding when one considers the two lineages arise from common progenitors, and that their juxtaposition in tissue allows for direct cell-cell communication.

We hypothesize that LEPs are the nexus of integration for variant responses that lead to stereotyped directional age-dependent outcomes in mammary gland. This is likely the result of a dynamic process of iterative feedback between LEPs and MEPs, as well as other cell types in the breast. LEPs from older women still maintained canonical LEP-specific features but exhibited genome-wide loss of lineage fidelity that implicated dysregulation of genes with known roles in breast cancer. This suggests that susceptibility to cancer entails loss of proper specification of the luminal lineage, and that age-dependent molecular changes in LEPs contributes to this loss. Age-dependent directional changes in LEPs were detectable in bulk tissue and implicated downregulation of chromatin and genome organizers, such as *SATB1*, suggesting means by which loss of lineage fidelity may be perpetuated genome-wide. The pathways affected by transcriptomic changes during aging were commonly linked with breast cancer, and the age-dependent changes in LEPs reflected relevant biology that distinguished normal tissue from breast cancers and predicted PAM50 breast cancer subtypes. Together our findings illustrate how age-dependent changes in LEPs contribute to the aging biology of breast tissue, and we propose that this biology reflects dysregulation convergent with processes associated with breast cancers.

Studies of gene expression during aging in human tissues have been largely restricted to analysis of bulk tissue lacking in cell-type-specific resolution and have been focused on directional changes with age using DE analysis. Bulk analyses make it impossible to separate effects of aging that are driven by the intrinsic changes that occur molecularly within each lineage vs. compositional changes that reflect shifts in cell type proportions. Lineage-specific analyses provide intermediate resolution between bulk RNA-seq and single-cell RNA-seq and allow for cost effective analysis of cell population-level responses and interactions. As such, lineage-resolution analyses also provide an avenue to validate computational deconvolution methods that have emerged to extract cell-type-specific contributions in bulk tissue (*Shen-Orr and Gaujoux, 2013*; *Titus et al., 2017*). We provided evidence that breast tissue-level changes with age are driven not just by changing compositions of the breast (*Benz, 2008*; *Garbe et al., 2012*), but by intrinsic molecular changes in the underlying cell populations. Indeed, while bulk tissue expression reflects cellular heterogeneity, we were able to identify age-dependent changes in bulk tissue that mirror the DE in LEPs with age, suggesting that LEPs contribute, if not drive, certain emergent properties of breast tissue.

These findings were further validated in our analysis of existing scRNA-seq studies of non-tumorigenic primary breast tissues (*Murrow et al., 2022*; *Nee et al., 2023*; *Pal et al., 2021*). We showed that our LEP-derived aging signature developed from lineage-specific RNA-sequencing captured aging biology of epithelial cell types at the single-cell level, with genes upregulated in older LEPs enriched in Luminal1 cells from the menopausal cohort and genes upregulated in younger LEPs enriched in Luminal1 cells from the premenopausal cohort (*Pal et al., 2021*). We further showed that our LEP-derived aging signature was enriched in women who had received chemotherapy and those with increased risk of developing breast cancer, lending further support to earlier findings of accelerated aging phenotypes in these groups (*Shalabi et al., 2021*; *Siddique et al., 2021*). Thus, the age-related decline in lineage fidelity within the luminal compartment was a histologically and molecularly observable phenomenon at the population level, which signified alterations in cell subpopulations. These changes appear to encompass shifts in subpopulation proportions and modifications in the gene expression of specific cell types and across defined cell states.

We found significant downregulation of the genome organizer *SATB1* in LEPs and breast tissue with age (adj. p<0.05), as well as concordant dysregulation of known SATB1-target genes (*Han et al., 2008*). Of note, *SATB1* was also downregulated in PAM50 LumA, LumB, and Her2 breast cancer subtypes relative to their matched normal tissue and had the lowest expression in luminal subtype cancers—the subtype most associated with aging. SATB1 has genome organizing functions in tumor progression (*Kohwi-Shigematsu et al., 2013*), and has been described as a key regulator of EMT in cancers (*Naik and Galande, 2019*). Although our results suggest that loss of SATB1-mediated genome organization with age is associated with the observed age-dependent dysregulation in the luminal lineage, SATB1's role in normal breast epithelia and in the context of aging remain to be fully elucidated. Moreover, we postulate that mechanisms of SATB1-mediated genome organization

in breast cancers are likely more complex than in aging normal epithelia due to the highly dysregulated state of cancers; this biology is also likely to be highly subtype-specific given the differential expression of *SATB1* across breast cancer subtypes. We also identified an additional 60 genes that showed concordant age-dependent directional changes in LEPs (adj. p<0.05) and in primary tissue (unadj. p<0.05) despite the difference in platforms (RNA-seq vs. microarray). Network and community analyses showed enrichment of genes involved in chromosomal rearrangement, including *BCL11A*—a subunit of the BAF (SWI/SNF) chromatin remodeling complex (*Kadoch et al., 2013*), *MLLT3*—a chromatin reader component of the SEC (*Moustakim et al., 2018*), *ZNF521*—a transcription co-factor with established roles in stem cells of diverse organs and tissues (*Scicchitano et al., 2019*), and *PHC1* and *PCGF3*—components of the PcG multiprotein PRC-1-like complex required for developmental maintenance of repressed but transcriptionally poised chromatin configuration through alteration of chromatin accessibility, folding, and global architecture of nuclear organization (*Illingworth, 2019*). *BCL11A*, *MLLT3*, *ZNF521*, *PHC1*, and *PCGF3* also showed subtype-specific dysregulation in breast cancers; *BCL11A*, *ZNF521*, *and PHC1* were specifically downregulated in luminal subtype cancers relative to their matched normal tissue. We speculate that chromatin and genome organization play a key role in maintaining the luminal lineage and that their dysregulation may mediate loss of lineage fidelity observed genome-wide with age.

The striking phenotypic changes in LEPs are starkly juxtaposed to MEPs, which have so far revealed few obvious signs of changes with age. Nevertheless, our experiments using heterochronous bilayers of MEPs and LEPs demonstrated that the chronological age of MEPs controls the biological age of LEPs, thus illustrating that MEPs do change with age and revealing the existence of a non-cell autonomous mechanism that integrates aging-imposed damage across the tissue (*Miyano et al., 2017*). Here, we showed that changes in MEPs largely involved changes in gene expression variances with age. Moreover, we showed that aging-associated increases in variances in both lineages drove a large fraction of the observed loss of lineage fidelity in epithelia of older women, comparable to the contribution of age-dependent directional changes. In our opinion, changes in variance are an underappreciated component of aging analyses. DE analysis is a ubiquitous statistical tool used in expression profiling studies, whereas only more recently have changes in variance been systematically analyzed (*Bashkeel et al., 2019*; *Brinkmeyer-Langford et al., 2016*; *de Jong et al., 2019*; *Sharma et al., 2018*; *Slieker et al., 2016*; *Viñuela et al., 2018*; *Xie et al., 2011*) and differential variance analytical tools developed (*Phipson et al., 2016*; *Phipson and Oshlack, 2014*; *Ran and Daye, 2017*).

For changes to be detected as significant in DE analysis, the following assumptions must be met: (i) the biological phenomenon causes dysregulation that is directional (e.g., genes are either upregulated or downregulated), and (ii) dysregulation occurs at the same time in the same genes, and in the same pattern across the majority of individuals in the group of interest (i.e., it is stereotypic). Although some aging processes may be deterministic, like telomere shortening, other processes may be stochastic, born from accumulated random physicochemical insults that manifest as an increase in noise in the system (*Todhunter et al., 2018*). In the latter case, the signal itself is the noise in the system. Another way to view this type of dysregulation is by observing the deviation from a set range. A change in the dynamic range of expression, for instance, of regulatory genes such as TFs that have very tuned or narrow windows of expression, can lead to dysregulation as expression deviates from the set range. This noise can lead to decoupling of tightly regulated networks.

Although we observed both increases and decreases in dynamic range with age, we specifically focused on increases in variance and their effect on loss of lineage fidelity with age, hypothesizing that genes that have very tightly tuned windows of expression in younger healthy individuals and that see large increases in variance in older individuals, are good candidates for susceptibility factors that could be predictive of breast cancer risk.

Moreover, in our view, this age-dependent differential variability between individuals is linked to the described increase in cellular heterogeneity in older individuals in single-cell studies, where aged cells were shown to have increased transcriptional variability and loss of transcriptional coordination compared to younger cells of the same tissue (*Enge et al., 2017*; *Kowalczyk et al., 2015*; *Levy et al., 2020*; *Martinez-Jimenez et al., 2017*). We had previously proposed that this within-sample transcriptional variability which leads to the identification of cell states and cell sub-populations in scRNAseq data, in turn, could lead to the observed shifts in transcriptomes at the cell population-level and bulk tissue-level in RNAseq data between individuals (*Todhunter et al., 2018*). As this gene

expression shift in the cell population-level and bulk tissue-level varies per individual and is dependent on the distribution of specific cell states and identified cell sub-populations in any specific individual, transcriptomic profiles of samples from some older individuals can deviate from the mean expression observed in the studied cohort. Thus, we suggest that the increase in cellular heterogeneity with age underlies the cell population-level increases in variances between the individuals. Indeed, our scRNA-seq analysis revealed how variations in cellular characteristics within specific cell lineages, as identified by defined cell states, were strongly linked to particular samples—underscoring the significance of the differential variability between individuals. This further supported the utility of DV analysis in RNA-seq studies that could identify genes for which increasing variances among individuals reflect the unique cell states and increasing cellular heterogeneity at the single-cell level. As such, the molecular signals of aging cells may not be fully captured as stereotyped directional changes. Instead, a large fraction of age-associated changes will be reflected as increases in measured variance in the molecular signal across an aged cohort.

We identified potential ligand-receptor pairs and cell-cell junction proteins, including tight junctions, desmosomes, and gap junction components, that potentially mediate dysregulated cell-cell and cell-microenvironment signaling within the epithelium. We provided experimental validation for the role of the gap junction protein GJB6 (Connexin-30) in mediating the ability of MEPs to impose an aging phenotype on LEPs (*Miyano et al., 2017*). It is unclear whether this occurs chemically through passage of ions or small molecules through gap junction channels, indirectly via gap junction-mediated structural proximity of LEPs and MEPs, or via signaling complexes with connexin-interacting proteins including cytoskeletal elements, tight and adherens junctions, and enzymes like kinases and phosphatases (*Dbouk et al., 2009*). How this occurs will require further exploration.

GSEA further identified age-dependent enrichment of gene sets in LEPs and MEPs that were commonly dysregulated in breast cancers, including gene sets related to inflammation and immunosenescence, processes synonymous with aging and cancer progression (*Fulop et al., 2017*); cell-cycle related targets of E2F transcription factors, which are thought to play a role in regulating cellular senescence (*Lanigan et al., 2011*); and targets of the oncogene *MYC*. Aging associated changes in the immune response were further implicated through our ligand-receptor pair analysis where we identified known immune-associated ligands and receptors that exhibited loss of lineage-specific expression in breast epithelia. We had previously shown that in situ innate and adaptive immune cell infiltration of the breast epithelia and interstitial stroma change with age consistent with a decline in immune surveillance and increased immunosuppression (*Zirbes et al., 2021*). How age-dependent changes in other cell types in the breast including stromal, vascular, and immune cell populations are linked to the dysregulation of epithelial signaling with age remains to be fully elucidated. Comprehensive transcriptomic profiling of FACS isolated primary immune, stromal, and vascular cell types from normal breast tissue had identified lineage-specific expression of ligands and receptors that could contribute to dynamic and reciprocal signaling between cell types in the breast interactome (*Del Toro et al., 2024*; *Thi et al., 2024*). Thus, reciprocal changes in these other cell types may contribute to the differential variability observed in epithelial cell types via cell-cell and cell-ECM interactions.

Although our age-specific analyses did not identify oncogenes that were DE between younger and older epithelia, gene set enrichment in LEPs and MEPs revealed a putative example of priming, a process in which certain pathway components undergo molecular changes that set the stage for potential dysregulation as aging progresses, a phenomenon we previously described in *Sayaman et al., 2021*. We did not detect changes in *MYC* expression with age, but MYC targets were among the gene sets that were significantly enriched in DE or DV analysis. MYC is amplified or overexpressed in ~35% of breast cancers and exerts pleiotropic effects across the genome (*Xu et al., 2010*). In the context of cancer progression, MYC can induce telomerase activity, which enables bypass of the replicative senescence barrier in mammary epithelial cells (*Garbe et al., 2014*). Although we do not have evidence for the direct involvement of MYC in this context, we speculate that secondary events such as demethylation at MYC binding sites at target genes could explain the enrichment of MYC relevant signatures. In another candidate example of priming, we observed significant downregulation of known EMT regulatory genes *GRHL2* and *SGSM2* in older LEPs (*Lin et al., 2019*; *Xiang et al., 2012*), but no loss of E-cadherin (*CDH1*) expression in LEPs with age. Given E-cadherin's pivotal role in maintaining cell-cell adhesion and cell polarity within epithelial structures, we did not expect dysregulation of E-cadherin in the context of aging in normal cells. As above, we speculate that further secondary

events downstream of *GRHL2* and *SGSM2* dysregulation could lead to the perturbations in E-cadherin expression necessary to drive EMT and oncogenic transformation.

That age-dependent changes are evident in epithelial lineages isolated from primary HMEC cultures suggest a degree of epigenetic stability that allow aged epithelial cells to occupy metastable states. Indeed, our examination of DNA methylation in matched LEP and MEP samples showed age-dependent differential methylation (DM) at promoter proximal regions of DE genes and negative correlation between DNA methylation and gene expression levels in these samples (*Miyano et al., 2017*; *Sayaman et al., 2021*). DM genes in older LEPs likewise showed enrichment for signaling pathways that have been shown to play a role in cancer progression (*Sayaman et al., 2021*). Moreover, LEPs of older women also exhibited loss of epigenetic suppression of retrotransposons that affect regulation of genes with oncogenic potential, specifically genes associated with luminal breast cancers (*Senapati et al., 2023*). Thus, age-dependent activation of gene expression networks in older breast epithelia that are stabilized through concomitant age-dependent changes in the epigenetic landscape could prime aged epithelia for oncogenic gene activation.

We proposed two distinct pathways through which we could explain how aging mechanisms contribute to aging-associated cancer susceptibility. (i) General dysregulatory mechanisms, which were reflected in directional age-dependent changes that share common features with known cancer mechanisms. These changes contribute to an increased overall susceptibility to cancer as individuals age. (ii) Individual-specific dysregulatory mechanisms, which were identified by variations in aged populations. These variations may help explain why certain individuals are more vulnerable to cancer initiation (i.e., not all aged individuals develop cancer), and why those who have cancer develop specific subtypes.

Using machine learning, we built a predictive elastic net model using the expression of age-dependent DE and DV genes identified in LEPs in breast tissue samples. This model could classify normal breast tissue from breast tumors and predict breast cancer subtypes in publicly available normal and cancer tissue transcriptomes from more >3000 women. Although association of mammary lineage-derived signatures with breast cancer subtype is well-established in the literature, our study notably demonstrates that changes in aging-associated genes specifically in the luminal lineage were robust predictors of breast cancer subtypes. These findings suggest that what is typically attributed to oncogenesis might actually be age-related changes within the luminal lineage. Moreover, based on our analysis of the relative contribution of the identified LEP-derived aging-associated DE and DV genes to the predictive model of breast cancer subtypes, we found DV genes to be strong predictors of PAM50 LumA, LumB and Her2 subtypes but held no strong predictive value for PAM50 Basal subtypes. Luminal subtypes are by far the most associated with aging, and we hypothesize that increased variances may underly the transcriptomic architecture of luminal cancer subtypes. Therefore, we speculate that inclusion of age-dependent DV genes derived from the luminal lineage could lead to better subtype predictors.

Collectively, our findings illustrate how tissue-level predictive biomarkers of breast cancer that have subtype-specific expression relative to matched normal samples are dysregulated with age at the cell population-level, specifically in the luminal lineage. The contribution of non-epithelial cell types to the age-dependent expression of these genes in bulk tissue remains the subject of future studies. Given that the mammary epithelium is the origin of breast carcinomas and age is the most significant risk factor for breast cancers, age-dependent changes in the transcriptomic landscape of luminal cells may be key contributing factors to the tissue-level dysregulation of cell-cell and cell-microenvironment signaling in breast cancers and may reflect relevant biology convergent with the development of frank tumors. Indeed, the variance in expression of these genes across aged individuals may reflect the differential susceptibility of certain individuals to specific breast cancer subtypes.

## Conclusions

Our studies show how directional and variant responses that contribute to aging biology are integrated in breast tissue of older women. We show that increased variances in the transcriptomic profiles of mammary epithelial lineages across individuals is a significant outcome of the aging process and is likely central to our understanding of the increased susceptibility to breast cancers with age. Strikingly, LEPs can integrate age-dependent signals from MEPs and almost exclusively exhibit the stereotyped, directional changes seen in aging epithelia that comprise a prominent signal detected in bulk tissue.

Age-dependent directional and variant changes in LEPs even shape the tissue-level expression of predictive biomarkers that classify normal tissue and breast cancer subtypes, illustrating how age-dependent dysregulation in LEPs may play a key role in tissue transformation into frank cancers. We also demonstrate how increased molecular noise during aging may lead to significant variances in the transcriptomes between aged individuals and propose that this mechanism could underlie differences in susceptibility to development of breast cancers, particularly luminal breast cancer subtypes. Because cancer susceptibility indicates a state that could be more easily pushed towards cancer initiation, we can consider the variances between aged individuals to occupy multiple metastable states, some of which represent susceptible phenotypes that can be perturbed towards development of specific cancers. We speculate that these variance-driven changes are examples of age-dependent priming events that push epithelial cells towards metastable states susceptible to malignant transformation. Therefore, the degree to which breast-cancer associated genes are variably expressed across the different cell populations of the breast and across different individuals may explain why breast cancers develop in only a subset of women in a subtype-specific manner.

## Methods

### Experimental model and subject details

The FACS-enriched luminal epithelial (LEP) and MEP myoepithelial (MEP) cells from finite lifespan, non-immortalized human mammary epithelial cells (HMECs) grown to 4th passage serve as our experimental model system. HMECs were derived from breast tissue organoids collected from reduction mammoplasties (RM). Deidentified surgical discard tissue was obtained with consent for research and publication under a Lawrence Berkeley National Laboratory (Berkeley, CA) approved IRB 22997, or from City of Hope under IRB 17185 for sample distribution and collection. Protocols have already been established by our lab for the propagation and maintenance of these cells in vitro which allow for highly reproducible source material (*Garbe et al., 2009*; *Labarge et al., 2013*).

We examined genome-wide transcription in 54 primary LEP and MEP samples from 19 women across a range of ages (*Figure 1—source data 1*). These epithelial lineages were isolated from HMECs from two age cohorts: younger <30 y women considered to be premenopausal (age range 16-29y, $m_{LEP}$ = 16, $m_{MEP}$ = 16 samples, n=11 subjects) and older >55 y women considered to be postmenopausal (age range 56-72y, $m_{LEP}$ = 11, $m_{MEP}$ = 11 samples, n=8 subjects). Transcriptomic profiles of 4th passage cells were compared to LEPs and MEPs isolated from uncultured organoids (<30 y, $n_{LEP}$ = 4, $n_{MEP}$ = 3, and >55 y, $n_{LEP}$ = 3, $n_{MEP}$ = 1); only transcripts with concordant lineage-specific expression in cultured and uncultured cells were retained for analyses.

### Materials design analysis

#### Group allocation

Samples were allocated based on age demographics of women donating tissue. As menopausal status was not available for all subjects, we used age cutoffs to identify two cohorts in this aging study. The two age groups were defined as younger <30 y women considered to be premenopausal, and older >55 y women considered to be postmenopausal; samples from middle-aged >30 y and <55 y women were excluded from analysis. Only finite lifespan cell strains derived from RM from healthy women were included; strains from prophylactic mastectomies (women with high-risk breast cancer mutations or family history) or normal-adjacent to tumor tissue were excluded. No group blinding or masking was used.

#### Replicates

For the purposes of this study using finite-lifespan cell strains derived from primary tissue, subject-level biological replicates (n) refer to data derived from tissue from different subjects. Sample-level biological replicates (m) are data derived from the same subject but from separate cell culture/co-culture, FACS isolation, sample and library preparation, and RNA-sequencing experiments. Sample replicates are typically bridge samples used across different experiment batches. Technical replicates (l) are data derived from a single subject from a single sample pre-processing experiment (e.g., qPCR). When sample replication for the same subject can be modeled, analyses are done at sample level; if not, analyses are done at subject level by taking the mean value of sample replicates (see Method Details

and Quantification and Statistical Analysis section). For clarity, subject (n), sample (m), and technical (l) replicates for each analysis are annotated in each figure. The datasets generated and analyzed during the current study include RNA-sequencing count data publicly available as part of GSE182338 (*Miyano et al., 2021*; *Sayaman et al., 2021*; *Sayaman et al., 2022*; *Shalabi et al., 2021*; *Todhunter et al., 2021*). Criteria for inclusion of samples and gene transcripts included in the analyses, and all exclusion criteria are described in the Method Details section.

## Sample-size estimation

Sample size for RNA-seq analyses was restricted to available organoids in the established HMEC bank that were isolated from reduction mammoplasties of healthy women who fall under the age range of interest: younger <30 y and older >55 y, that could be expanded in primary culture. Power analyses for the *limma::voom* DE pipeline were conducted post hoc using the R package *ssizeRNA::check. power* function (*Bi and Liu, 2019*). Power calculation for lineage-specific DE between LEP and MEP with DE genes defined at BH adj. p<0.001 and fold-change ≥2 yielded (i) an average power (ave.pw)= 0.91 and FDR average (fdr.ave)=0.00062 in younger <30 y (n=11 subjects); and (ii) ave.pw=0.90 and fdr.ave=0.00047 in older >55 y (n=8). Power calculation for age-dependent DE between young <30 y and old >55 y with DE genes defined at BH adj. p<0.05 yielded a range of (iii) ave.pw=0.69–0.76 and fdr.ave=0.040–0.045 in LEPs (for n=8 and n=11, respectively); (iv) ave.pw=0.87–0.89 and fdr.ave=0. 083–0.047 in MEPs (for n=8 and n=11, respectively).

## Statistical reporting

Statistical analysis methods are described in full in the Quantification and Statistical Analysis section. Exact p-values are shown in figures when feasible, otherwise significance levels are annotated. Exact p-values and summary statistics at defined significance level thresholds are provided in source data; full summary statistics are available upon request.

# Method details

## Breast tissue collection and HMEC culture

Pre-stasis (primary) HMECs were initiated and maintained according to previously reported protocols using M87A medium containing cholera toxin and oxytocin at 0.5 ng/ml and 0.1 nM, respectively (*Garbe et al., 2009*; *Labarge et al., 2013*). For experiments, 4th passage HMECs were cultured to sub-confluence prior to FACS-sorting. HMEC strains used in this study for RNA-seq are provided (*Figure 1—source data 1A*).

## Flow cytometry

FACS-enriched LEPs and MEPs were isolated from 4th passage finite-lifespan HMEC from reduction mammoplasties from two age cohorts: younger <30 y women considered to be premenopausal (age range 16-29y) and older >55 y women considered to be postmenopausal (age range 56-72y). LEP and MEP enrichment was performed across multiple studies (*Miyano et al., 2021*; *Sayaman et al., 2021*; *Sayaman et al., 2022*; *Shalabi et al., 2021*; *Todhunter et al., 2021*). Enrichment was conducted by FACS using well-established LEP-specific (CD227, MUC1 or CD133, PROM1) and MEP-specific (CD271, NGFR or CD10, MME) cell-surface markers. Protocols were validated to sort similar populations regardless of antibody combination. Briefly, breast epithelial cells were stained and sorted following standard flow cytometry protocol. Primary HMEC strains for RNA-seq were stained with anti-human CD227-FITC (BD Biosciences, clone HMPV) or anti-human CD133-PE (BioLegend, clone7), and anti-human CD271-APC (BioLegend, clone ME20.4). Primary organoids were stained with anti-human CD133-PE (BioLegend, clone7) and anti-human CD271-APC (BioLegend, clone ME20.4).

## Cell co-cultures

In co-culture study (*Miyano et al., 2017*), FACS-enriched MEPs from 4th passage HMEC were re-plated on six-well plates and cultured until the cells were confluent. The cells were treated with Mitomycin C (Santa Cruz Biotechnology, sc-3514) at 10 µg/ml for 2.5 hr.

In the co-culture study with shGJB6, MEP cell strains used in KD experiments were specifically selected to be the samples from older women with increased expression of *GJB6* relative to MEP from

younger women – these samples had ~two-fold increase expression of *GJB6*. FACS-enriched control and shGJB6 transduced MEPs from older >55 y women were plated on six-well plates and cultured until the cells were confluent. FACS-enriched 4th passage LEPs from younger <30 y women were seeded directly on the mitomycin C-treated or shRNA transduced MEP layer. LEPs from co-cultures were separated after 10 days for gene expression qPCR analysis by FACS using anti-human CD133-PE (BioLegend, clone7) and anti-human CD271-APC (BioLegend, clone ME20.4). For Gap junction inhibition assay, cells were cultured with indicated concentration of 18-alpha-Glycryrhetinic acid (Sigma, G8503) for 7 days; LEPs from co-culture were then separated using FACS with anti-CD227-FITC (BD Biosciences, 559774, clone HMPV) and anti-CD10-PE (BioLegend, 312204, clone HI10a).

## RNA isolation and qPCR

Total RNAs were isolated from enriched LEPs and MEPs with Quick-RNA Microprep Kit (Zymo Research, R1050). For RNA-seq, isolated RNAs were submitted to Integrative Genomic Core at City of Hope (IGC at COH) for library preparation and sequencing. For qPCR, cDNAs were synthesized with iScript Reverse Transcription Supermix (Bio-Rad, 1708840) according to the manufacturer's manual. Quantitative gene expression analysis was performed by CFX384 real-time PCR (Bio-Rad) with iTaq Universal SYBR Green Supermix (Bio-Rad, 1725125). Data were normalized to RPS18 or TBP by relative standard curve method.

Forward and reverse primer sequences generated in this study are indicated below:

GJB6 forward and reverse primers:
5'-CTACAGGCACGAAACCACTCG-3', 5'ACCCCTCTATCCGAACCTTCT-3'
ELF5 forward and reverse primers:
5'-TAGGGAACAAGGAATTTTTCGGG-3', 5'-GTACACTAACCTTCGGTCAACC-3'
TBP forward and reverse primers:
5'-GAGCTGTGATGTGAAGTTTCC-3', 5'-TCTGGGTTTGATCATTCTGTAG-3'
RPS18 forward and reverse primers:
5'-GGGCGGCGGAAAATAG-3', 5'-CGCCCTCTTGGTGAGGT-3'

Sequences for shGJB6 and shCtrl were ggatacttgctccattcatac and gcttcgcgccgtagtctta, respectively. shCtrl (CSHCTR001LVRU6GP) and shGJB6 Lenti-virus vector (HSH06069132LVRU6GP) were purchased from GeneCopoeia.

## RNA-sequencing

Transcriptomic profiling of LEPs and MEPs from two age cohorts: younger <30 y (m=32 LEP and MEP samples, n=11 subjects) and older >55 y (m=22, n=8) women (*Figure 1—source data 1A*) was performed via RNA-sequencing as part of the LaBarge sequencing collection GSE182338 (*Miyano et al., 2021*; *Sayaman et al., 2021*; *Sayaman et al., 2022*; *Shalabi et al., 2021*; *Todhunter et al., 2021*). Briefly, RNA sequencing libraries were prepared with Kapa RNA mRNA HyperPrep Kit (Kapa Biosystems, Cat KR1352) or KAPA stranded mRNA-seq (Kapa Biosystems, Cat KK8420) according to the manufacturer's protocol using 100 ng of total RNA from each sample for polyA RNA enrichment. Sequencing was performed on Illumina HiSeq 2500 with single read mode, and real-time analysis was used to process the image analysis. RNA-sequencing reads were trimmed using *Trimmomatic* (*Bolger et al., 2014*), and processed reads were mapped back to the human genome (hg19) using *TOPHAT2* software (*Kim et al., 2013*). *HTSeq* (*Anders and Huber, 2010*) and *RSeQC* (*Wang et al., 2012*) were applied to generate the count matrices.

RNA-sequencing data pre-processing was conducted in *DESeq2* (*Love et al., 2014*) and *edgeR* (*Robinson et al., 2010*) on the entirety of the LaBarge sequencing collection GSE182338 (m=120 LEP and MEP samples, n=48 subjects) as described in *Miyano et al., 2021*; *Sayaman et al., 2021*; *Sayaman et al., 2022*; *Shalabi et al., 2021*; *Todhunter et al., 2021* including samples not included in this study. RNA-seq transcript Ensembl IDs were mapped to corresponding gene symbols, Entrez IDs and UniProt IDs using *EnsDb.Hsapiens.v86* (*v2.99.0*) database (*Rainer, 2017*). We restricted analysis to the 17,328 genes with comparable dynamic ranges and consistent lineage-specific expression between primary organoid and 4th passage LEPs and MEPs in both age cohorts (linear regression $R^2 \geq 0.88–0.91$, p<0.0001) (*Figure 1—figure supplement 1A–D*). ComBat batch-adjusted regularized

log (rlog) expression values (*Johnson et al., 2007*; *Leek et al., 2020*; *Love et al., 2014*) were used for visualization and downstream analysis.

### Breast tissue public transcriptomic data sets

For differential expression analysis in bulk normal primary breast tissue, GSE102088 (*Song et al., 2017*) microarray data (n=114) were downloaded from the Gene Expression Omnibus (GEO) database using the *GEOquery* (*Davis and Meltzer, 2007*). For machine learning, three data sets were used: (1) TCGA RNA-seq FPKM data from matched normal or PAM50 Normal, Luminal A (LumA), Luminal B (LumB), Her2 and Basal subtype breast cancer tissues (n=1201) were downloaded using *TCGAbiolinks* (*Colaprico et al., 2016*) package; (2) GTEx RNA-seq count data from female subjects (n=180) were downloaded using the recount3 (*Collado-Torres et al., 2017*; *Wilks et al., 2021*) and FPKM transformed; and (3) GSE81540 (*Brueffer et al., 2020*; *Brueffer et al., 2018*; *Dahlgren et al., 2021*) RNA-seq FPKM data from PAM50 Normal, LumA, LumB, Her2 and Basal subtype breast cancer tissues (n=3184) were downloaded from GEO.

### TCGA survival data

Curated TCGA survival data was downloaded from *Liu et al., 2018*. Survival data were restricted to breast cancers and female subjects. Overall survival (OS) and progression-free interval (PFI) (*Hudis et al., 2007*; *Punt et al., 2007*) time and event data were used. Early and late cancer stage was defined from the annotated AJCC pathologic tumor stage: early stage included Stages I, IA, IB, II, IIA, and IIB; and late stage included Stages III, IIIA, IIIB, IIIC, and IV. Patients with missing age at pathologic diagnosis and cancer stage were excluded. Survival data for n=1072 patients were merged with scaled patient-level expression data (mean value of sample replicates) from primary tumors.

### scRNA-seq public data sets

For scRNA-seq analyses, three data sets were used: (1) GSE161529 (*Pal et al., 2021*) scRNA-seq data from 13 non-tumorigenic breast tissue samples (19-69y) from reduction mammoplasties were kindly provided as a Seurat object by the group of Dr. Andrea Bild; (2) GSE174588 (*Nee et al., 2023*) scRNA-seq data from 11 non-tumorigenic noncarrier (24-50y) and 11 *BRCA1$^{+/mut}$* (21-54y) breast tissue samples from reduction mammoplasties, prophylactic mastectomies, and contralateral to DCIS/tumor were kindly provided as a Seurat object by Dr. Kai Kessenbrock's group with annotated cell states; (3) GSE198732 (*Murrow et al., 2022*) scRNA-seq data from 28 healthy reduction mammoplasty tissue samples (19-39y) were downloaded from NCBI GEO as a Seurat object.

## Quantification and statistical analysis

### Differential analyses

For differential analyses of LEP and MEP samples, a combination of lineage and age group was modeled. Differential expression (DE) was performed in *limma voom* (*Law et al., 2014*; *Ritchie et al., 2015*) on sample-level data from 17,328 genes with eBayes moderation and RNA-seq batch modeled as a covariate, and with adjustment for biological replicates. Differential variability (DV) was performed in *MDSeq* (*Ran and Daye, 2017*) on batch-adjusted subject-level data from 14,601 genes whose variances could be estimated after outlier removal. For lineage-specific DE analyses, contrasts between LEP and MEP in younger <30 y and in older >55 y women were performed. Lineage-specific DE thresholds were set at Benjamini-Hochberg (BH) adjusted p<0.001 and absolute log$_2$ fold changes, abs(lfc) ≥1 in each age cohort. LEP-specific and MEP-specific genes were defined as those with lineage-specific DE in younger <30 y women. For age-dependent analyses, contrasts between <30 y and >55 y LEPs and <30 y and >55 y MEPs were performed, and age-dependent directional or variant changes were defined at DE or DV BH adj. p<0.05 in each lineage.

Age-dependent DE analysis of normal primary breast tissue was performed on publicly available GSE102088 microarray data (n=114 subjects,<30 y n=35,>30y<55 y n=68,>55 y n=11) (*Song et al., 2017*) in *limma* with eBayes moderation. Significant DE between age groups in bulk tissue were defined at BH adj. p<0.05 and nominal significance at unadj. p<0.05. *scRNA-seq analysis:* scRNA-seq analyses were carried out in *Seurat* (*Hao et al., 2021*). All datasets were preprocessed were preprocessed to select for identified epithelial cell types and subjected to dimensionality reduction via Uniform Manifold Approximation and Projection (UMAP). Genes that were not

expressed in at least 10 cells were excluded from downstream analysis. Epithelial cells were subset from each dataset, log normalized by a scale factor of 10,000, variable genes were identified with *FindVariableFeatures* using VST as a selection method, PCA was performed using variable genes, and UMAPs for each dataset were generated using the first five dimensions. Epithelial subsets were renamed to be consistent across datasets with Luminal1 cells representing mature luminal cells and progenitors, Luminal2 cells containing hormone sensing luminal cells, and Basal cells being comprised of basal and myoepithelial populations. *scCustomize* (*Marsh, 2023*) package was used to visualize the expression of single lineage markers on UMAPs from each dataset. Single-cell Gene Set Enrichment Analysis (scGSEA) was performed using the *escape* (*Borcherding and Andrews, 2021*) package that utilizes *UCell* (*Andreatta and Carmona, 2021*) to execute and visualize GSEA across individual cells. Comparisons between cell types and groups were conducted using standardized mean differences with significance levels calculated by subtracting the means of two groups and dividing by the pooled standard deviation not accounting for sample size of each group (*Andrade, 2020*).

## Gene Set Enrichment Analysis (GSEA)

Fast gene set enrichment analysis (*fgsea*) (*Korotkevich et al., 2021*) was used to identify age-dependent enrichment of Molecular Signatures Database (MSigDB) hallmark gene sets (*Liberzon et al., 2015*) in LEPs or MEPs using DE and DV rank-ordered test statistics. Enriched gene sets were defined as those with enrichment BH adj. p<0.05. For bulk tissue GSEA analysis (GSE102088,<30 y n=35,>55 y n=11), gene sets were constructed from age-dependent genes in LEPs: (i) 251 genes that were differentially upregulated in young <30 y LEPs; and (ii) 220 genes that were that were differentially upregulated in old >55 y LEPs. Age-dependent enrichment was assessed in bulk tissue using DE rank-ordered test statistics; enrichment was similarly defined at BH adj. p<0.05.

## Single-sample Gene Set Enrichment Analysis (GSEA)

Gene set enrichment scores for individual samples were computed single-sample GSEA (ssGSEA) (*Subramanian et al., 2005*; *Barbie et al., 2009*) implemented in the GSVA R package (*Hänzelmann et al., 2013*).

## Lineage-specific ligand-receptor pair interactions and functional network analysis

Ligand-receptor pairs (LRPs) (*Ramilowski et al., 2015*) gene symbols were mapped to Ensembl IDs using *EnsDb.Hsapiens.v86* database (*Rainer, 2017*). Lineage-specific LRPs were defined based on either the LEP-specific or MEP-specific (DE adj. p<0.001 and fold-change ≥2) expression of either the ligand and/or its cognate receptor in the younger cohort. Lineage-specific LRP interactions were considered to be lost in the older cohort when lineage-specific DE of the ligand and/or its cognate receptor was lost in the older cells (not passing the DE at adj. p<0.001, abs(lfc) ≥1 threshold). Functional network enrichment of LRPs in the younger cohort and LRPs lost in the older cohort were performed using the Search Tool for the Retrieval of Interacting Genes/Proteins (STRING) database (https://string-db.org/) and enriched KEGG pathways (false discovery rate, FDR p<0.05) were reported.

## Age-dependent DE Protein-protein Interactions and Functional Network Analysis

Protein-protein interaction (PPI) analysis was performed using the STRING database (https://string-db.org/). All possible PPI are considered using all active interaction sources and setting minimum require interaction score to the lowest confidence threshold of 0.150. Network was visualized in *igraph* (*Csárdi and Nepusz, 2006*), and only the largest fully connected main network of genes was plotted. Community detection was performed on this main network using optimal community structure algorithm in terms of maximal modularity score in *igraph* (*Brandes et al., 2008*). Each community was then analyzed in STRING for functional network enrichment (FDR *P*<0.05) and common functional terms were summarized and reported.

## Non-parametric Kruskal-Wallis and Wilcoxon Test

*Limma*-based genome-wide DE analysis was not performed on publicly available gene expression datasets from breast cancer tissue. Instead, analysis was limited to genes of interest, and either non-parametric Wilcoxon test (unpaired for independent samples and paired for non-independent samples) or Kruskal-Wallis (KW) test (*rstatix v.0.5.0::kruskal_test*) (**Kassambara, 2020b**), a one-way ANOVA on ranks, was used to determine differences in $\log_2$(FPKM +1) values between two or more groups, respectively. Wilcoxon and KW p-values were corrected for multiple testing (BH) across all features examined; in the TCGA cohort, KW test was performed across PAM50 breast cancer subtypes as well as matched normal tissue; p-values were adjusted across all age-dependent DE and DV genes identified in LEPs. For multiple groups, post hoc pairwise comparison between groups was performed using Wilcoxon test (*rstatix v.0.5.0::wilcox_test*) (**Kassambara, 2020b**). Post hoc Wilcoxon p-values were corrected for multiple testing (BH) across all pairwise comparisons and across all features examined; in the TCGA cohort, Wilcoxon p-values were adjusted across pairwise subtype comparisons and across all DE and DV genes. Paired Wilcoxon tests were also perfored to compare primary tumors to their matched normal tissue dependent of patient ID and p-values were adjusted across subtypes. Wilcoxon and KW BH-adj. p-values were likewise annotated (*ggpubr v.0.2.5*) (**Kassambara, 2020a**) at different significance levels p<0.05 (*),<0.01 (**),<0.001 (***),<0.0001 (****). Non-parametric Wilcoxon and KW tests were similarly used to compare ssGSEA signature scores between two or more groups respectively.

## Lepage test on location and scale

Two-sample Lepage test (*nonpar v.0.1–2*) (**Pepler, 2017**) is a joint non-parametric test of equality for location (central tendency) and scale (variability). Lepage test was performed on the subject-level *ComBat* batch-adjusted normalized rlog expression values of genes encoding for junction proteins between younger and older cells in each lineage. Significant age-dependent modulation of genes for cell-surface junction proteins in LEPs and MEPs were defined at p<0.05.

## Kolmogorov-Smirnov test on lineage-specific DE lfc

Non-parametric two-sample Kolmogorov-Smirnov (KS) test (*stats::ks.test*) on the equality of distributions performed to compare distributions of lineage-specific DE lfc in younger and in older cells. Significance defined at p<0.05.

## T-test on the differences of DE lfc between age groups

One-sided t-test (*stats::t.test*) performed on the distribution of pairwise differences in lineage-specific DE lfc between age groups (lfc in young - lfc in old) to test if the mean of all values is different from 0. Significance defined at p<0.05, nominal significance defined at p≤0.1.

## T-test on qPCR values between experimental treatments

Two-sided Student's t-test performed to compare normalized expression between the two groups in co-culture experiments. Significance defined at p<0.05.

## Fisher's exact test

Median expression levels of age-dependent DE and DV genes from LEPs were assessed in the TCGA cohort across PAM50 LumA, LumB, Her2, and Basal intrinsic subtypes. Contingency tables reflecting number of genes by PAM50 subtype and by highest and lowest median expression levels for: DE genes upregulated in young LEP; DE genes upregulated in old LEP; DV genes with higher variance in young LEP; and DV genes with higher variance old LEP were tabulated. Fisher's exact test were performed in R (*stats::fisher.test*) on each of the contigency tables and p-values were adjusted using the Bonferroni method.

## Unsupervised hierarchical clustering

Unsupervised hierarchical clustering was implemented using Ward's clustering criterion (ward.D2) agglomerative method with Euclidean distances as distance metric. Hierarchical clustering of gene correlation matrices were implemented using complete agglomerative method with 1 – Pearson

correlations as as distance metric. Heatmaps were generated using *gplots* (*v.3.0.3::heatmap.2*) *and pheatmap* (*v. 1.0.12*) (*Kolde, 2019*; *Warnes et al., 2020*) packages. Dendrograms were plotted using the *dendextend* (*v1.13.4*) (*Galili, 2015*) package. Approximately unbiased (AU) *p*-values and bootstrap probability (BP) were calculated and annotated using *pvclust* (*v.2.2–0*) (*Suzuki and Shimodaira, 2006*) package which assesses uncertainty in hierarchical clustering analysis. Clusters with AU p≥0.95 were highlighted.

## Machine learning

Machine learning (ML) multi-class prediction of normal breast tissue and PAM50 breast cancer subtypes was performed in *caret* (*v.6.0–88*) R package (*Kuhn, 2024*) using an elastic net model from *glmnet* (*v.4.1–2*) (*Friedman et al., 2022*) based on tissue expression of 536 mapped age-dependent DE and DV genes identified in LEPs. ML was carried out in three large publicly available RNA-seq datasets of normal and cancer breast tissue: GTEx, TCGA and GSE81540 (*Brueffer et al., 2020*; *Brueffer et al., 2018*; *Dahlgren et al., 2021*) with analysis restricted to tissues from women annotated as normal or PAM50 LumA, LumB, Her2, and Basal subtypes. The ML model was trained using 10-fold cross-validation with 3 repeats in 75% of TCGA data (n=873) using a hybrid subsampling technique via the SMOTE algorithm in the *DMwR* (*v.0.4.1*) package (*Torgo, 2010*), and optimizing for mean balanced accuracy. Model performance was then evaluated in the 25% of TCGA (n=288) and an independent dataset of normal tissues from GTEx and breast cancer tissues from GSE81540 (n=3364). ML multi-class prediction performance was evaluated in each test set using the *MultiROC* (*v.1.1.1*) R package (*Wei and Wang, 2020*): (i) macro-average area under the ROC curve (AUC), calculated as the average of all group results; (ii) micro-average AUC, calculated by stacking all groups together; and (iii) AUC of each group vs. the rest. Gene predictors were identified as genes with scaled variable importance contribution to the predictive model. Genes with scaled variable importance ≥25% in prediction of at least one class were visualized; gene expression the top 5 predictors in each class were further analyzed in the TCGA breast cancer cohort.

## Cox proportional hazards regression analysis

Multivariate Cox proportional hazards regression models (*Harrell, 2015*), implemented in the *survival* (*v.3.5–3::coxph*) R package (*Therneau, 2023*; *Therneau and Grambsch, 2000*), were used to simultaneously assess the effect of all top ML predictors – defined as genes with subtype scaled variable importance ≥25% – in each subtype on OS and PFI, with age at diagnosis and cancer stage (early vs. last) as additional covariates. Analyses were performed in the PAM50 LumA, LumB, and Basal subtypes; PAM50 Her2 subtype was underpowered, and models did not converge and was excluded from the report. Forest plots of hazard ratios with 95% confidence interval and Wald p-values annotated were plotted using the *forestmodel* (*v.0.6.4*) R package (*Kennedy and Wang, 2022*). The fraction of top predictors with significant contribution (p<0.05) to OS and PFI were reported.

## Availability of data and materials

Pre- stasis human mammary epithelial cell (HMECs) strains included in this study are available upon request and will be provided as they are available. Forward and reverse primer sequences generated in this study are provided in the Methods section. The datasets generated and analyzed during the current study include RNA-sequencing count data publicly available as part of GSE182338 (*Miyano et al., 2021*; *Sayaman et al., 2021*; *Sayaman et al., 2022*; *Shalabi et al., 2021*; *Todhunter et al., 2021*). The gene expression data that support the findings of this study are available from GSE102088 *Song et al., 2017*; GSE81540 *Brueffer et al., 2020*; *Brueffer et al., 2018*; *Dahlgren et al., 2021*; The Cancer Genome Atlas (TCGA) Research Network: https://www.cancer.gov/tcga; and The Genotype-Tissue Expression (GTEx) Project: https://gtexportal.org/. Single-cell RNAseq data sets used for validation are available from GSE161529 *Pal et al., 2021*; GSE174588 *Nee et al., 2023*; and GSE198732 (*Murrow et al., 2022*). Analysis was conducted using standard R/Bioconductor packages and statistical tests implemented in R. All package versions, model design, and parameters are described in detail in Methods. Summary statistics at defined significance levels are provided in source data; full summary statistics are provided via FigShare (https://figshare.com/s/2a7ceffccfe3f35f3ce8).

## Acknowledgements

We would like to dedicate this manuscript to the memory of Susan Samson, a dedicated patient advocate who was with us from the beginning of this work. We would like to acknowledge the contributions of Dr. James Garbe to the HMEC bank; our current and previous research associates Jennifer Lopez, Jessica Bloom and Jonathan Lee for their technical support; our patient advocate Sandy Preto; and the City of Hope Bioinformatics Core led by Dr. Xiwei Wu, specifically Dr. Min-Hsuan Chen who generated the RNA-sequencing raw count data, and Dr. Jinhui Wang who prepared the RNA-sequencing library and performed the sequencing on the Illumina HiSeq2500 platform. Dr. Kai Kessenbrock, Dr. Andrea Bild, and Dr. Lyndsay Murrow and Dr. Zev Gartner for their assistance accessing scRNA-seq data from breast tissues. Dr. Keely Walker, Editing Manager at the City of Hope Office of Faculty & Institutional Support, for their assistance in copy editing the revised manuscript. The results shown and referenced here are based in part upon data generated by the TCGA Research Network: https://www.cancer.gov/tcga; and The Genotype-Tissue Expression (GTEx) Project: https://gtexportal.org/ supported by the Common Fund of the Office of the Director of the National Institutes of Health (NIH), and by NCI, NHGRI, NHLBI, NIDA, NIMH, and NINDS. The investigators are grateful for support from: the NIH/NCI Cancer Metabolism Training Program Postdoctoral Fellowship T32CA221709 and the NIH/NCI Mentored Research Scientist Development Award to Promote Diversity K01CA279498 additionally supported by the NIH Office of the Director and Office of Data Science Strategy (RWS); the United States Department of Energy under contract no. DE-AC02-05CH11231 (MRS); and from the NIH U01CA244109, R33AG059206, R01EB024989, R01CA237602; the Department of Defense/ Army Breast Cancer Era of Hope Scholar Award BC141351 and Expansion Award BC181737, Conrad N Hilton Foundation, Yvonne Craig-Aldrich Fund for Cancer Research, and City of Hope Center for Cancer and Aging (MAL). Research reported in this publication included work performed in the Integrative Genomics and Bioinformatics, and Analytical Cytometry Cores supported by the National Cancer Institute of the National Institutes of Health under grant number P30CA033572. The content is solely the responsibility of the authors and does not necessarily represent the official views of the National Institutes of Health. The funders had no role in study design, data collection and analysis, decision to publish, or preparation of the manuscript.

## Additional information

### Funding

| Funder | Grant reference number | Author |
|---|---|---|
| National Cancer Institute | T32CA221709 | Rosalyn W Sayaman |
| National Cancer Institute | K01CA279498 | Rosalyn W Sayaman |
| National Cancer Institute | U01CA244109 | Mark A LaBarge |
| National Institute on Aging | R33AG059206 | Mark A LaBarge |
| National Institute of Biomedical Imaging and Bioengineering | R01EB024989 | Mark A LaBarge |
| National Cancer Institute | R01CA237602 | Mark A LaBarge |
| Department of Defense | BC141351 | Mark A LaBarge |
| Department of Defense | BC181737 | Mark A LaBarge |
| Conrad N. Hilton Foundation | | Mark A LaBarge |
| Yvonne Craig-Aldrich Fund for Cancer Research | | Mark A LaBarge |
| City of Hope Center for Cancer and Aging | | Mark A LaBarge |
| United States Department of Energy | DE-AC02-05CH11231 | Martha R Stampfer |

| Funder | Grant reference number | Author |
|--------|------------------------|--------|

The funders had no role in study design, data collection and interpretation, or the decision to submit the work for publication.

## Author contributions

Rosalyn W Sayaman, Conceptualization, Data curation, Software, Formal analysis, Funding acquisition, Validation, Investigation, Visualization, Methodology, Writing – original draft, Writing – review and editing; Masaru Miyano, Conceptualization, Data curation, Formal analysis, Validation, Investigation, Visualization, Methodology, Writing – original draft, Writing – review and editing; Eric G Carlson, Software, Formal analysis, Validation, Investigation, Visualization, Methodology, Writing – original draft, Writing – review and editing; Parijat Senapati, Validation, Investigation, Writing – review and editing; Arrianna Zirbes, Investigation, Writing – review and editing; Sundus F Shalabi, Michael E Todhunter, Data curation, Investigation, Writing – review and editing; Victoria E Seewaldt, Susan L Neuhausen, Dustin E Schones, Writing – review and editing; Martha R Stampfer, Resources, Data curation, Investigation, Writing – review and editing; Mark A LaBarge, Conceptualization, Resources, Supervision, Funding acquisition, Writing – original draft, Writing – review and editing

## Author ORCIDs

Rosalyn W Sayaman ⓘ https://orcid.org/0000-0003-1343-0619
Masaru Miyano ⓘ https://orcid.org/0000-0002-1490-4743
Eric G Carlson ⓘ https://orcid.org/0000-0002-2784-1729
Arrianna Zirbes ⓘ https://orcid.org/0000-0002-3849-2616
Sundus F Shalabi ⓘ http://orcid.org/0000-0002-8440-2474
Martha R Stampfer ⓘ https://orcid.org/0000-0002-3801-5086
Dustin E Schones ⓘ https://orcid.org/0000-0001-7692-8583
Mark A LaBarge ⓘ https://orcid.org/0000-0003-2405-4719

## Ethics

The FACS-enriched luminal epithelial (LEP) and MEP myoepithelial (MEP) cells from finite lifespan, non-immortalized human mammary epithelial cells (HMECs) grown to 4th passage serve as our experimental model system. HMECs were cultured from breast tissue organoids collected from reduction mammoplasties (RM). Deidentified surgical discarded tissue was obtained non-prospectively with consent for research and publication under a Lawrence Berkeley National Laboratory (Berkeley, CA) approved IRB 22997, or from City of Hope (Duarte, CA) under IRB 17185 for sample distribution and collection.

## Decision letter and Author response

Decision letter https://doi.org/10.7554/eLife.95720.sa1
Author response https://doi.org/10.7554/eLife.95720.sa2

# Additional files

## Supplementary files

MDAR checklist

## Data availability

The datasets generated and analyzed during the current study include RNA-sequencing count data publicly available as part of GSE182338 (*Miyano et al., 2021*; *Sayaman et al., 2021*; *Sayaman et al., 2022*; *Shalabi et al., 2021*; *Todhunter et al., 2021*). The gene expression data that support the findings of this study are available from GSE102088 (*Song et al., 2017*); GSE81540 (*Brueffer et al., 2020*; *Brueffer et al., 2018*; *Dahlgren et al., 2021*); The Cancer Genome Atlas (TCGA) Research Network: https://www.cancer.gov/tcga; and The Genotype-Tissue Expression (GTEx) Project: https://gtexportal.org/. Single-cell RNAseq data sets used for validation are available from GSE161529 (*Pal et al., 2021*); GSE174588 (*Nee et al., 2023*); and GSE198732 (*Murrow et al., 2022*). Analysis was conducted using standard R/Bioconductor packages and statistical tests implemented in R. All package versions, model design, and parameters are described in detail in Methods. Summary statistics at defined significance

levels are provided in source data; full summary statistics are provided via FigShare (https://figshare.com/s/2a7ceffccfe3f35f3ce8).

The following datasets were generated:

| Author(s) | Year | Dataset title | Dataset URL | Database and Identifier |
|---|---|---|---|---|
| Sayaman RW, Miyano M, Shalabi S, Todhunter ME, Stampfer MR, LaBarge MA | 2021 | Genome-wide loss of lineage fidelity is a hallmark of aging breast epithelia and reflects a biology convergent with susceptibility to cancer initiation | https://www.ncbi.nlm.nih.gov/geo/query/acc.cgi?acc=GSE182338 | NCBI Gene Expression Omnibus, GSE182338 |
| Sayaman RW, Miyano M | 2025 | Sayaman, Miyano, et al., eLife 2024 | https://doi.org/10.6084/m9.figshare.21311178 | figshare, 10.6084/m9.figshare.21311178 |

The following previously published datasets were used:

| Author(s) | Year | Dataset title | Dataset URL | Database and Identifier |
|---|---|---|---|---|
| Nee K, Nguyen Q, Kessenbrock K | 2021 | Preoplastic stromal cells promote BRCA1-mediated breast tumorigenesis | https://www.ncbi.nlm.nih.gov/geo/query/acc.cgi?acc=GSE174588 | NCBI Gene Expression Omnibus, GSE174588 |
| Gartner ZJ, Murrow LM | 2022 | Mapping hormone-regulated cell-cell interaction networks in the human breast at single-cell resolution | https://www.ncbi.nlm.nih.gov/geo/query/acc.cgi?acc=GSE198732 | NCBI Gene Expression Omnibus, GSE198732 |
| Song MA | 2017 | Expression data from normal breast tissues | https://www.ncbi.nlm.nih.gov/geo/query/acc.cgi?acc=GSE102088 | NCBI Gene Expression Omnibus, GSE102088 |
| Saal LH | 2018 | Clinical Value of RNA Sequencing–Based Classifiers for Prediction of the Five Conventional Breast Cancer Biomarkers: A Report From the Population-Based Multicenter Sweden Cancerome Analysis Network—Breast Initiative [superseries] | https://www.ncbi.nlm.nih.gov/geo/query/acc.cgi?acc=GSE81540 | NCBI Gene Expression Omnibus, GSE81540 |
| Smyth GK, Chen Y, Pal B, Visvader JE | 2021 | scRNA-seq profiling of breast cancer tumors, BRCA1 mutant pre-neoplastic mammary gland cells and normal mammary gland cells | https://www.ncbi.nlm.nih.gov/geo/query/acc.cgi?acc=GSE161529 | NCBI Gene Expression Omnibus, GSE161529 |

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

## Appendix 1

### Supplementary Methods
#### Resource availability
##### Lead contact
Further information and requests for resources and reagents should be directed to and will be fulfilled by the Lead Contact, Mark A. LaBarge (mlabarge@coh.org).

##### Materials availability
Human mammary epithelial cells (HMECs) derived from subjects included in this study are available upon request.

Forward and reverse primer sequences generated in this study are indicated below:

GJB6 forward and reverse primers:
5'-CTACAGGCACGAAACCACTCG-3', 5'ACCCCTCTATCCGAACCTTCT-3'
ELF5 forward and reverse primers:
5'-TAGGGAACAAGGAATTTTTCGGG-3', 5'-GTACACTAACCTTCGGTCAACC-3'
TBP forward and reverse primers:
5'-GAGCTGTGATGTGAAGTTTCC-3', 5'-TCTGGGTTTGATCATTCTGTAG-3'
RPS18 forward and reverse primers:
5'-GGGCGGCGGAAAATAG-3', 5'-CGCCCTCTTGGTGAGGT-3'

Sequences for shGJB6 and shCtrl were ggatacttgctccattcatac and gcttcgcgccgtagtctta, respectively. shCtrl (CSHCTR001LVRU6GP) and shGJB6 Lenti-virus vector (HSH06069132LVRU6GP) were purchased from GeneCopoeia.

### Experimental model and subject details
The FACS-enriched luminal epithelial (LEP) and MEP myoepithelial (MEP) cells from finite lifespan, non-immortalized human mammary epithelial cells (HMECs) grown to 4th passage serve as our experimental model system. HMECs were derived from normal breast tissue organoids collected from reduction mammoplasties (RM) prepared at Lawrence Berkeley National Laboratory (Berkeley, CA) with approved IRB for sample distribution and collection from specific locations. Protocols have already been established by our lab for the propagation and maintenance of these cells in vitro which allow for highly reproducible source material (*Garbe et al., 2009*; *Labarge et al., 2013*).

We examined genome-wide transcription in 54 primary LEP and MEP samples from 19 women across a range of ages (*Figure 1—source data 1*). These epithelial lineages were isolated from HMECs from two age cohorts: younger <30 y women considered to be premenopausal (age range 16-29y, $m_{LEP}$ = 16, $m_{MEP}$ = 16 samples, n=11 subjects) and older >55 y women considered to be postmenopausal (age range 56-72y, $m_{LEP}$ = 11, $m_{MEP}$ = 11 samples, n=8 subjects). Transcriptomic profiles of 4th passage cells were compared to LEPs and MEPs isolated from uncultured organoids (<30 y, $n_{LEP}$ = 4, $n_{MEP}$ = 3, and >55 y, $n_{LEP}$ = 3, $n_{MEP}$ = 1); only transcripts with concordant lineage-specific expression in cultured and uncultured cells were retained for analyses.

### Materials design analysis
#### Group allocation
Samples were allocated based on age demographics of women donating tissue. As menopausal status was not available for all subjects, we used age cutoffs to identify two cohorts in this aging study. The two age groups were defined as younger <30 y women considered to be premenopausal, and older >55 y women considered to be postmenopausal; samples from middle-aged >30 y and <55 y women were excluded from analysis. Only finite lifespan cell strains derived from RM from healthy women were included; strains from prophylactic mastectomies (women with high-risk breast cancer mutations or family history) or normal-adjacent to tumor tissue were excluded. No group blinding or masking was used.

#### Replicates
For the purposes of this study using finite-lifespan cell strains derived from primary tissue, subject-level biological replicates (n) refer to data derived from tissue from different subjects. Sample-level

biological replicates (m) are data derived from the same subject but from separate cell culture/co-culture, FACS isolation, sample and library preparation, and RNA-sequencing experiments. Sample replicates are typically bridge samples used across different experiment batches. Technical replicates (l) are data derived from a single subject from a single sample pre-processing experiment (e.g., qPCR). When sample replication for the same subject can be modeled, analyses are done at sample level; if not, analyses are done at subject level by taking the mean value of sample replicates (see Method Details and Quantification and Statistical Analysis section). For clarity, subject (n), sample (m), and technical (l) replicates for each analysis are annotated in each figure. The datasets generated and analyzed during the current study include RNA-sequencing count data publicly available as part of GSE182338 (*Miyano et al., 2021*; *Sayaman et al., 2021*; *Sayaman et al., 2022*; *Shalabi et al., 2021*; *Todhunter et al., 2021*). Criteria for inclusion of samples and gene transcripts included in the analyses, and all exclusion criteria are described in the Method Details section.

## Sample-size estimation

Sample size for RNA-seq analyses was restricted to available organoids in the established HMEC bank that were isolated from reduction mammoplasties of healthy women who fall under the age range of interest: younger <30 y and older >55 y, that could be expanded in primary culture. Power analyses for the limma::voom DE pipeline were conducted post hoc using the R package ssizeRNA (*v.1.3.2*)::check.power function which calculates average power (ave.pw) and true average FDR (ave.fdr) for a given sample size when FDR is controlled for using the BH step-up procedure (*Bi and Liu, 2019*). These values are calculated for four DE analyses groups: lineage-specific DE between LEP and MEP in (i) young <30 y group and (ii) old >55 y; and age-dependent DE between young <30 y and old >55 y (iii) LEPs and (iv) MEPs. Input values include: total number of genes (nGenes)=17,328; calculated means of sample-level count data in control group, with LEPs serving as control group in lineage-specific analysis and young <30 y as control group in age-dependent analysis; and dispersion measures calculated from sample-level normalized values in *edgeR* (*v.3.22.5*) (*Robinson et al., 2010*) using the estimateCommonDisp function. Power calculation is run using 10 simulations with replacement with the following DE analysis-specific values: FDR rate of 0.001 for lineage-specific and 0.05 for age-dependent analysis; proportion of non-differentially expressed genes nGenes – DE nGenes/nGenes where DE nGenes is the number of genes at defined thresholds of BH adj. p<0.001 and fold-change ≥2 for lineage-specific DE genes and BH adj. p<0.05 for age-dependent DE genes; proportion of up-regulated genes among all DE genes; fold-change values of DE genes; and sample size of treatment group where we conservatively use subject-level (n) replication size.

Power calculation for lineage-specific DE between LEP and MEP with DE genes defined at BH adj. p<0.001 and fold-change ≥2 yielded (i) an average power (ave.pw)=0.91 and FDR average (fdr.ave)=0.00062 in younger <30 y (n=11 subjects); and (ii) ave.pw=0.90 and fdr.ave=0.00047 in older >55 y (n=8). Power calculation for age-dependent DE between young <30 y and old >55 y with DE genes defined at BH adj. p<0.05 yielded a range of (iii) ave.pw=0.69–0.76 and fdr.ave=0.040–0.045 in LEPs (for n=8 and n=11, respectively); (iv) ave.pw=0.87–0.89 and fdr.ave=0.083–0.047 in MEPs (for n=8 and n=11, respectively).

## Statistical reporting

Statistical analysis methods are described in full in the Quantification and Statistical Analysis section. Exact p-values are shown in figures when feasible, otherwise significance levels are annotated. Exact p-values and summary statistics at defined significance level thresholds are provided in source data; full summary statistics are available upon request.

## Method details

All computational analyses were conducted using R (3.5.0) (*R Development Core Team, 2018*; https://www.R-project.org/) and Bioconductor (3.7) (*Huber et al., 2015*; https://www.bioconductor.org/) unless otherwise noted.

*Breast tissue collection and HMEC culture:* Primary HMECs were established and maintained according to previously reported protocol using M87A medium containing cholera toxin and oxytocin at 0.5 ng/ml and 0.1 nM, respectively (*Garbe et al., 2009*; *Labarge et al., 2013*). For experiments, 4th passage HMECs were cultured for 4–6 days (depending on strain) to sub-confluence prior to FACS-sorting. Cell cultures were fed every 2 days up to and including the day before FACS-sorting. HMEC strains used in this study were listed in *Figure 1—source data 1* for RNA-seq.

## Disassociation of uncultured cells from organoids

For dissociation of uncultured cells from organoids, organoids were digested with 0.5% trypsin/EDTA for 10 min at 37 °C with agitation. After trypsin treatment, organoids were disrupted by vigorous shaking for 30 s. Cells were then passed through a 40 µm cell strainer (BD Falcon).

## Flow cytometry

LEP and MEP enrichment was performed across multiple studies (*Miyano et al., 2021*; *Sayaman et al., 2021*; *Sayaman et al., 2022*; *Shalabi et al., 2021*; *Todhunter et al., 2021*). Enrichment was conducted by FACS (BD FACSVantage SE, FACSAriaIII, FACS AriaSORP or Bio-Rad S3 Cell Sorter) using well-established LEP-specific (CD227, MUC1 or CD133, PROM1) and MEP-specific (CD271, NGFR or CD10, MME) cell-surface markers. Protocols were validated to sort similar populations regardless of antibody combination and enrichment methodology. Briefly, breast epithelial cells were stained and sorted following standard flow cytometry protocol. Primary HMEC strains for RNA-seq were stained with anti-human CD227-FITC (BD Biosciences, clone HMPV) or anti-human CD133-PE (BioLegend, clone7), and anti-human CD271-APC (BioLegend, clone ME20.4). Primary organoids were stained with anti-human CD133-PE (BioLegend, clone7) and anti-human CD271-APC (BioLegend, clone ME20.4).

## Cell co-cultures

In co-culture study (*Miyano et al., 2017*), FACS-enriched MEPs from 4th passage HMEC were re-plated on six-well plates and cultured until the cells were confluent. The cells were treated with Mitomycin C (Santa Cruz Biotechnology, sc-3514) at 10 µg/ml for 2.5 hr.

In the co-culture study with shGJB6, MEP cell strains used in KD experiments were specifically selected to be the samples from older women with increased expression of *GJB6* relative to MEP from younger women – these samples had ~two-fold increase expression of *GJB6*. FACS-enriched control shGJB6 transduced MEPs from older >55 y women were plated on six-well plates and cultured until the cells were confluent. FACS-enriched 4th passage LEPs from younger <30 y women were seeded directly on the mitomycin C-treated or shRNA transduced MEP layer. LEPs from co-cultures were separated by FACS after 10 days for gene expression qPCR analysis. Co-cultured LEPs were stained with anti-human CD133-PE (BioLegend, clone7) and anti-human CD271-APC (BioLegend, clone ME20.4) and were separated using BD FACS ARIAIII. For Gap junction inhibition assay, cells were cultured with indicated concentration of 18-alpha-Glycyrrhetinic acid (Sigma, G8503) for 7 days. LEP from co-culture was separated using BD FACSVantage with anti-CD227-FITC (BD Biosciences, 559774, clone HMPV) and anti-CD10-PE (BioLegend, 312204, clone HI10a).

## RNA isolation and qPCR

Total RNAs were isolated from enriched LEPs and MEPs with Quick-RNA Microprep Kit (Zymo Research, R1050). For RNA-seq, isolated RNAs were submitted to Integrative Genomic Core at City of Hope (IGC at COH) for library preparation and sequencing. For qPCR, cDNAs were synthesized with iScript Reverse Transcription Supermix (Bio-Rad, 1708840) according to the manufacturer's manual. Quantitative gene expression analysis was performed by CFX384 real-time PCR (Bio-Rad) with iTaq Universal SYBR Green Supermix (Bio-Rad, 1725125). Data were normalized to RPS18 or TBP by relative standard curve method.

## RNA-seq sequencing library preparation and sequencing with Illumina Hiseq2500

RNA sequencing libraries were prepared with Kapa RNA mRNA HyperPrep Kit (Kapa Biosystems, Cat KR1352) or KAPA stranded mRNA-seq (Kapa Biosystems, Cat KK8420) according to the manufacturer's protocol. Briefly, 100 ng of total RNA from each sample was used for polyA RNA enrichment. The enriched mRNA underwent fragmentation and first strand cDNA synthesis. The combined 2nd cDNA synthesis with dUTP and A-tailing reaction generated the resulting ds cDNA with dAMP to the 3' ends. The barcoded adaptors were ligated to the ds cDNA fragments. A 10-cycle of PCR was performed to produce the final sequencing library. The libraries were validated with the Agilent Bioanalyzer DNA High Sensitivity Kit and quantified with Qubit. Sequencing was performed on Illumina HiSeq 2500 with the single read mode of 51cycle. Real-time analysis (*RTA v2.2.38*) software was used to process the image analysis.

## Sequence alignment and gene counts

RNA-Seq reads were trimmed to remove sequencing adapters using *Trimmomatic* (***Bolger et al., 2014***). The processed reads were mapped back to the human genome (hg19) using *TOPHAT2* software (***Kim et al., 2013***). *HTSeq* (***Anders and Huber, 2010***) and *RSeQC* (***Wang et al., 2012***) software were applied to generate the count matrices and strand information, respectively with default parameters.

## RNA-seq pre-processing

RNA-sequencing data pre-processing was conducted on the entirety of the LaBarge sequencing collection GSE182338 (***Miyano et al., 2021***; ***Sayaman et al., 2021***; ***Sayaman et al., 2022***; ***Shalabi et al., 2021***; ***Todhunter et al., 2021***) across organoids and 4th passage HMECs (m=120 LEP and MEP samples, n=48 subjects) including samples not included in this study. The experimental design group was defined by the combination of the culture condition (organoid, 4th passage), cell type (LEP, MEP) and age/risk status (normal risk RM younger <30 y, normal risk RM older >55 y, and PM/ CLTT/PTT without or with germline mutation) of the samples. Raw counts for 34,623 transcripts from RNA-sequencing of FACS-sorted LEPs and MEPs were normalized and regularized log (rlog) transformed in *DESeq2* (*v1.20.0*) (***Love et al., 2014***) after removal of 30,196 transcripts with zero values across all samples. For initial QA, transformation was run blind to the design matrix (blind = TRUE). Sample QA revealed one outlier sample (subject ID 160, MEP from organoid) that was removed. Rlog transformation was repeated after outlier sample removal without blinding the transformation to the design matrix (~RNA seq Batch +Design Group) (blind = FALSE). Rlog values were batch-adjusted using *ComBat* (*sva v.3.35.2*) (***Johnson et al., 2007***; ***Leek et al., 2020***) function with the experimental design group as covariate in the model matrix (~Design Group). *ComBat* batch-adjusted rlog values were used for quality control and assessment of transcripts and filtered as discussed below.

## RNA-seq filtering

During quality control assessment, 5 entries (no feature, ambiguous, too low quality, not aligned, aligned not unique) were filtered out. Count data were imported into *edgeR* (*v.3.22.5*) (***Robinson et al., 2010***) as a *DGEList* object. Transcripts with low counts were determined using the *edgeR::filterbyExpr* function with experimental design group and batch as covariates in the design matrix (~0 + Design Group + RNA seq Batch); 14,647 transcripts with low counts were removed. Next, genes with highly discordant expression levels between organoids and 4th passage LEPs and MEPs grown in 2D culture were assessed. Mean subject-level batch-adjusted rlog values were calculated for each lineage and age group. Linear regression was performed on the gene expression value means from cells isolated from organoid vs. 4th passage culture in each of the RM LEP < 30 y, LEP > 55 y, MEP <30 y, MEP >55 y subsets, and transcripts with absolute value of model residuals ≥ 6 (~4*sd) in either of the four subsets were considered outliers; 423 transcripts were flagged for exclusion in the DGEList object. Finally, we considered genes that did not maintain consistent lineage-specific expression between organoids and 4th passage cells grown in 2D culture. Normalization factors were calculated for the filtered data using *edgeR::calcNormFactors* function using TMM method. DE analysis in *limma voom* (*v.3.36.5*) (***Law et al., 2014***; ***Ritchie et al., 2015***) was performed between LEPs and MEPs from younger <30 women in both organoids and 4th passage cells (see Quantification and Statistical Analysis). We used a less stringent filter cut-off to fully capture even nominal lineage-specific expression and genes with lineage-specific DE significance of adj. p<0.1 between LEPs and MEPs in younger <30 women in either organoid and 4th passage 2D culture were identified. Of this set, 2220 genes showed culture-dependent effect and did not maintain consistent lineage-specific expression between organoids and 4th passage cells in 2D culture; these genes were subsequently excluded. A final set of 17,328 genes were used for all downstream analyses of 4th passage and organoid data.

## Batch-adjustment of RNA-seq data

RNA-seq count data were pooled from five experiments and three studies conducted at different times (***Miyano et al., 2021***; ***Sayaman et al., 2021***; ***Sayaman et al., 2022***; ***Shalabi et al., 2021***; ***Todhunter et al., 2021***). Each experiment was defined to be an RNA-seq batch. Batch effects were found during QC in hierarchical clustering (*hclust*) and principal component analysis (PCA) (*prcomp*) of LEP and MEP samples based on normalized rlog expression values.

For visualization and down-stream analysis of normalized expression data, rlog values were corrected for batch effects using *ComBat* – an empirical Bayes approach for adjusting data for batch effects that is robust to outliers in small sample sizes (*Johnson et al., 2007*). The experimental design group was defined by the combination of culture condition (organoid, 4th passage), cell type (LEP, MEP) and age group/risk group (average risk RM younger <30 y, average risk RM older >55 y, or higher risk PM/CLTT/PTT with or without germline mutation) and was included in the model matrix (~Design Group). *ComBat* batch-adjustment was applied in the *sva* (*v3.35.2*) package (*Leek et al., 2020*). *ComBat* batch-adjusted data were then filtered to remove the set of genes described above. Filtered, batch-adjusted data were checked with PCA, and linear regression analysis to confirm removal of PC association with RNA-seq batch. Hierarchical clustering was also used pre- and post-*ComBat* treatment for visualization of batch effects and the clustering of bridge samples.

Filtered *ComBat* batch-adjusted rlog values were subsetted for samples of interest and used for visualization (*ggplot v.2_3.3.3, gplots v.3.0.3::heatmap.2*; *Warnes et al., 2020*; *Wickham, 2016*) and downstream analysis of expression values. Subject-level data were calculated as the mean value of the batch-adjusted rlog values for subjects with multiple samples.

For differential expression (DE) analysis of RNA-seq count data in *limma voom* (*v3.36.5*) (*Law et al., 2014*; *Ritchie et al., 2015*), RNA-seq batch was included in the linear model along with the above design group (~0 + Design Group +Batch). For differential variability (DV) analysis of RNA-seq count data in *MDSeq* (*v.1.0.5*) (*Ran and Daye, 2017*), since we were not able to model multiple samples from the same subject, *ComBatSeq* batch-adjustment (*sva_devel*) (*Zhang et al., 2020*) of the count data was first performed using the above design group in the model matrix. *ComBatSeq* batch-adjusted count data were then normalized using TMM method (*MDSeq::normalize.counts*). Subject-level data were generated as the mean value of *ComBatSeq* batch-adjusted normalized data for subjects with multiple samples and used in the DV analysis.

## Annotation of RNA-seq data

RNA-seq transcript Ensembl IDs were mapped to corresponding gene symbols, Entrez IDs and UniProt IDs using *EnsDb.Hsapiens.v86* (*v2.99.0*) database (*Rainer, 2017*).

## RNA-seq data used in this manuscript

Transcriptomes in LEPs and MEPs from reduction mammoplasty HMECs at $4^{th}$ passage (n=11 < 30 y, n=8 > 55 y) and uncultured organoids (n=4 < 30 y, n=3 > 55 y LEPs; n=3 < 30 y, n=1 > 55 y MEPs) were extracted from annotated pre-processed RNA-seq counts and rlog values from GSE182338 (*Miyano et al., 2021*; *Sayaman et al., 2021*; *Sayaman et al., 2022*; *Shalabi et al., 2021*; *Todhunter et al., 2021*) and used during QC analysis to determine exclusion of genes that did not maintain concordant lineage-specific expression between organoid and $4^{th}$ passage (see *RNA-seq Filtering*). Gene expression data isolated $4^{th}$ passage LEPs and MEPs from two age cohorts: younger <30 y women considered to be premenopausal (age range 16-29y, $m_{LEP}$ = 16, $m_{MEP}$ = 16 samples, n=11 subjects) and older >55 y women considered to be postmenopausal (age range 56-72y, $m_{LEP}$ = 11, $m_{MEP}$ = 11 samples, n=8 subjects) were used for downstream analysis.

## Breast tissue public transcriptomic data sets

For differential expression analysis in bulk normal primary breast tissue, GSE102088 (*Song et al., 2017*) microarray data (n=114) were downloaded from the Gene Expression Omnibus (GEO) database using the *GEOquery* (*Davis and Meltzer, 2007*). For machine learning, three data sets were used: (1) TCGA RNA-seq FPKM data from matched normal or PAM50 Normal, Luminal A (LumA), Luminal B (LumB), Her2 and Basal subtype breast cancer tissues (n=1201) were downloaded using TCGAbiolinks (*Colaprico et al., 2016*) package; (2) GTEx RNA-seq count data from female subjects (n=180) were downloaded using the recount3 (*Collado-Torres et al., 2017*; *Wilks et al., 2021*) and FPKM transformed; and (3) GSE81540 (*Brueffer et al., 2020*; *Brueffer et al., 2018*; *Dahlgren et al., 2021*) RNA-seq FPKM data from PAM50 Normal, LumA, LumB, Her2 and Basal subtype breast cancer tissues (n=3184) were downloaded from GEO.

## GSE102088 data

Normalized microarray $\log_2$ expression data, pheno data and feature data from normal primary breast tissues from n=114 women (GSE102088) (*Song et al., 2017*) were downloaded from the Gene

Expression Omnibus (GEO) database (https://www.ncbi.nlm.nih.gov/geo/) using the *GEOquery* (*v.2.48.0*) package (*Davis and Meltzer, 2007*). The experimental design group was defined by age groups (younger <30 y, middle aged >30 y<55 y, and older >55 y) for differential expression analysis and visual*ization.*

## The cancer genome atlas data

Normalized and pre-processed TCGA RNA-seq FPKM expression values and clinical data from TCGA were downloaded using *TCGAbiolinks* (*v.2.8.4*) (*Colaprico et al., 2016*) package. RNA-seq expression data were imported using the following parameters: project="TCGA-BRCA"; data. category="Transcriptome Profiling"; data.type="Gene Expression Quantification"; workflow. type="HTSeq - FPKM"; legacy = FALSE. TCGA expression Ensembl IDs were mapped to gene symbols using *EnsDb.Hsapiens.v86*. For visualization and Kruskal-Wallis tests for differences across groups, samples were restricted to those derived from female subjects and samples annotated as either matched normal or PAM50 Normal, LumA, LumB, Her2 and Basal subtypes (n=1201) and FPKM +1 values were log2 transformed. For Machine Learning, samples were restricted to matched normal samples and PAM50 subtypes LumA, LumB, Her2 and Basal (n=1161). PAM50 Normal samples were excluded due to small sample size. FPKM values of transcripts mapping to a gene were summed, and FPKM +0.1 values were log2 transformed consistent with GSE81540 pre-processing.

## GTEx Data

GTEx raw count data, sample pheno data and feature data were downloaded using the recount3 (*v1.2.3*) Bioconductor package (*Collado-Torres et al., 2017*; *Wilks et al., 2021*). For Machine Learning samples were restricted to those derived from female subjects (n=180). Raw counts were normalized and FPKM transformed using the *getRPKM* function and values were divided by 2 for paired-end data. FPKM values of transcripts mapping to a gene were summed, and FPKM +0.1 values were log2 transformed consistent with GSE81540 pre-processing.

## GSE81540 data

Normalized expression from breast cancers with matched pheno and clinical PAM50 annotation (GSE81540) (*Brueffer et al., 2020*; *Brueffer et al., 2018*; *Dahlgren et al., 2021*) were downloaded from GEO. Available pre-processed data represent summed transcript FPKM values for each gene that were log2 transformed (FPKM +0.1). Raw data were not provided. For Machine Learning samples were restricted to PAM50 subtypes LumA, LumB, Her2 and Basal (n=3184). No sex information was annotated, and all samples were used.

## TCGA survival data

Curated TCGA survival data was downloaded from *Liu et al., 2018*. Survival data were restricted to breast cancers and female subjects. Overall survival (OS) and progression-free interval (PFI) (*Hudis et al., 2007*; *Punt et al., 2007*) time and event data were used. Early and late cancer stage was defined from the annotated AJCC pathologic tumor stage: early stage included Stages I, IA, IB, II, IIA, and IIB; and late stage included Stages III, IIIA, IIIB, IIIC, and IV. Patients with missing age at pathologic diagnosis and cancer stage were excluded. Survival data for n=1072 patients were merged with scaled patient-level expression data (mean value of sample replicates) from primary tumors.

## scRNA-seq public data sets

For visualization of epithelial lineage-marker expression at single-cell resolution and single-cell gene set enrichment analysis (scGSEA), three data sets were used: (1) GSE161529 (*Pal et al., 2021*) scRNA-seq data from 13 non-tumorigenic breast tissue samples (19-69y) from reduction mammoplasties were kindly provided as a Seurat object by the group of Dr. Andrea Bild; (2) GSE174588 (*Nee et al., 2023*) scRNA-seq data from 11 non-tumorigenic noncarrier (24-50y) and 11 *BRCA1*[+/mut] (21-54y) breast tissue samples from reduction mammoplasties, prophylactic mastectomies, and contralateral to DCIS/tumor were kindly provided as a Seurat object by Dr. Kai Kessenbrock's group with annotated cell states; (3) GSE198732 (*Murrow et al., 2022*) scRNA-seq data from 28 healthy reduction mammoplasty tissue samples (19-39y) were downloaded from NCBI GEO as a Seurat object.

### SSP OMINER ChIP-Seq data

Cistromics ChIP-seq data for each gene of interested was obtained from The Signaling Pathways Project (SPP) Ominer database and restricted to female reproductive system mammary gland data across species. Graphical visualization of ChIP-Atlas MACS binding scores (±10 kb from TSS) for *GJB6*, *CLDN10*, *CLDN11*, *DSC3*, and *DSG3* were downloaded from Ominer.

### Functional annotation of genes of interest

Functional roles of genes highlighted in the manuscript were first explored in The Human Gene Database (*GeneCards v5.0*, https://www.genecards.org/) and subsequent literature search. Transcription factors were identified based on *Lambert et al., 2018*.

## Quantification and statistical analysis

All quantification and statistical analyses were conducted using using R (3.5.0) (*R Development Core Team, 2018*; https://www.R-project.org/) and Bioconductor (3.7) (*Huber et al., 2015*; https://www.bioconductor.org/) unless otherwise noted.

### Linear regression on expression values between organoids and 4th passage cells

Transcriptomes in LEPs and MEPs from reduction mammoplasty HMECs at 4th passage (n=11 < 30 y, n=8 > 55 y) and uncultured organoids (n=4 < 30 y, n=3 > 55 y LEPs; n=3 < 30 y, n=1 > 55 y MEPs) were compared using mean subject-level batch-adjusted rlog values calculated for each lineage and age group in organoids and 4th passage cells. Linear regression was performed on the gene expression value means from cells isolated from organoid vs. 4th passage culture in each of the RM LEP <30 y, LEP >55 y, MEP <30 y, MEP >55 y subsets. Regression lines and 95% confidence intervals were plotted along with the y-intercept and slope of the line. Linear regression $R^2$ and p-value were reported. Residuals were calculated from the linear model (*stats::lm*). Distribution of residuals were plotted, and mean value and sd of the residuals were calculated. Transcripts with absolute value of model residuals ≥6 (~4*sd) in either of the four subsets were considered outliers.

### Differential expression analysis

For lineage-specific and age-dependent differential expression (DE) analysis, the RNA-seq *edgeR* (*v.3.22.5*) (*Robinson et al., 2010*) filtered and normalized expression of 17,328 genes (see *DGEList* object in RNA-seq Pre-Processing) were subsetted for 4th passage LEP and MEP RM samples (<30 y, m=32 LEP and MEP samples, n=11 subjects;>55 y, m=22, n=8). DE analysis was performed on sample-level data with linear modeling in *limma* (*v.3.36.5*) (*Law et al., 2014*; *Ritchie et al., 2015*) using the *voom* implementation. The experimental design group was defined by the combination of cell type and age group. Batch was modeled into the design matrix (~0 + Design Group +Batch). The four contrast terms included comparisons of MEP vs. LEP lineages in younger <30 y and MEP vs. LEP in older >55 y groups for lineage-specific DE, and older vs. younger cells in LEP and older vs. younger cells in MEP lineages for age-dependent DE. Sample replicates, as well as the paired nature of MEP/LEP samples were accounted for by calculating the correlation between measurements using the *limma::duplicateCorrelation* function (*Smyth et al., 2005*) and blocking for subject ID. Because the calculated correlations changed the *voom* weights slightly, *voom* was re-run for a second time. Correlations were then re-calculated using the new *voom* weights. Linear modeling (*limma::lmFit*) was performed on the *voom* transformed data, with blocking for subject ID. Empirical Bayes moderation (*limma::eBayes*) of computed statistics was then applied.

For lineage-specific DE, contrasts between LEP and MEP in younger <30 y and in older >55 y women were assessed. Lineage-specific DE threshold was set at Benjamini-Hochberg (BH) adj. p<0.001, abs(lfc) ≥1 in each age cohort. LEP-specific and MEP-specific genes were defined as those with lineage-specific DE in younger <30 y women. MEP-specific and LEP-specific genes were indicated by (+) lfc and (-) lfc respectively. Lineage-specific DE was then annotated as lost, gained or maintained in older women. For age-dependent analyses, contrasts between <30 y and >55 y LEPs and <30 y and >55 y MEPs were assessed, and age-dependent directional changes were defined at DE BH adj. p<0.05 in each lineage. Genes upregulated in older cells and younger cells were indicated by (+) lfc and (-) lfc, respectively.

DE analysis was also performed in *limma* (*v.3.36.5*) (*Ritchie et al., 2015*) on a publicly available microarray dataset from normal primary breast tissue, GSE102088 (*Song et al., 2017*). Normalized

log$_2$ expression data were downloaded from the Gene Expression Omnibus (GEO) using *GEOquery* (*v.2.48.0*) (*Davis and Meltzer, 2007*). Gene-level data were calculated as the mean expression across all probes mapped to a gene. The experimental design group was defined by age group: younger <30 y (n=35), middle-aged >30 y<55 y (n=68), and older >55 y (n=11), in the design matrix (~0 + Design Group). The three contrast terms included pairwise comparisons between the three age groups. Linear modeling and empirical Bayes moderation of computed statistics with trend parameter = TRUE were applied. Significant age-dependent DE in normal primary tissue was defined as genes with DE adj. p<0.05, while nominally significance level was defined as genes with DE unadj. p<0.05.

## Differential variability analysis

For differential variability (DV) analysis, RNA-seq count data were batch-adjusted using *ComBatSeq* (*sva_devel*) (*Zhang et al., 2020*) and bach-adjusted values were normalized using TMM method (*MDSeq v.1.0.5::normalize.counts*); the mean value of *ComBatSeq* batch-adjusted normalized expression data was then taken for subjects with multiple samples. DV analysis was performed using *MDSeq* (*v.1.0.5*) (*Ran and Daye, 2017*) on subject-level *ComBatSeq* batch-adjusted normalized count data. The experimental design group was defined by the combination of cell type and age group in the design matrix (~Design Group). Outlier removal, which removes outlier genes that are influential upon the effects of covariates, was performed prior to DV analysis with the following parameters: minimum valid sample size threshold was set to 5 samples and significance level cutoff of outlier selections was set to 0.05. Analysis was restricted to the 14,601 genes whose variances could be estimated after outlier removal (*MDSeq::remove.outlier*) at cut-off significance level = 0.05. The two contrast terms included comparisons of older vs. younger cells in LEP and older vs. younger cells in MEP lineages for age-dependent DV. *MDSeq* was run with default parameters. Testing results were extracted (*MDSeq::extract.ZIMD*) using an inequality test with the following parameters: lfc threshold for inequality testing was set at 0 and with BH *p*-value adjustment method.

For age-dependent analyses, contrasts between <30 y and >55 y LEPs and <30 y and >55 y MEPs were performed, and age-dependent variant changes were defined at DV BH adj. p<0.05 in each lineage. Genes with increases in variances (higher lfc dispersion) in older cells and younger cells were indicated by (+) lfc and (-) lfc in dispersion, respectively.

## scRNA-seq analysis

scRNA-seq analyses were carried out in *Seurat* (*Hao et al., 2021*). All datasets were preprocessed to select for identified epithelial cell types and subjected to dimensionality reduction via Uniform Manifold Approximation and Projection (UMAP). For the (*Pal et al., 2021*) and (*Murrow et al., 2022*) datasets cells were included if they expressed 500–6000 unique transcripts, contained 1500–40,000 mRNA molecules, and had less than 15% contamination of mitochondrial genes. For the (*Nee et al., 2023*) dataset cells were included if they expressed 500–4000 unique transcripts, contained 1500–40,000 mRNA molecules, and had less than 10% contamination of mitochondrial genes. Genes that were not expressed in at least 10 cells were excluded from downstream analysis. Epithelial cells were subset from each dataset, log normalized by a scale factor of 10,000, variable genes were identified with *FindVariableFeatures* using VST as a selection method, PCA was performed using variable genes, and UMAPs for each dataset were generated using the first 5 dimensions. Epithelial subsets were renamed to be consistent across datasets with Luminal1 cells representing mature luminal cells and progenitors, Luminal2 cells containing hormone sensing luminal cells, and Basal cells being comprised of basal and myoepithelial populations. *scCustomize* (*Marsh, 2023*) package was used to visualize the expression of single lineage markers on UMAPs from each dataset. Single-cell Gene Set Enrichment Analysis (scGSEA) was performed using the *escape* (*Borcherding and Andrews, 2021*) package that utilizes *UCell* (*Andreatta and Carmona, 2021*) to execute and visualize GSEA across individual cells. Comparisons between cell types and groups were conducted using standardized mean differences with significance levels calculated by subtracting the means of two groups and dividing by the pooled standard deviation not accounting for sample size of each group (*Andrade, 2020*).

## Gene Set Enrichment Analysis (GSEA)

Fast gene set enrichment analysis (*fgsea v.1.6.0*) (*Korotkevich et al., 2021*) was used to identify age-dependent enrichment of Molecular Signatures Database (MSigDB) (*v7.2*, http://www.gsea-msigdb.org/gsea/msigdb/index.jsp) hallmark gene sets (*Liberzon et al., 2015*) in LEPs or MEPs. MSigDB

gene symbols were mapped to Ensembl IDs using *org.Hs.eg.db* (*v.3.6.0*) (*Carlson, 2018*) and mapped to the RNA-seq datasets. Fast GSEA was performed on either rank-ordered DE t-statistics (*limma*) or DV test statistic for dispersion (*MDSeq*) using 1000 permutations. Minimal and maximum gene set sizes were set to 15 and 500, respectively. Enriched gene sets were defined as those with enrichment BH adj. $p<0.05$. For bulk tissue GSEA analysis (GSE102088,<30 y n=35,>55 y n=11), gene sets were constructed from age-dependent genes in LEPs: (i) 251 genes that were differentially upregulated in young <30 y LEPs; and (ii) 220 genes that were that were differentially upregulated in old >55 y LEPs. Age-dependent enrichment was assessed in bulk tissue using DE rank-ordered test statistics; enrichment was similarly defined at BH adj. $p<0.05$.

## Single-sample Gene Set Enrichment Analysis (GSEA)

Gene set enrichment scores for individual samples were computed single-sample GSEA (ssGSEA) (*Subramanian et al., 2005*; *Barbie et al., 2009*) implemented in the GSVA R package (*Hänzelmann et al., 2013*).

## Lineage-specific ligand-receptor pair interactions

Ligand-receptor pairs (LRPs; *Ramilowski et al., 2015*) gene symbols were checked against *EnsDb. Hsapiens.v86* (*v2.99.0*) database (*Rainer, 2017*) gene symbols and then mapped to Ensembl IDs. Both ligand and receptor median expressions in LEPs and MEPs from younger and older women were calculated from *ComBat* batch-adjusted normalized rlog values. Lineage-specific LRPs were defined based on either the LEP-specific or MEP-specific (DE adj. $p<0.001$ and fold-change $\geq2$) expression of either the ligand and/or its cognate receptor in the younger cohort. Lineage-specific LRP interactions were considered to be lost in the older cohort when lineage-specific DE of the ligand and/or its cognate receptor was lost in the older cells (not passing the DE at adj. $p<0.001$, abs(lfc) $\geq1$ threshold). Lineage-specific LRP interactions between ligands and cognate receptors in the younger cohort and lineage-specific LRP interactions lost in the older cohort were visualized in an interactome map (*migest v.1.8.1, circlize v.0.4.8*) (*Abel, 2019*; *Gu et al., 2015*). Ligands were connected by chord diagrams from the cell type expressing it (cell type-L-gene symbol) to the cell type expressing its cognate receptor (cell type-R-gene symbol). Functional network enrichment of LRPs in the younger cohort and LRPs lost in the older cohort were performed in STRING database (https://string-db.org/) and enriched KEGG pathways (FDR $p<0.05$) were reported.

## Age-dependent DE protein-protein interactions and functional network analysis

Protein-protein interaction (PPI) analysis was performed using the *STRING* database (*v.11*, https://string-db.org/). All possible PPI are considered using all active interaction sources and setting minimum require interaction score to the lowest confidence threshold of 0.150. Network was visualized in *igraph* (v.1.2.5) (*Csárdi and Nepusz, 2006*), and only the largest fully connected main network of genes was plotted. Community detection was performed on this main network using optimal community structure algorithm in terms of maximal modularity score in *igraph* (*igraph::cluster_optimal*) (*Brandes et al., 2008*). Each community was then analyzed in *STRING* for functional network enrichment (FDR $p<0.05$) and common functional terms were summarized and reported.

## Non-parametric Kruskal-Wallis and Wilcoxon test:

*Limma*-based genome-wide DE analysis was not performed on publicly available gene expression datasets from breast cancer tissue. Instead, analysis was limited to genes of interest, and either non-parametric Wilcoxon test (unpaired for independent samples and paired for non-independent samples) or Kruskal-Wallis (KW) test (*rstatix v.0.5.0::kruskal_test*) (*Kassambara, 2020b*), a one-way ANOVA on ranks, was used to determine differences in $\log_2$(FPKM +1) values between two or more groups respectively. Wilcoxon and KW p-values were corrected for multiple testing (BH) across all features examined; in the TCGA cohort, KW test was performed across PAM50 breast cancer subtypes as well as matched normal tissue; p-values were adjusted across all age-dependent DE and DV genes identified in LEPs. For multiple groups, post hoc pairwise comparison between groups was performed using Wilcoxon test (*rstatix v.0.5.0::wilcox_test*) (*Kassambara, 2020b*). Post hoc Wilcoxon p-values were corrected for multiple testing (BH) across all pairwise comparisons and across all features examined; in the TCGA cohort, Wilcoxon p-values were adjusted across pairwise subtype comparisons and across all DE and DV genes. Paired Wilcoxon tests were also

perfored to compare primary tumors to their matched normal tissue dependent of patient ID and p-values were adjusted across subtypes. Wilcoxon and KW BH-adj. p-values were likewise annotated (*ggpubr v.0.2.5*) (*Kassambara, 2020a*) at different significance levels p<0.05 (*),<0.01 (**),<0.001 (***),<0.0001 (****). Non-parametric Wilcoxon and KW tests were similarly used to compare ssGSEA signature scores between two or more groups respectively.

## Lepage test on location and scale

Two-sample Lepage test (*nonpar v.0.1–2*) (*Pepler, 2017*) is a joint non-parametric test of equality for location (central tendency) and scale (variability). Lepage test was performed on the subject-level *ComBat* batch-adjusted normalized rlog expression values of genes encoding for junction proteins between younger and older cells in each lineage. Significant age-dependent modulation of genes for cell-surface junction proteins in LEPs and MEPs were defined at p<0.05.

## Kolmogorov-Smirnov test on lineage-specific DE lfc:

Non-parametric two-sample Kolmogorov-Smirnov (KS) test (*stats::ks.test*) on the equality of distributions performed to compare distributions of lineage-specific DE lfc in younger and in older cells. Significance defined at p<0.05.

## T-test on the differences of DE lfc between age groups

One-sided t-test (*stats::t.test*) performed on the distribution of pairwise differences in lineage-specific DE lfc between age groups (lfc in young - lfc in old) to test if the mean of all values is different from 0. Significance defined at p<0.05, nominal significance defined at p≤0.1.

## T-test on qPCR values between experimental treatments

Two-sided Student's t-test performed to compare normalized expression between the two groups in co-culture experiments. Significance defined at p<0.05.

## Fisher's exact test

Median expression levels of age-dependent DE and DV genes from LEPs were assessed in the TCGA cohort across PAM50 LumA, LumB, Her2 and Basal intrinsic subtypes. Contingency tables reflecting number of genes by PAM50 subtype and by highest and lowest median expression levels for: DE genes upregulated in young LEP; DE genes upregulated in old LEP; DV genes with higher variance in young LEP; and DV genes with higher variance old LEP were tabulated. Fisher's exact test were performed in R (*stats::fisher.test*) on each of the contingency tables and p-values were adjusted using the Bonferroni method.

## Unsupervised hierarchical clustering

Unsupervised hierarchical clustering was implemented using Ward's clustering criterion (ward.D2) agglomerative method with Euclidean distances as distance metric. Heatmaps were generated using *gplots* (*v.3.0.3::heatmap.2*) *and pheatmap* (*v. 1.0.12*) (*Kolde, 2019*; *Warnes et al., 2020*) packages. Dendrograms were plotted using the *dendextend* (*v1.13.4*) (*Galili, 2015*) package. Approximately unbiased (AU) p-values and bootstrap probability (BP) were calculated and annotated using *pvclust* (*v.2.2–0*) (*Suzuki and Shimodaira, 2006*) package which assesses uncertainty in hierarchical clustering analysis. Clusters with AU p≥0.95 were highlighted.

## Machine earning

Machine Learning gene expression datasets of normal and cancer breast tissue were obtained: GSE81540 (n=3184) (*Brueffer et al., 2020*; *Brueffer et al., 2018*; *Dahlgren et al., 2021*) GTEx (n=180) and TCGA (n=1161) as described above. Analysis was restricted to tissues from women annotated as normal or PAM50 LumA, LumB, Her2, and Basal subtypes (PAM50 Normal excluded due to small sample size). Gene features include the 536 of 591 age-dependent DE and DV genes between younger and older LEPs (adj. p<0.05) found in common across the GSE81540, GTEx and TCGA datasets. Training and cross-validation was performed on 75% of TCGA data (matched normal = 84, PAM50 LumA = 425, LumB = 156, Her2=62, Basal = 146). Test sets for model performance evaluation include 25% of TCGA the model had not seen (matched normal = 28, PAM50 LumA = 141, LumB = 51, Her2=20, Basal = 48) and an independent dataset composed of normal tissue from GTEx (n=180) and breast cancer tissue from GSE81540 (PAM50 LumA = 1709,, LumB = 767,

Her2=348, Basal = 360). Machine Learning was performed using the *caret* (*v.6.0–88*) R package (*Kuhn, 2024*) using elastic net implementation from *glmnet* (*v.4.1–2*) (*Friedman et al., 2022*) using 10-fold cross validation with 3 repeats. To address class imbalance, we used a hybrid subsampling technique implemented using the SMOTE algorithm in the *DMwR* (*v.0.4.1*) package (*Torgo, 2010*). Custom summary functions of model performance were defined, and the model was optimized for mean balanced accuracy which we found to perform better when large class imbalances exist. Model performance in test sets were evaluated using AUC calculated with the *MultiROC* (*v.1.1.1*) R package (*Wei and Wang, 2020*) and reported as follows: (i) AUC of each group vs. the rest; (ii) micro-average AUC calculated by stacking all groups together; and (iii) macro-average AUC calculated as the average of all group results. Variable importance values were scaled for each class (range from 0–100%), and gene predictors were identified as genes with scaled variable importance contribution to the predictive model (scaled variable importance >0% in at least one class). Genes with scaled variable importance ≥25% in prediction of at least one class were visualized; gene expression the top 5 predictors in each class were further analyzed in the TCGA breast cancer cohort.

## Cox proportional hazards regression analysis

Multivariate Cox proportional hazards regression models (*Harrell, 2015*), implemented in the *survival* (*v.3.5–3::coxph*) R package (*Therneau, 2023*; *Therneau and Grambsch, 2000*), were used to simultaneously assess the effect of all top ML predictors – defined as genes with subtype scaled variable importance ≥25% – in each subtype on OS and PFI, with age at diagnosis and cancer stage (early vs. last) as additional covariates. Analyses were performed in the PAM50 LumA, LumB, and Basal subtypes; PAM50 Her2 subtype was underpowered and models did not converge and was excluded from the report. Forest plots of hazard ratios with 95% confidence interval and Wald p-values annotated were plotted using the *forestmodel* (*v.0.6.4*) R package (*Kennedy and Wang, 2022*). The fraction of top predictors with significant contribution (p<0.05) to OS and PFI were reported.

