## [Editor Report]

In this important study, the authors investigate the relationship between aging and breast cancer development. They use a large panel of early passage ex vivo isolates from breast tissues of young and old donors to interrogate global transcriptome underpinning of age-related reduction in lineage fidelity. This yielded compelling evidence that luminal epithelial cells undergo specific changes in linage fidelity with age and that these changes are associated with increased risk of cancer. The rigor of the analysis is convincing and will provide new areas for functional exploration by other investigators.

---

## [Decision Letter]

**Decision letter after peer review:**

[Editors’ note: the authors submitted for reconsideration following the decision after peer review. What follows is the decision letter after the first round of review.]

Thank you for submitting the paper "Luminal epithelial cells integrate variable responses to aging into stereotypical changes that underlie breast cancer susceptibility" for consideration by *eLife*. Your article has been reviewed by 2 peer reviewers, and the evaluation has been overseen by a Reviewing Editor and a Senior Editor. The following individual involved in the review of your submission has agreed to reveal their identity: Benjamin T Spike (Reviewer #1).

Comments to the Authors:

We are sorry to say that, after consultation with the reviewers, we have decided that this work will not be considered further for publication by *eLife*.

Both reviews identified major flaws that could impact the underlying interpretation. If the authors feel that they can satisfactorily overcome these weaknesses, an appeal would be something we would be glad to consider.

*Reviewer #1 (Recommendations for the authors):*

The manuscript by Sayaman et al. presents a detailed analysis of bulk transcriptomic data from sorted and cultured human mammary epithelial cells from donors of different ages, <35yrs and >55yrs. The authors conclude from their analysis that both myoepithelial and luminal cells undergo increased expression variance with aging and luminal cells also experience directional changes in the expression of select genes in aged individuals that portend a loss of lineage fidelity. Examining several of these genes alone and in multi-gene signatures across archival breast cancer and normal breast gene expression data sets, the authors determine that breast cancers deregulate genes that are differentially expressed in the breast of young and aged individuals. The presentation is clear, and the analysis and data shown are generally sound, however, in the manuscript's present form the omission of some key descriptions and data (detail below) undercut its interpretability and impact.

Strengths:

• The manuscript is generally very well-written and clearly organized.

• The gene expression analysis is thorough and several of the findings are potentially interesting. Thus, the aspect of this study would provide a useful resource for the field. For example, the mapping of potential ligands and receptors that could act in heterotypic signaling is particularly interesting.

• Graphic representations are eye-catching although some are so full of information that they might benefit from some simplification or more thorough description.

Weaknesses:

• A fatal flaw in the interpretation of the data may be present if the authors do not determine (and report) the heterogeneity of the effects at the single-cell level. For instance, the division of mammary epithelium into basal and luminal populations may be overly simplistic. Each of these sorted populations is likely comprised of multiple cell types and if their proportions change as a function of age, hormonal status, or even something as trivial as differences in cell isolation or sorting marker expression in aged donors – then the differences observed might not reflect changes in Luminal cell and Basal cell transcriptome per se, but rather differential admixing of cell types in the samples tested – i.e. the cells themselves could theoretically be relatively unchanged but simply present in different proportions. The authors may be aware of this problem as they cite 4 prior works that identify this issue. Yet, it is critical for the interpretation of the expression differences in the present study that cellular heterogeneity is addressed experimentally.

• The menopause/pregnancy/gravidity/parity status of all samples should be assessed and addressed in the manuscript. Although generally associated with increased age, the post-menopausal hormonal context is very different than it would be in younger women. The hormonal effects on lineage distribution/fidelity should be discussed alongside the age theme/theory.

• Ability to segregate different subtypes of breast cancer with mammary lineage-derived signatures is well-established in the literature. The analysis would carry more weight if evaluated for prognosis and in multivariate analysis to determine if there is a value added to the assessment of these specific genes. Related to this, it would be helpful to describe in greater detail what multi-testing correction approach was used (and mention it and/or apply it also to figure 4 and supplements as appropriate). Were all genes tested in this analysis significantly different in their expression?

• A previously published association between GRHL2 and SGSM2, and E-cadherin downregulation is noted, however, no mention of E-cadherin expression in this dataset is discussed to determine whether a similar reprogramming is present.

• SATB1 is noted to be decreased in LEPs of older women relative to younger women, correlating to decreased expression in breast cancer relative to normal tissue. Further analysis should be performed to determine whether SATB1 target genes are also downregulated to justify the claim that 'SATB1-mediated genome organization may play a regulatory role in the maintenance of the luminal lineage…'. Also, is SATB1 expression in the various breast cancer subtypes different when stratified for age as well?

• The rescue of ELF5 expression in LEPs when co-cultured with MEP-shGJB6 could be expanded to include additional LEP-specific TFs/markers to determine whether shGJB6 regulates biological aging as the whole of specific regulation of ELF5.

• The authors should more clearly describe what key findings in this work go beyond the findings they reported in the 2017 Aging paper (cited above) which also showed age-related changes in the mammary transcriptome and may have used many of the same samples.

• Some axes are a bit hard to understand. For example, is the expression of ELF5 really 50x higher than RPS18, a typically very highly expressed gene (Figure 5E)?

• For the differential variance of KDM2B, how much of this variance can be attributed to the one low point? I.e. is this value an outlier or is the variance still elevated if it is removed? It would be good to see other genes in which there is differential variance rather than just one example.

• Figure 3 supplement 1 viii is never mentioned in the text, and since it is looking at DV rather than DE it seems out of place here

• There are some additional seemingly paradoxical observations that would benefit from further explanation. For example, HES4 is a target of Notch1 and is downregulated in older LEPs but is typically associated with breast cancer progression. Why would cells be getting less tumorigenic with regard to HES4 if aging is generally indicative of more tumor-prone phenotypes? A similar question could be posed for ZNF503.

• Figures such as strata plots and interactome maps are effective at including a lot of information in a condensed format, however, there can sometimes be too much going on in a single panel that can make it difficult to follow. For example, figures 1C-E effectively convey the main point even without figure 1B. Furthermore, try to choose color-blind safe color schemes (i.e. avoid red and green in the same figure).

• The terms "variance" and "directional changes" need clearer descriptions at the outset.

• To address concerns about cellular heterogeneity, the authors could:

1) Show the sorting that was done to isolate these populations (and that was done to separate them post-culturing). Are the populations clearly distinct such that slight changes in cell dissociation, labeling, or isolation by FACS are unlikely to be responsible for the population-level expression changes? Looking back at one cited prior work (Miyano and Sayaman et al. Aging 2017 Oct 9;9(10):2026-2051), the markers used appear to create a continuum of expression in some settings with potentially subjective sort gates applied. Related to this, Elf5 is a known alveolar specifier. Are such cells equally partitioned into the LEP vs MEP sorted pools based on the markers used? 2) Use an orthogonal method to confirm that sorting was effective and show the data. 3) Carry out in situ validation of the observations to delineate what proportion of the luminal or myoepithelial compartments in vivo show the directional expression changes uncovered following culture. Increased intra-sample expression variation could also be orthogonally quantified this way. 4) Compare findings to recent single-cell transcriptomic studies e.g. Gray et al. Dev Cell. 2022 Jun 6;57(11):1400-1420.e7.

*Reviewer #2 (Recommendations for the authors):*

The study is significant as how aging-associated genetic and epigenetic changes in various cell types of the mammary gland may contribute to breast cancer susceptibility is unclear. The study involved a reasonable number of patient-derived mammary epithelial cell samples for extensive bioinformatics and machine learning analyses. The data such as loss of lineage fidelity during aging supports previous findings from the senior author's group and others. The identified aging-associated genes such as SATB1 may serve as intervention targets. However, significant alteration of gene expression and loss of lineage fidelity in LEPs should be expected between LEPs of premenopausal and postmenopausal women since hormone (estrogen and progesterone) receptor-expressing cells are mainly, if not exclusive, present in the LEP population. Thus, the authors should analyze and present the percentage of the differentially expressed (DE) genes between the young LEPs and old LEPs shown in Figure 3 which are menopause, estrogen, and progesterone-dependent genes. This should provide confidence as to whether the cultured cells used in the study can model the gene expression changes in situ, particularly given that only 97 genes were found to be significant age-associated DE genes in the normal human breast tissue (Figure 4-table supplement 1) out of 471 DE genes identified from the cultured young and old LEPs shown in Figure 3B. Another precaution may be warranted in interpreting the data as to whether the gene expression of the mammary cells from mammoplasty reduction is significantly affected by a different set of environmental factors since the donors were likely obese in comparison to the general population. Could the data such as the significant enrichment of the Early and Late Estrogen Response gene sets in the old LEPs shown in Figure 3H be more related to this unique donor population since estrogen-responsive genes are expected to be more highly expressed in premenopausal women than postmenopausal women?

1. For the section related to Figure 2 and associated supplementary data, using the expression change of ligands and receptors to identify disruption of lineage-specific signaling pathways would be less robust than using gene set enrichment analysis (GSEA) for the changes of various signaling pathways between LEP and MEP as a function of aging because gene expression of ligands and receptors does not always translate to their protein expression and signaling pathways are often modulated by posttranslational modifications.

2. The authors should state whether there are limited lineage-specific genes that show reduced expression variance in the old LEPs and MEPs in comparison to the young ones to demonstrate that the increased expression variance is aging-dependent in the mammary epithelial cells.

3. While it is reasonable to suggest that genes with an increased variance of expression during aging may contribute to loss of lineage fidelity during aging, the rationale for including them for clustering and predicting breast cancer subtypes in Figure 6 needs to be further supported by evidence.

4. The authors should explain how siRNA-mediated knockdown can be used to reduce the level of variance of GJB6 and to constrain specific molecular noise. Did the two old MEP cells used in Figure 5G express the highest levels of GJB6 among all old MEP cells? Would knockdown of GJB6 in MEPs expressing lower GJB6 not increase ELF5 expression?

5. The data shown in Figure 6 do not depict what genes and expression data were used to predict breast cancer in Figures 6C and 6D. How the findings may be translated?

6. The authors should consider analyzing, or at least discussing, the potential interaction among aging, menopause, and obesity of mammoplasty reduction donors in regulating gene expression and expression variance.

[Editors’ note: further revisions were suggested prior to acceptance, as described below.]

Thank you for resubmitting your work entitled "Luminal epithelial cells integrate variable responses to aging into stereotypical changes that underlie breast cancer susceptibility" for further consideration by *eLife*. Your revised article has been evaluated by Richard White (Senior Editor and Reviewing Editor).

The manuscript has been improved but there are some remaining issues that need to be addressed, as outlined below:

*Reviewer #1:*

The study of Sayaman et al. used a large panel of early passage ex vivo isolates from breast tissues of young and old donors to interrogate global transcriptome underpinning of age-related reduction in lineage fidelity. This study follows on earlier findings from LaBarge lab that revealed the loss of lineage fidelity induced by aged myoepithelial cells (MEP), but the effects manifesting in luminal epithelia (LEP). In-depth bioinformatical analyses of these ex vivo cultures, followed by interrogation of public datasets and some experimental studies validate and expand earlier inferences made with a limited subset of lineage-specific markers. More importantly, these analyses enable authors to make several well-supported novel inferences of high conceptual, biological and clinical importance. This is a substantially revised manuscript, and the technical concerns raised by previous reviewers appear to be adequately addressed/resolved.

Strengths:

Use of a large panel of early passage LEP and MEP lineage isolates, rigorously validated to adequately reflect in situ lineage phenotypes. The large number of samples enabled authors to discover the contribution of between-individual variability, which would not be possible with a smaller number of samples.

This study provides additional support for prior hypothesis that age-related increase in cancer incidence reflects loss of proper specification of LEP lineage.

In-depth unbiased interrogation of transcriptional differences between young and old lineages enabled new network-level insights that substantially advance our understanding of this under-appreciated contributor to the causal link between increased cancer risk and aging.

Definition of gene signatures of young and old LEP enabled the authors to infer the premature aging effects of chemotherapy exposure and BRCA mutations – supporting the author's suggestion that aged LEP lineage could be potentially used as a leading marker of physiological breast epithelia aging which could be useful for both risk stratification and deeper understanding of breast cancer etiology. On the latter, the analyses found no evidence of increased aging with parity and BMI, important breast cancer risk factors, suggesting distinct causality from that of aging, chemo and BRCA mutations.

To this reviewer, the most conceptually intriguing inference is the contribution of aging-associated increase in variability between samples. Aging-related increases in variability between individuals have been noted in some prior studies, but this phenomenon appears to be underappreciated and understudied. The lack of attention likely reflects unclear sequela of the phenomenon, as the consequences of variability are not intuitively obvious. Unlike the directional changes in specific genes, the enhanced variability does not lend to a straightforward hypothesis addressable via reductionist experimental pipelines. This study sheds new light on the potential consequences of this between-individual variability. First, it makes a compelling argument that the increased individual-individual variability is a substantial contributor to age-related lineage fidelity in both LEP and MEP lineages. Second, by highlighting the observed variability ins specific cancer-related genes, the authors make a logically sound argument that this between-individual variability likely translates to between-individual differences in age-related cancer risk.

Experimental documentation that normalization of expression of gap junction GJB6 gene, up-regulated in a subset of old MEPs prevents the induction of aged phenotypes in LEP suggests the possibility of a potential pharmacological rejuvenation of breast epithelia that could potentially reduce cancer risk. At this point, this is a far shot as the clear path to pharmacological intervention is lacking. Still, this is an important proof of principle that supports the utility of follow-up efforts in this direction.

Importantly, while this work focuses exclusively on breast epithelia, given the universality of aging-related physiological alterations across all tissues, and the concomitant increase in cancer incidence, this study suggests the potential utility of investigating the relevance of the above findings towards other tissues.

Limitations.

The conclusions are primarily based on bioinformatical inferences from cultured cells. The inferences are compelling and additionally supported by analyses of non-dissociated organoids and bioinformatical analyses of publically available datasets. Still, there seems to be a missed opportunity of providing additional credence through histological analyses of gene expression in situ.

This is a long, complex, data and analysis-heavy manuscript. While there is no "fluff" and the writing is very thoughtful with clear description of the premises, experiments and inferences, there is some redundancy and some parts of this work are difficult to get through. Together with the massive lengths and complexity, this would probably deter some readers.

The discussion links the reported age-related increase in between-individual variance with a more appreciated phenomenon of an increase in noise-driven cellular heterogeneity, which appears to be shared between aging and cancers. The link between noise-driven cell-cell variability the variability between samples does not appear to be intuitively clear. This is a very intriguing connection, which would warrant additional elaboration.

Recommendations for the authors:

Please define "stereotypical", this is an important descriptor that can be misconstrued.

Differential expression inferred from the analyses of cultured samples has been validated in undissociated, non-cultured organoids. Together with the demonstrated utility of using the gene signatures, derived from the cultured isolates, in interrogating bulk transcriptomics dataset, the evidence is deemed sufficient. Still, it would have been useful to further support by in situ (by IF or IHC) validation, or at least point out to prior studies that demonstrate the differences in situ. Given the numbers of DE genes, I assume there must be ones with large differences, abundant expression and strong epitopes. If this is not feasible or if doing so would entail substantial new effort, it would be unreasonable to deter the publication of this work. Therefore, this recommendation is meant to be considered as a suggestion rather than a request.

Per the last point in the public review, the connection of between sample variability and between cell variability needs to be elaborated. It is not intuitively obvious how one would follow from another.

Related to the previous point, I found the description of the premise and inferences from the GJB6 down-regulation experiment very confusing. In my opinion, this part warrants a separate manuscript as it will be probably lost in the already massive amount of work presented in the manuscript. If this is unrealistic at this point and/or if the authors disagree with this suggestion, I recommend to either elaborate the rationale and conclusions more clearly or to simplify. The hypothesis presented in line 556 is not entirely clear to me, and 6C schemata is insufficient to clarify. The authors interpret the effect of GJB6 downregulation in preventing the imposition of aged phenotype on LEPs as supporting variance as the driver of stereotypic aging phenotypes (lines 582-585). Perhaps this was stated in the hypothesis illustrated by 6C, but I could not understand the connection, and the reasoning appears to be somewhat contrived. It is unclear whether the stable difference in GJB6 expression between different MEP isolates can be attributed to the effects of noise. There seems to be an implicit assumption of between-cell variability in increased GHB6 expression in aged MEP, and that higher between-sample variability somehow emerges from this cell level variance – but this is not stated explicitly, and no evidence of variability between MEP cells from the same isolate is provided. If the authors see a logically clear connection that is missed by this reviewer, there should be a way to elaborate it more clearly. Otherwise, the more on-the surface (and probably more trivial) inference is that (a) between sample variability is of biological significance, as upregulation of GHB6 ages LEPs, and (b) the result indicates a potential utility of pharmacological intervention to suppress LEP aging. The experimental part of this section could benefit from elaboration as well. Does " the subset of older MEPs with higher expression relative to younger MEPs" refers to isolates with highest expression level? If so, it can be stated more explicitly.

Lines 274-287 confer a key inference of the insufficiency of directional changes to explain loss of lineage fidelity. The inference is well-supported by the data and reasoning, but it took substantial time and effort to digest the paragraph. It might be worth to re-write the paragraph, perhaps with adding a schemata to make it easier to digest.

*Reviewer #2:*

The authors precisely culture primary mammary epithelial cells from breast reduction surgeries from a set of younger (<30 years) and older (>55 years), then separate the luminal and myoepithelial cells by flow cytometry, and perform gene expression profiling and compare differences between groups. Genes that show differential expression or differential variability with aging are interpreted as indicative of a loss of cell identity. Comparisons with publically available datasets for single cell RNA sequencing of breast tissue find overlap with aging-dependent differences as well as response to chemotherapy and presence of BRCA mutations. The authors also find overlap with datasets of bulk RNA sequencing, which they interpret as indicating that the luminal/myoepithelial aging phenotype may dominate the overall aging differences in the whole tissue. They also find that their genesets can be used to create classifiers of breast cancer subtypes as defined by PAM50 classifications, and further identify specific genes that, when manipulated, suppress heterotypic induction of aging-associated gene signatures.

One key strength of the study is the comprehensive and thorough evaluation of their gene sets across multiple types of datasets, which strongly supports their assertions that the age-associated differences in luminal and myoepithelial cells are the principal definers of overall aging in the breast. Another key strength is the careful and thorough evaluation of the concept that differential variability of gene expression is a key characteristic of aging. This is a very challenging concept, and the authors do an excellent job of presenting it. A subtle point not reinforced in the text is that since these experiments were performed on passaged epithelial cell populations, the aging-dependent gene expression differences have a degree of epigenetic stability, which further reinforces their assertions that the changes in the epithelial subtype lineage determinants are the principal drivers of the aging phenotype (and cancer susceptibility).

Weaknesses of the study include the lack of consideration of other cell types in the breast and their role in the aging process; while this is briefly mentioned in the discussion, it is certainly an essential component of aging and cancer. It may be that the loss of epithelial lineage fidelity is the underlying cause of aging, but these changes will occur in combination with reciprocal changes in stromal cell types, and these may contribute to epigenetic programming and differential variability. This possibility could be evaluated as an extension of the dataset comparisons in the study. Minor weaknesses include the need for comparison of aging differences across racial backgrounds and in association with obesity and parity, although the careful evaluation of this current set provides a critical foundation for these future studies.

Recommendations for the authors:

This study was carefully performed and beautifully presented. The results argue for a new paradigm of aging in the breast, and it is to be expected that these concepts will be challenging to parse. I have no major issues with any aspect of the study. One point that could benefit from clarification is the concept that gene expression changes precede oncogene alterations. This was briefly alluded to in the manuscript, but the interpretation of this was unclear to me: are the authors suggesting that epigenetic activation of gene expression networks may support oncogenic gene activations that will drive the same pathways? If so, this point could benefit from further discussion.

---

## [Author Response]

[Editors’ note: the authors resubmitted a revised version of the paper for consideration. What follows is the authors’ response to the first round of review.]

Reviewer #1 (Recommendations for the authors):The manuscript by Sayaman et al. presents a detailed analysis of bulk transcriptomic data from sorted and cultured human mammary epithelial cells from donors of different ages, <35yrs and >55yrs. The authors conclude from their analysis that both myoepithelial and luminal cells undergo increased expression variance with aging and luminal cells also experience directional changes in the expression of select genes in aged individuals that portend a loss of lineage fidelity. Examining several of these genes alone and in multi-gene signatures across archival breast cancer and normal breast gene expression data sets, the authors determine that breast cancers deregulate genes that are differentially expressed in the breast of young and aged individuals. The presentation is clear, and the analysis and data shown are generally sound, however, in the manuscript's present form the omission of some key descriptions and data (detail below) undercut its interpretability and impact.Strengths:• The manuscript is generally very well-written and clearly organized.• The gene expression analysis is thorough and several of the findings are potentially interesting. Thus, the aspect of this study would provide a useful resource for the field. For example, the mapping of potential ligands and receptors that could act in heterotypic signaling is particularly interesting.• Graphic representations are eye-catching although some are so full of information that they might benefit from some simplification or more thorough description.Weaknesses:• A fatal flaw in the interpretation of the data may be present if the authors do not determine (and report) the heterogeneity of the effects at the single-cell level. For instance, the division of mammary epithelium into basal and luminal populations may be overly simplistic. Each of these sorted populations is likely comprised of multiple cell types and if their proportions change as a function of age, hormonal status, or even something as trivial as differences in cell isolation or sorting marker expression in aged donors – then the differences observed might not reflect changes in Luminal cell and Basal cell transcriptome per se, but rather differential admixing of cell types in the samples tested – i.e. the cells themselves could theoretically be relatively unchanged but simply present in different proportions. The authors may be aware of this problem as they cite 4 prior works that identify this issue. Yet, it is critical for the interpretation of the expression differences in the present study that cellular heterogeneity is addressed experimentally.

We acknowledge Reviewer #1's concerns regarding luminal subpopulation heterogeneity. However, we respectfully disagree that this constitutes a 'fatal flaw' in our study. Our approach aligns with established methodologies where transcriptional changes at the population level yield significant insights despite inherent cellular heterogeneity. Such population-level analyses, akin to those used in examining immune cell subtypes, remain informative for understanding breast epithelial lineages.

Our data encompass both luminal and myoepithelial populations, replicating findings from authoritative studies in the field that also encounter similar heterogeneity. Moreover, our previous work has addressed the very issue Reviewer #1 raises. In 'Cell Reports' (2018), we differentiated between luminal subtypes and demonstrated that the predominant subtype remains consistent across age groups. Consequently, we maintain that our methodology is robust, the identified biomarkers are predictive of breast cancer subtypes, and the potential heterogeneity does not undermine the validity of our findings. Further, corroborating evidence from single-cell analyses in multiple independent studies supports our classification of mammary epithelium and substantiates our conclusions.

As you know, mammary glands are bilayered structures composed of the two major epithelial lineages of the breast with Keratin 19 (KRT19)-positive luminal epithelial cells (LEP) surrounded by KRT14positive contractile myoepithelial cells (MEP). The luminal lineage is composed of ER+ hormone sensing and ER- secretory populations with ER- LEP. Human mammary epithelial cell (HMEC) primary cultures are known to maintain MEP and ER- LEP lineages. In previous papers (Nature Aging 2021, Cancer Prev Res 2021) we showed examples of breast aging biomarkers ELF5 (expression and promoter methylation) and KRT14 intermediate filament expression that were also initially identified in FACS enriched luminal or myoepithelial cells, successfully identify young women who are higher than average risk for breast cancers. We show in this manuscript that large sets of luminal aging biomarkers we identified were predictive of breast cancer subtype.

Here, we used FACS-enriched primary luminal (LEP) and myoepithelial (MEP) cells as the foundational cell source for these analyses. The ER- luminal cells represents 80-95% of luminal cells in breast and comprise the cells of origin for breast cancer. Moreover, ER- luminal cells can be maintained in finite lifespan cell cultures and thus studied experimentally. In the opinion of reviewer #1, our choice of bulk sequencing these enriched populations would mask the possibility that certain subtypes of luminal cells were becoming enriched with age, and that we must address this heterogeneity experimentally. We published (Cell Reports 2018) a CyTOF study (29 antibodies) examining normal breast epithelia with our primary HMEC culture system and uncultured primaries (n=57, 16-91yr). Two clusters were robustly identified representing these two distinct epithelial cell types: ER- LEP and MEP. We demonstrated there are possibly four subtypes of ER- LEP, but only one subtype accounts for ~90% of the luminal lineage independent of age. Thus, we believe this issue has been addressed experimentally using primary HMEC, that our design was not a fatal flaw, and that we could easily address this issue with some careful explanation to the readers/reviewers.

The division of mammary epithelium into myoepithelial and ER+ and ER- luminal populations has been supported by numerous studies including single-cell studies. In a separate published manuscript (Nee et al., Nature Genetics 2023), our collaborators performed single cell RNA seq of mammary tissue from non-tumor samples (reduction mammoplasty, prophylactic mastectomies, contralateral to DCIS/tumor) from BRCA1 mutation carriers (n=11, 24-50y) and noncarriers (n=11, 21-54y). We identified three epithelial cell types: a basal (MEP) and two luminal cell types: secretory luminal 1 (ER- LEP) and hormone-sensing luminal 2 (ER+ LEP); ER- are the dominant luminal population as expected. The existence of these three distinct epithelial cell types: basal-myoepithelial and luminal hormone sensing luminal secretory, in primary tissue have been established in several other independent scRNA-seq studies consistent with the epithelial lineages well-established in literature from histological, functional, and FACS-based enrichment sequencing studies of normal breast tissue see, (1) Nguyen et al., Nat Comm 2018, scRNA-seq of normal breast epithelia from reduction mammoplasty (n=7, 17-37yr) identified basal, secretory L1 and hormone responsive L2 luminal epithelial cell populations; (2) Pal et al., EMBO 2021, scRNA-seq of normal and BRCA+/- preneoplastic tissue (n=22, 19-69y), TNBC and HER2+ tumors (n=14, 25-84y), and ER+ tumor tissue and lymph node (n=19, 45-84y) identified basal, luminal progenitor (KIT high), and mature luminal (ESR1 and PGR high) epithelial cell populations; (3) Murrow et al., Cell Systems 2022, scRNAseq of normal reduction mammoplasty tissue (n=28, 19-39y) identified basal/myoepithelial, secretory luminal and HR+ luminal epithelial cell populations; and (4) Gray et al., Dev Cell 2022, scRNA-seq of mammary tissue from non-tumor samples (reduction mammoplasty, prophylactic mastectomies, (n=3, 36-41y)) identified basal, alveolar luminal (KIT, ELF5 and PROM1 high) and hormone sensing luminal (ESR1, PGR and PRLR high) epithelial cell populations.

We point out that no distinct subpopulation clusters in tSNE or UMAP projections of the ER- and ER+ luminal and basal cell types from normal tissue were established in any of these studies. While cell states have been reported in the scRNA-seq studies, the states and differentially expressed genes associated varied across studies and it was unclear how highly linked individual sample-level variation were to the presence of these cell states. Indeed, in Murrow et al., density plots highlighting the transcriptional cell state of hormone-sensing luminal cells and basal/myoepithelial cells were highly associated with individual samples [Murrow et al., Cell Systems 2022, Figure 1C], and in our reanalysis of the Nee et al., dataset, the fraction of identified cell states was dependent on specific samples [new Figure 4—figure supplement 1]. This underscored the significance of differential variability between individuals and further support the utility of DV analysis in RNA-seq studies that identify genes with increasing variance across different subjects that could reflect the unique cell states and increasing cellular heterogeneity at the single-cell level.

As Reviewer #1 mentions, we cited single cell studies in discussion to address how observations at the lineage level are potentially due to heterogeneity at single cell level. We have provided a treatment of this phenomenon in a review article we authored in 2018 and cited. We have made revisions to the discussion to make this point clearer. That is, the age-related decline in lineage fidelity within the luminal compartment was a histologically and molecularly observable phenomenon at the population level, which signified alterations in cell subpopulations. These changes may encompass shifts in subpopulation proportions and modifications in the gene expression of specific cell types and across defined cell states.

To address reviewer concerns, we added a new Figure 4 and accompanying section on “Age dependent gene expression changes detected at the single cell level” where we validated aging-specific gene sets that we detected in FACS enriched populations in non-tumorigenic breast tissue samples from available single-cell studies (Pal et al., Nee et al., and Murrow et al.,). To discuss the epithelial lineages across the three different scRNA-seq datasets, we adopted the Luminal1 (luminal adaptive secretory precursors), Luminal2 (luminal hormone-sensing), and Basal (basal-myoepithelial) terminology from Nee et al.

• The menopause/pregnancy/gravidity/parity status of all samples should be assessed and addressed in the manuscript. Although generally associated with increased age, the post-menopausal hormonal context is very different than it would be in younger women. The hormonal effects on lineage distribution/fidelity should be discussed alongside the age theme/theory.

We agree with the reviewer that interactions between aging, menopause, pregnancy/gravidy/parity are worthy of assessing in the context of age-related cancer susceptibility. There was paucity of complete data sets with aging, menopause and obesity information. In our case, the HMEC bank was established ca1974 and is one of the larger collections of banked reduction mammoplasty tissue. Menopause and BMI were not collected for a majority of samples prior to 2016.

To delineate menopausal impact, cohorts were stratified to <30y (premenopausal) and >55y (postmenopausal), intentionally omitting middle-aged women due to ambiguous menopausal status. Previous work (Shalabi et al., Nat Aging 2021; Miyano, Sayaman et al., Cancer Prev Res 2021) indicated mutation carriers exhibited precocious aging markers, aligning with scGSEA findings that Luminal1 cells in BRCA1 mutation carriers aged faster than non-carriers [new Figure 4].

Furthermore, our analysis distinguished aging-related gene expression from obesity and parity influences (referencing Burkholder et al., BCR 2020; Santucci-Pereira et al., BCR 2019) [new Figure 3—figure supplement 2]. Overlap with obesity and parity-related genes was minimal, corroborating the specificity of our aging LEP signature was unaffected by these factors [new Figure 4]. Thus, we affirm our findings are predominantly age-driven.

• Ability to segregate different subtypes of breast cancer with mammary lineage-derived signatures is well-established in the literature. The analysis would carry more weight if evaluated for prognosis and in multivariate analysis to determine if there is a value added to the assessment of these specific genes. Related to this, it would be helpful to describe in greater detail what multi-testing correction approach was used (and mention it and/or apply it also to figure 4 and supplements as appropriate). Were all genes tested in this analysis significantly different in their expression?

In our study, we demonstrated that mammary lineage-derived signatures that could predict breast cancer subtypes involved aging-associated genes that were specifically changing in the luminal lineage. These findings suggest that what is typically attributed to oncogenesis might actually be age related changes within the luminal lineage. Our machine learning model, using a non-linear ensemble method, captures the nuanced expression patterns of genes that vary with age, not just those that are differentially expressed. This could explain why certain older individuals are more susceptible to breast cancer initiation and subtype development.

While we expected aging-related DE and DV genes to be more associated with cancer initiation, we subsequently conducted survival analysis as Reviewer #1 suggested, assessing the contribution of top ML predictors (scaled variable importance ≥ 25%) in each subtype to overall survival (OS) and progression-free interval (PFI). We employed multivariate Cox proportional hazards regression models to test the simultaneous effect of all top predictors on survival time, adjusting for age at diagnosis and cancer stage (early vs. late). In PAM50 LumA cancers, we found 13% and 17% of top predictors (k_LumA_top _pred_=47) were associated with OS and PFI respectively; in PAM50 LumB cancers, 5% and 16% of top predictors (k_LumB_top _pred_=37) were associated with OS and PFI; and in in PAM50 Basal cancers, 0% and 3% of top predictors (k_LumB_top _pred_=14) were associated with OS and PFI (Wald *p*<0.05) [new Figure 8—figure supplement 2]. These suggest that subtype-specific expression of genes dysregulated with age in the LEP lineage also impact biology underlying survival outcomes specifically in patients with luminal type cancers.

Multiple testing adjustment in TCGA analysis was performed using Benjamini-Hochberg for selected genes that were discussed. To improve on the analysis, we have formalized the Kruskal-Wallis and post-hoc pair-wise Wilcoxon statistical tests comparing breast cancer subtypes and matched normal samples across all age-dependent DE and DV genes identified in LEPs (see Methods). We found 98% of age-dependent DE and DV genes (580 of 589 genes, respectively, in the TCGA cohort) to be differential expressed between PAM50 intrinsic subtypes (KW BH adj. p<0.05, post hoc Wilcoxon BH adj. p<0.05) [new Figure 7—source data 1]. When we further analyzed median expression of DE and DV genes across PAM50 LumA, LumB, Her2 and Basal subtypes, we found differential distribution of median expression across subtypes: (i) the largest fraction of DE genes upregulated in young LEP had the highest expression in Basal and lowest expression in LumB subtypes; (ii) the largest fraction of DE genes upregulated in old LEP had highest expression in LumA and lowest expression in Basal subtypes; and (iii) the largest fraction of DV genes with higher variance old LEP had highest expression in Basal and lowest expression in LumB subtypes (Fisher’s exact test Bonferroni adj. p<0.05); in contrast (iv) DV genes with higher variance in young LEP showed no association between subtype and median expression levels [new Figure 7—figure supplement 1].

• A previously published association between GRHL2 and SGSM2, and E-cadherin downregulation is noted, however, no mention of E-cadherin expression in this dataset is discussed to determine whether a similar reprogramming is present.

In addressing the expert reviewer's observations regarding our findings on E-cadherin (CDH1), we acknowledge the lack of differential expression with age in LEPs (adj. p=0.86). Despite the positive correlation between the expression of GRHL2 and SGSM2 in LEPs (R=0.47, p=0.014), our analysis indicates no significant linear association with CDH1 expression [see Author response image 1]. Given E-cadherin’s pivotal role in maintaining cell-cell adhesion and cell polarity within epithelial structures, its stable expression in the context of aging in normal cells aligns with expectations and is consistent with established biological principles, particularly as perturbations in E-cadherin expression are often correlated with oncogenic transformation.

The point raised by the reviewer is indeed insightful and has been incorporated into our Discussion section. This addition underscores the concept of 'priming' in cellular pathways during aging—a process we have previously documented (Sayaman, Miyano et al., bioRxiv 2021). In this process, certain pathway components undergo expression changes that set the stage for potential dysregulation as aging progresses.

Moreover, it is important to note that while the initiation of cancer requires a precipitating oncogenic event, our findings suggest that the cellular groundwork for accelerated malignancy progression may be laid by such age-associated changes. In a parallel observation, we have noted dysregulation of MYC targets in the aging milieu, although MYC itself does not exhibit such dysregulation. This nuance has been elaborated upon as a salient point in our discussion, emphasizing the complexity of gene regulation and pathway interaction in the aging landscape.

**Author response image 1. sa2fig1:** *CDH1* shows no age-dependent change in expression in luminal epithelial cells. (**A**) Expression of *CDH1* in young vs. old LEP with DE adj. *p-values*. (**B-D**) Linear regression of expression values in LEPs between (**B**) *GRHL2* and *SGSMS2*, (**C**) *GRHL1* and *CDH1*, (**D**) *SGSM2* and *CDH1*.

• SATB1 is noted to be decreased in LEPs of older women relative to younger women, correlating to decreased expression in breast cancer relative to normal tissue. Further analysis should be performed to determine whether SATB1 target genes are also downregulated to justify the claim that 'SATB1-mediated genome organization may play a regulatory role in the maintenance of the luminal lineage…'. Also, is SATB1 expression in the various breast cancer subtypes different when stratified for age as well?

Our investigation into the role of SATB1 in modulating aging in the luminal lineage of breast cells involved an in-depth analysis of 778 unique gene targets, previously identified as being influenced by SATB1 in the MDA-MB-231 breast cancer cell line (Han et al., Nature 2008). Out of these, we identified 515 SATB1-target genes within our dataset, with a significant proportion (64%) demonstrating correlated expression (|R|≥0.5) across LEP and MEP samples [Refer to Figure 5—figure supplement 1]. Moreover, we identified 26 SATB1-target genes that exhibited age-dependent differential expression (DE) specifically in the LEP lineage [Refer to Figure 5—figure supplement 1]. This finding lends support to the proposition that SATB1 contributes to the regulation of the aging phenotype manifest in LEPs.

Furthermore, our application of single-sample Gene Set Enrichment Analysis (ssGSEA) on SATB1activated and SATB1-repressed gene sets, which were differentially expressed in LEPs, revealed a significant enrichment of age-dependent DE SATB1-activated genes in younger LEPs when compared to their older counterparts (adj. p=0.002) [Refer to Figure 5—figure supplement 1].

We assessed *SATB1* expression across different breast cancer subtypes stratified by age in the TCGA cohort, focusing on two age groups with sufficient sample sizes: Middle-Aged (>30y and <55y, n=481) and Old (>55y, n=712). In concordance with our previous observations [as indicated in Figure 5], the PAM50 LumA, LumB, and Her2 subtypes exhibited significantly lower SATB1 expression compared to PAM50 Basal and Normal intrinsic subtypes (post hoc Wilcoxon adj. p<0.05) [Author response image 2].

Together, these results suggest that SATB1-mediated genome organization may play a regulatory role in the maintenance of the luminal lineage and the observed genome-wide dysregulation with age and in breast cancers in subtype-specific manner. However, we caution that mechanisms of SATB1-mediated genome organization in breast cancers are likely more complex than in aging normal epithelia due to the highly dysregulated state of cancers; this biology is also likely to be highly subtype-specific given the differential expression of *SATB1* across breast cancer subtypes.

• The rescue of ELF5 expression in LEPs when co-cultured with MEP-shGJB6 could be expanded to include additional LEP-specific TFs/markers to determine whether shGJB6 regulates biological aging as the whole of specific regulation of ELF5.

**Author response image 2. sa2fig2:** Differential expression of *SATB1* in breast cancer between PAM50 subtypes is consistent across age groups. Gene expression levels of SATBI in the TCGA cohort stratified by age group: Young <30y (n=12), MiddleAged >30y <55y (n=481) , and Old >55y (n=712). Kruskal-Wallis (KW) test performed across PAM50 breast cancer subtypes; p-values adjusted across age groups. Post hoc pairwise Wilcoxon test adj. p-value significance levels annotated; p-values adjusted across age groups.

ELF5, a pivotal transcription factor in alveologenesis, orchestrates the differentiation of progenitor cells into the luminal secretory lineage, as outlined in Lee et al. (Development 2013). Consistent with our prior research (Miyano, Sayaman et al., Aging 2017; Cancer Prev Res 2021), ELF5 emerges as a marker of senescence in luminal cells. Specifically, in our work (Miyano, Sayaman et al., Aging 2017), we demonstrated the necessity of direct LEP/MEP cell-cell contact for sustaining the LEP lineage in vitro, evidenced by the assessment of several LEP markers. Notably, ELF5 and PROM1 exhibited age dependent changes within a week of LEP/MEP co-culture, with ELF5 showing the most pronounced response [Author response image 3]. Conversely, the LEP marker KRT19 required a more prolonged coculture period to exhibit changes [Author response image 3]. Additionally, genome-wide expression shifts in co-cultured LEPs revealed that the age of the MEP used for co-culture significantly influenced the clustering of the LEPs [Author response image 3].

**Author response image 3. sa2fig3:** Expression of *ELF5* and target genes show age specific changes. Adapted from [79]. (**A**) Schematic shows co-culture conditions with young LEP atop of young or old MEP, respectively. Bar graph shows differences in LEP-specific gene expression of LEP markers, and of IGFBP6, a MEP-specific gene, in co-cultured LEPs after 10 days culture on young or old MEP feeders. * p<O.05 and ** p<O.01 annotated. (**B**) (**i**) Schematic of experimental outline for extended co-cultures. LEP were separated by FACS after 10 days co-culture either with young MEP (YY) or old MEP (Y/O). LEP from Y/Y and Y/O were further co-cultured with young MEP (Y/Y/Y and Y/O/Y) or older MEP (Y/O/O) for 7 days. (**ii**) Bar graphs showing normalized expression levels of KRT19, ELF5 and IGFBP6 in cocultured LEP following extended co-culture experiments. (**C**) Unsupervised hierarchical clustering of <30y LEP in Y/Y (n=3) and Y/O (n=3) co-cultures in parallel with <30y (n=5) and >55y (n=4) 4p LEP and MEP isogenic to the MEP strains used in co-culture (k=20,577 mapped genes). Percent Approximately Unbiased (AU) p-values denoted in red, and percent Bootstrap Probability (BP) in green; AU p≥95% are highlighted in red. (**D**) Gene-gene correlation matrix of ELF5 and 92 ELF5-target genes found to have absolute correlation 2 0.5 with ELF5 in LEP from <30y and >55y age groups across 9 HMEC strains. (**E-F**) Hierarchical clustering based on expression levels of ELF5 and the anti-/correlated ELF5-target genes (**E**) in <30y and >55y 4p pre-stasis LEP (n=9) and (**F**) in Y/Y (n=3) and Y/O (n=3) co-cultures with <30y (n=5) and >55y (n=4) 4p LEP isogenic to the MEP strains used in co-culture. Percent AU p-values denoted in red, and percent BP in green.

We further elucidated that the age-associated reduction of ELF5 impacts the expression of 92 target genes in the T47D breast cell line (Kalyuga et al., PLoS Biol 2012), following an age-dependent trajectory [Author response image 3]. These ELF5 target genes also exhibited similar age-related shifts during co-culture [Author response image 3]. Acknowledging the established impact of co-culture duration on the expression of LEP markers and considering the downstream age-related effects of ELF5 observed in co-culture, our experiments concentrated on MEP-induced variations in ELF5 expression in LEPs through the knockdown (KD) of aging-related genes in MEPs, serving as an experimental proof of concept. The intricate role of GJB6 in the context of biological aging and its relationship with ELF5 will be addressed comprehensively in a subsequent publication.

• The authors should more clearly describe what key findings in this work go beyond the findings they reported in the 2017 Aging paper (cited above) which also showed age-related changes in the mammary transcriptome and may have used many of the same samples.

For clarity, we have refined our manuscript. Our foundational work established the gradual loss of lineage fidelity as a consequence of aging, characterized by a decline in the specific expression of certain lineage markers, without affecting the expression of other key lineage-specific markers or the overall phenotypic and histological distinctions between luminal epithelial progenitors (LEPs) and myoepithelial progenitors (MEPs) as reported in (Miyano, Sayaman et al., Aging 2017). This study analyzed the expression of five LEP-specific and five MEP-specific biomarkers through qPCR, with microarray data from LEP and MEP samples (n=4 young <30y and n=5 old >55y) serving as a validation tool. The limited sample size meant that the microarray analysis was predominantly effective for differential expression (DE) analysis between LEP and MEP samples, where effect sizes were significant enough for reliable detection. Age-related effects were evaluated within gene sets through the use of hierarchical clustering algorithms, although not explicitly quantified at the individual gene level through DE analysis between the younger and older cells. Additionally, our Aging 2017 publication recognized non-cell autonomous aging mechanisms via our LEP/MEP co-culture system.

Building upon the findings from our 2017 study, we proposed that the aging-related alterations observed could affect the genome-wide preservation of lineage fidelity, potentially influencing susceptibility to cancer initiation in breast tissue. Employing a more robust RNA-seq dataset, which includes the cell strains from the earlier microarray analysis, we demonstrated that the reduction in lineage fidelity at the genomic expression level is a global aging phenomenon. This comprehensive dataset enabled us to establish a statistical definition for loss of fidelity and conduct a detailed statistical analysis of age dependent differential expression (DE) and differential variability (DV), revealing two distinct patterns of age-related changes—directional and variant responses—that are implicated in the observed loss of lineage fidelity with age. Lastly, we provided evidence that these age-related DE and DV alterations in LEPs are biologically significant and relate to breast cancer subtype differentiation.

• Some axes are a bit hard to understand. For example, is the expression of ELF5 really 50x higher than RPS18, a typically very highly expressed gene (Figure 5E)?

Thank you, this was an oversight during plotting in Prism, normalized values were incorrectly inputted with extra 0s. We have corrected the figure.

• For the differential variance of KDM2B, how much of this variance can be attributed to the one low point? I.e. is this value an outlier or is the variance still elevated if it is removed? It would be good to see other genes in which there is differential variance rather than just one example.

In our differential variability (DV) analysis, we incorporated an initial step of outlier detection and removal using the MDSeq methodology, which systematically excluded genes with extreme outliers that could disproportionately influence the effects of specific covariates. This process resulted in the retention of 14,601 out of the initial 17,328 genes, with a significance cut-off level set at 0.05. *KDM2B* was not identified as an outlier during this stage. Our methodological approach does not support the selective exclusion of individual samples and subsequent recalculation of variance, as this could introduce bias based on investigator discretion. In response to your query, it was observed that the variance of *KDM2B* in old LEPs remained higher than in young LEPs, even after the removal of the sample with the lowest variance value, although the difference was mitigated.

Following the suggestion from the reviewer, we have now incorporated additional illustrative examples of genes that display an increase in variance with age, presented as boxplots in the newly added Figure 3—figure supplement 3. These particular genes form a subset of the DV transcription factors previously depicted in Figure 3—figure supplement 3, and their relevance to breast cancer pathogenesis has been further expounded upon within the revised manuscript.

• Figre 3 supplement 1 viii is never mentioned in the text, and since it is looking at DV rather than DE it seems out of place here

Thank you, we have added the missing figure reference to DV gene *KDMB2* in the differential variability section and the figure has been moved to [new Figure 3—figure supplement 3] focused on DV results and expanded to include other DV genes as suggested by Reviewer #1 above.

• There are some additional seemingly paradoxical observations that would benefit from further explanation. For example, HES4 is a target of Notch1 and is downregulated in older LEPs but is typically associated with breast cancer progression. Why would cells be getting less tumorigenic with regard to HES4 if aging is generally indicative of more tumor-prone phenotypes? A similar question could be posed for ZNF503.

We note that our findings in age-related downregulation in *HES4* and upregulation of *ZNF503* were limited to the luminal epithelial lineage [Figure 3—figure supplement 1] and changes reflected in bulk tissue during tumorigenesis might be influenced by other cell types, as well as the breast cancer subtype-specific expression of these genes.

**Author response image 4. sa2fig4:** *HES4* and *ZNF503* are differentially expressed in breast cancer between PAM50 subtypes. Gene expression levels of (**A**) HES4 and (**B**) ZNF503 in the TCGA cohort. Kruskal-Wallis (KW) test performed across PAM50 breast cancer subtypes as well as matched normal tissue; pvalues adjusted across all age-dependent DE and DV genes identified in LEPs found in TCGA dataset. Post hoc pairwise Wilcoxon test adj. p-value significance levels annotated; p-values adjusted across all pairwise comparisons and acriss all DE and DV genes.

Examination of TCGA data set for instance showed *HES4* to be expressed lower in normal tissue and PAM50 LumA and LumB subtypes and higher in Her2 and Basal subtypes; *ZNF503* was expressed lower in PAM50 LumA and LumB subtypes and higher in normal tissue and Her2 and Basal subtypes [Author response image 4]. Of note, Notch1 signaling and *ZNF503* inhibition of *GATA3* are prognostic specifically in more aggressive basal/triple negative breast cancers (Mittal et al., Mol Cancer Ther 2014, Yuan et al., PLoS one 2015, Liu et al., Biomarkers 2023). However, we caution on direct comparison and interpretation of biology between aging, cancer initiation, and cancer progression due to complex mechanisms underlying these different phenomena.

• Figures such as strata plots and interactome maps are effective at including a lot of information in a condensed format, however, there can sometimes be too much going on in a single panel that can make it difficult to follow. For example, figures 1C-E effectively convey the main point even without figure 1B. Furthermore, try to choose color-blind safe color schemes (i.e. avoid red and green in the same figure).

We appreciate the comment regarding color-blind color schemes. The red and green have historical links to our current publications and manuscripts in review where LEP are depicted in green and MEP in red that are all tied together. For consistency with current work, it is relevant for us to keep the current color scheme.

• The terms "variance" and "directional changes" need clearer descriptions at the outset.

We have edited the Introduction, adding the following descriptions of the terms age-dependent directional and variant responses: “Directional responses reflect stereotyped changes associated with upregulation or downregulation of gene expression between younger and older cohorts; variant responses reflect increases gene expression variance within a cohort associated with the heterogeneity of individuals within a group.”

• To address concerns about cellular heterogeneity, the authors could:1) Show the sorting that was done to isolate these populations (and that was done to separate them post-culturing). Are the populations clearly distinct such that slight changes in cell dissociation, labeling, or isolation by FACS are unlikely to be responsible for the population-level expression changes? Looking back at one cited prior work (Miyano and Sayaman et al. Aging 2017 Oct 9;9(10):2026-2051), the markers used appear to create a continuum of expression in some settings with potentially subjective sort gates applied. Related to this, Elf5 is a known alveolar specifier. Are such cells equally partitioned into the LEP vs MEP sorted pools based on the markers used?

We provided the individual FACS gating plots for all samples including isolated 4^th^ passage cultured cells [new Figure 1 and Figure 1—figure supplement 1] and organoids [new Figure 1—figure supplement 1], as well as lineage-specific expression of canonical markers of LEPs and MEPs in the isolated populations [new Figure 1 and Figure 1—figure supplement 1]. We also provided FACS plots of KRT19+ and KRT14+ staining in the isolated LEP and MEP populations [new Figure 1—figure supplement 1]. To demonstrate the consistency of our FACS sorting strategy (regardless of FACS markers used) and HMEC culture system, we showed the unsupervised hierarchical clustering of all LEP and MEP samples used in the study; of note replicate bridge samples (indicated by subject ID) robustly clustered together regardless of FACS markers used or RNA-seq batch [new Figure 1—figure supplement 1 and updated Figure 1—source data 1].

We also provided scRNA-seq analysis showing expression of the FACs CD markers used in this study [new Figure 4, middle and right panels] that showed the CD markers are highly LEP or MEP specific.

*ELF5* was one of the most differentially expressed genes between our FACs sorted LEP and MEP populations in both organoids and 4^th^ passage cultures (lineage-specific DE adj. *p*<0.0001), with median regularized log expression in <30y and >55y LEPs and in <30y and >55y MEPs shown in new Figure 6—figure supplement 1. Note the figures displayed regularized log values which transforms the count data to the log2 scale in a way which minimizes differences between samples for rows with small counts, and which normalizes with respect to library size.

Multiple lines of evidence have shown the robustness of our FACS gating strategy in isolating LEP and MEP populations from HMEC and organoid systems.

2) Use an orthogonal method to confirm that sorting was effective and show the data.

The FACS methodology we employ has undergone rigorous validation within our facility over the past 13 years, as well as through several external studies from the LaBarge laboratory. This validation process has spanned various techniques including the use of independent lineage-specific CD markers for FACS, immunofluorescence assays with Keratin 19 and Keratin 14, and assessments based on cell culture morphology and functional assays. We have consistently observed that isolated LEP and MEP populations express the expected lineage-specific markers [refer to new Figure 1 and Figure 1—figure supplement 1], corroborated by findings from published studies employing CyTOF (PellissierVatter et al., Cell Rep 2018), mass spectrometry (Hinz et al., iScience 2021), DNA methylation array (Sayaman, Miyano et al., bioRxiv 2021), and whole-genome bisulfite sequencing (Senapati et al., Genome Res 2023).

The revised Figure 1 and its supplementary materials demonstrate that the two distinct FACS strategies we utilized effectively distinguished LEP from MEP populations, confirmed by multiple recognized LEP and MEP markers. We have taken steps to verify the reproducibility of isolating LEPs and MEPs using different LEP markers (CD227, MUC1 and CD133, PROM1) and MEP markers (CD10, MME and CD271, NGFR), ensuring comparability. This reproducibility is evident in the unsupervised hierarchical clustering of samples, which consistently groups samples from the same subject together, irrespective of the FACS strategy or RNA-seq batch used [see new Figure 1—figure supplement 1 and the updated Figure 1—source data 1]. Intracellular FACS analysis confirmed that isolated LEP and MEP populations express markers KRT19+ and KRT14+, respectively, at both the mRNA and protein levels, as detected by intracellular FACS staining. Furthermore, our differential expression (DE) linear models now include an RNA-seq batch factor that accounts for the FACS sorting strategy.

Additionally, we present single-cell RNA sequencing (scRNA-seq) analysis that validates the specificity of the FACS CD markers employed in our study [refer to new Figure 4]. This analysis revealed that canonical LEP marker CD227 (MUC1) was prominently expressed in both Luminal1 secretory (ER-) and Luminal2 hormone-sensing (ER+) cell types, with Luminal1 cells expressing LEP marker CD133 (PROM1) and Luminal2 cells expressing the androgen receptor (AR). MEP markers CD271 (NGFR) and CD10 (MME) were found to be highly expressed in the Basal cell type. It is noteworthy that the CD133/CD271 marker combination was exclusively used to isolate LEP and MEP populations in organoid systems, thereby excluding the ER+ luminal population. Given that the ER+ population does not proliferate in fourth-passage cultures, our analysis was focused solely on the ER- luminal and MEP populations.

3) Carry out in situ validation of the observations to delineate what proportion of the luminal or myoepithelial compartments in vivo show the directional expression changes uncovered following culture. Increased intra-sample expression variation could also be orthogonally quantified this way.

As described in our methods, lineage-specific DE analysis was run in both our primary uncultured epithelia and primary cultured samples, and genes that showed culture effect in terms of lineage expression were excluded from analysis [see Supplementary Methods]. Briefly, we identified genes with at least nominal lineage-specific expression in either organoids or 4^th^ passage cultures (lineage specific DE adj. *p*<0.01) and excluded genes which showed culture-dependent effects and did not maintain consistent lineage-specific expression between organoids and 4^th^ passage cells. A final set of 17,328 genes with dynamic ranges and consistent lineage-specific expression between primary organoid and 4^th^ passage LEPs and MEPs in both age cohorts (linear regression *R^2^*>0.88 to 0.91, *p<*0.0001) [Figure 1—figure supplement 2] were used in all downstream analyses.

We were only able to run RNA-seq on a smaller set of reduction mammoplasty organoid samples (<30y, n_LEP_=4, n_MEP_=3 , and >55y, n_LEP_=3, n_MEP_=1) as we were not able to isolate enough RNA from limited organoid material from all isogenic samples used in our study. While we were not powered to run agingdependent DE or DV analysis on our FACS sorted primary organoids, we showed that our organoid samples robustly clustered (AU p≥0.95) by lineage and age based on the expression of 495 agedependent DE genes (DE adj. *p*<0.05) identified in both our 4^th^ passage LEPs and MEPs [Figure 3—figure supplement 1]. As identification of DV genes was highly dependent on specific samples and the heterogeneity observed in specific individuals, a comparable assessment could not be performed in the available organoid samples.

We further provided independent validation of our age-dependent DE gene sets in new Figure 4 and accompanying section “Age-dependent gene expression changes detected at the single cell level” based on available single-cell studies in non-tumorigenic breast tissues (Pal et al., Nee et al., and Murrow et al.,). Briefly, we used single-cell GSEA (scGSEA) to show concordant and significant enrichment of our age-dependent DE genes upregulated in old LEPs in the Luminal 1 secretory cells from the menopausal cohort [new Figure 4, left panel], and DE genes downregulated in old LEPs in Luminal 1 cells from the premenopausal cohort [new Figure 4, right panel]. We also provided further investigation on how these age-dependent gene sets were enriched in non-chemo/chemo, noncarriers/BRCA1^mut/+^, nulliparous/parous, and non-obese/obese cohorts [new Figure 4].

4) Compare findings to recent single-cell transcriptomic studies e.g. Gray et al. Dev Cell. 2022 Jun 6;57(11):1400-1420.e7.

Gray et al., findings from BRCA1/BRCA2/RAD51 mutation carriers (n=13, 25-65y) and noncarriers (n=3, 36-41y) were consistent with finding in our earlier publication that women carrying high risk mutations in BRCA1/BRCA2/PALB2 exhibit accelerated aging phenotypes including basalization of the luminal lineage (e.g., luminal epithelial cells that acquired myoepithelial markers) as compared to agematched average risk women who were non-carriers (Shalabi et al., Nature Aging 2021).

We further included new Figure 4 that places our LEP aging signatures in the context of three recent scRNA-seq datasets (Pal et al., Nee et al., and Murrow et al.,) as described above.

Reviewer #2 (Recommendations for the authors):The study is significant as how aging-associated genetic and epigenetic changes in various cell types of the mammary gland may contribute to breast cancer susceptibility is unclear. The study involved a reasonable number of patient-derived mammary epithelial cell samples for extensive bioinformatics and machine learning analyses. The data such as loss of lineage fidelity during aging supports previous findings from the senior author's group and others. The identified aging-associated genes such as SATB1 may serve as intervention targets.

We thank Reviewer #2 for their thoughtful comments and good opinion of our work. Our collection of primary HMEC is among the largest of its kind, but metadata (other than chronological age) are lacking for most of the specimens collected before 2016, which includes all the specimens used in this manuscript. We address the comments pointwise.

However, significant alteration of gene expression and loss of lineage fidelity in LEPs should be expected between LEPs of premenopausal and postmenopausal women since hormone (estrogen and progesterone) receptor-expressing cells are mainly, if not exclusive, present in the LEP population.

We have made clarifications in the manuscript regarding the presence of hormone receptor cells in the LEP population. The luminal lineage is composed of ER+ hormone-sensing and ER- secretory populations with ER- LEP. The ER- luminal cells represents 80-95% of luminal cells in breast and comprise the cells of origin for breast cancer. Moreover, ER- luminal cells can be maintained in finite lifespan cell cultures and thus studied experimentally. As ER+ luminal cells do not grow in our HMEC culture system, only the ER- luminal cell population is the subject of our loss of lineage fidelity study.

While we did see age-dependent enrichment of early and late estrogen response genes in cultured non-hormone sensing populations from the older cohort based on our GSEA analysis, the % overlap with the age-dependent DE genes (adj. *p*<0.05) were limited (~4%) and did not represent a large majority of our aging signatures [see Author response table 1].

**Author response table 1. sa2table1:** Number and percent of age-dependent DE genes in LEPs (k=471 genes) overlapping with estrogen and progesterone-dependent gene sets.

	N Genes MSigDB Gene Set	N Genes DE with Age in LEP	%Genes MSigDB Gene Set	%Genes DE with Age in LEP
HALLMARK ESTROGEN RESPONSE EARLY	216	10	4.6	2.1
HALLMARK ESTROGEN RESPONSE LATE	218	16	7.3	3.4
GOBP CELLULAR RESPONSE TO ESTROGEN STIMULUS	16	2	12.5	0.4
GOBP RESPONSE TO ESTROGEN	70	4	5.7	0.8
GOBP PROGESTERONE RECEPTOR SIGNALING PATHWAY	9	0	0	0
GOBP CELLULAR RESPONSE TO PROGESTERONE STIMULUS	6	0	0	0
GOBP RESPONSE TO PROGESTERONE	40	0	0	0

Thus, the authors should analyze and present the percentage of the differentially expressed (DE) genes between the young LEPs and old LEPs shown in Figure 3 which are menopause, estrogen, and progesterone-dependent genes.

To address the potential confounding factor of menopause in our aging study, we strategically defined our younger cohort as under 30 years of age, a demographic largely representing premenopausal women, and our older cohort as over 55 years of age, typically postmenopausal. Middle-aged women were not included to avoid the uncertainty of their menopausal status, thus focusing the study on aging and menopause collectively. Our age-dependent differential expression (DE) gene sets are independently validated in the newly added Figure 4 and its corresponding section, "Age-dependent gene expression changes detected at the single cell level." This validation was based on available single-cell studies of non-tumorigenic breast tissues. Using single-cell Gene Set Enrichment Analysis (scGSEA), we observed a concordant significant enrichment of age-dependent DE genes upregulated in older Luminal1 cells (ER- luminal population) from the menopausal cohort [refer to new Figure 4, left panel], and downregulated DE genes in Luminal1 cells from the premenopausal cohort [refer to new Figure 4, right panel].

Furthermore, our prior studies have shown that young women who are carriers of BRCA1, BRCA2, or PALB2 germline mutations exhibit accelerated cytoskeletal and ELF5 loss changes that correspond to a biological age significantly ahead of their actual chronological age (Shalabi et al., Nat Aging 2021; Miyano, Sayaman et al., Cancer Prev Res 2021). This observation aligns with our scGSEA results from the Nee et al. dataset, where Luminal1 cells from women with BRCA1 mutations did not present as 'youthful' as those from age-matched women at average risk [see new Figure 4, right panel]. These findings lead us to believe that the aging changes we have documented are primarily due to aging itself, rather than menopause.

In examining the potential intersection between the 471 age-dependent DE genes in LEPs and those involved in estrogen and progesterone pathways, we analyzed four estrogen-related and three progesterone-related Molecular Signatures Database (MSigDB) Hallmark and Gene Ontology Biological Process (GOBP) gene sets. The analysis revealed minimal overlap; only about 4% of the DE genes were related to estrogen responses, involving 19 genes, and there was no detectable overlap with progesterone-associated genes [refer to Author response table 1]. Consequently, our findings suggest that the aging signatures we have identified are not predominantly linked to hormonal changes.

This should provide confidence as to whether the cultured cells used in the study can model the gene expression changes in situ, particularly given that only 97 genes were found to be significant age-associated DE genes in the normal human breast tissue (Figure 4-table supplement 1) out of 471 DE genes identified from the cultured young and old LEPs shown in Figure 3B.

In the updated version of our manuscript, we have furnished independent in situ corroboration for our identified age-dependent differentially expressed (DE) genes from young and old LEPs. Initially, we confirmed that both LEP and MEP samples procured from primary organoids exhibited robust clustering, with high statistical support (Approximate Unbiased p-value AU ≥ 0.95), according to lineage and age. This was determined by the expression patterns of the age-dependent DE genes (adjusted p < 0.05) that were also recognized in our 4^th^ passage LEPs and MEPs [refer to Figure 3—figure supplement 1].

Additionally, we have presented further independent validation of our age-dependent DE gene sets in the newly introduced Figure 4, drawing from available single-cell study data on non-tumorigenic breast tissues (cited as Pal et al., Nee et al., and Murrow et al.). Using single-cell Gene Set Enrichment Analysis (scGSEA), we demonstrated significant and consistent enrichment of our age-dependent DE genes within the secretory Luminal1 population, particularly within premenopausal and menopausal cohorts [as shown in new Figure 4]. Moreover, we delved into the enrichment of these agedependent gene sets across various cohorts, delineating differences between non-chemo/chemo, noncarriers/BRCA1mut/+, nulliparous/parous, and non-obese/obese populations [illustrated in new Figure 4].

Another precaution may be warranted in interpreting the data as to whether the gene expression of the mammary cells from mammoplasty reduction is significantly affected by a different set of environmental factors since the donors were likely obese in comparison to the general population. Could the data such as the significant enrichment of the Early and Late Estrogen Response gene sets in the old LEPs shown in Figure 3H be more related to this unique donor population since estrogen-responsive genes are expected to be more highly expressed in premenopausal women than postmenopausal women?

In addressing your concerns regarding the potential confounding effects of obesity on our data, we initially sought to contextualize obesity within the general population, particularly among women undergoing reduction mammoplasty. According to the 2017-2018 NHANES dataset from NIDDKD, 27.5% of women in the US were categorized as overweight, and 41.9% as obese (which includes the severely obese category), leaving 30.6% classified as normal weight or underweight. Additionally, when examining the prevalence of obesity across various adult age groups in the United States, the distribution appeared consistent [Author response image 5].

**Author response image 5. sa2fig5:** Patient BMI does not affect expression of early and late estrogen response pathway signatures in the studied reduction mammoplasty dataset. (**A**) Overweight and obesity statistics from the 2017-2018 NHANES data via the NIDDKD. Percentage of US adults with overweight and obesity by sex (left); and prevalence of obesity among adults age 20 and over by age and sex (right). (**B-C**) BMI distribution in the GSEI 02088 reduction mammoplasty (RM) dataset across all samples (**B**) and by age group (**C**). (**D**) Early (left) and late (right) estrogen response GSEA signature scores in GSE102088 RM dataset by BMI group.

Further analysis of the GSE102088 dataset, which includes data from 114 reduction mammoplasty cases reported in Song et al. (Oncotarget 2017), showed that the BMI distribution of our gene expression dataset was reflective of the broader population, with no significant bias towards obesity among women who underwent reduction mammoplasty (Fisher’s exact test p=0.36) [Author response image 5]. Notably, literature and best practices in plastic surgery suggest a BMI below 30 to mitigate complications during reduction mammoplasty, which aligns with our observations. Moreover, no significant correlation was found between BMI and age group within the GSE102088 dataset (Fisher’s exact test p=0.56) [Author response image 5], leading us to conclude that our age-dependent findings reported in bulk breast tissue are not unduly influenced by the BMI distribution of the donor population.

In the absence of BMI data for our human mammary epithelial cell (HMEC) sample set, we referenced the obesity gene expression signature detailed by Burkholder et al. (BCR 2020) to evaluate any potential overlap with our age-related gene expression patterns. This comparison revealed just a single gene overlap each within our differential expression (DE) and differential variability (DV) signatures from aging LEPs, specifically FN1 and ZNF613 [new Figure 3—figure supplement 2].

Finally, upon evaluating the GSE102088 dataset for enrichment of early and late estrogen response gene sets across different BMI categories [Author response image 5], we did not observe any significant enrichment. Consequently, we maintain that the BMI distribution within this donor population does not play a substantial role in the estrogen-dependent enrichment we detected in the cultured cell data.

1. For the section related to Figure 2 and associated supplementary data, using the expression change of ligands and receptors to identify disruption of lineage-specific signaling pathways would be less robust than using gene set enrichment analysis (GSEA) for the changes of various signaling pathways between LEP and MEP as a function of aging because gene expression of ligands and receptors does not always translate to their protein expression and signaling pathways are often modulated by posttranslational modifications.

Gene Set Enrichment Analysis (GSEA) is predicated on the ranking of differentially expressed (DE) genes, which can sometimes miss the nuances of cell-cell communication, especially where the disruption is due to the loss of either lineage-specific expression of ligands or their cognate receptors. This limitation is exemplified in our approach depicted in [new Figure 2], which demonstrates scenarios where GSEA may not fully capture the complexity of ligand-receptor interactions.

In the instance of younger cells (<30 years), as shown in [Figure 2], we note active lineage-specific signaling in both directions, and GSEA successfully identifies the associated pathways for both ligands and receptors. Conversely, [Figure 2] presents a scenario in which the age-related loss of LEP specific receptor expression occurs. Here, LEP receptor DE is identified and its pathway is detected by GSEA. The illustration in [Figure 2] represents a contrasting case where the MEP-specific expression of the cognate ligand is lost with age. Although the MEP ligand DE and its pathway are recognized by GSEA, the LEP receptor does not show DE, hence its pathway remains undetected by GSEA.

Our methodology overcomes this by including the LEP receptor in our functional enrichment analysis in [Figure 2], thereby accounting for the MEP-directed LEP receptor signaling pathway. This approach acknowledges the shift in cell-cell signaling homeostasis to a state where only LEP → LEP signaling predominates. We view these disruptions in cell-cell signaling as critical areas for further research.

Equally noteworthy, as seen in [Figure 2], even when the lineage-specific expression of both LEP and MEP ligands is preserved, our functional enrichment methodology still captures the dysregulation of both MEP ligand and LEP ligand pathways due to loss of the LEP receptor. This is particularly relevant when such dysregulation is due to the absence of the cognate receptor, metaphorically similar to the ineffectiveness of communication that occurs when one is 'shouting into the void'.

2. The authors should state whether there are limited lineage-specific genes that show reduced expression variance in the old LEPs and MEPs in comparison to the young ones to demonstrate that the increased expression variance is aging-dependent in the mammary epithelial cells.

The genes with decrease in variance with age (DV adj. p<0.05, log2 fold change in dispersion<0) were reported in [Figure 3—source data 1]. We did not focus on genes which showed a decrease in variance with age as interpretation of this phenomenon was more complicated and required gene-level examination. For one, it could mean that the dynamic range in older cells remained within the expected functional range of the gene as seen in younger cells. However, a decrease in variance could also reflect gene expression being turned on or off in the older cohort.

3. While it is reasonable to suggest that genes with an increased variance of expression during aging may contribute to loss of lineage fidelity during aging, the rationale for including them for clustering and predicting breast cancer subtypes in Figure 6 needs to be further supported by evidence.

This question is at the heart of the manuscript, and we have included this response in the Results and Discussion sections. We proposed that aging mechanisms operate through at least two distinct pathways: (i) General dysregulatory mechanisms, which were reflected in directional age-dependent changes that share common features with known cancer mechanisms. These changes contribute to an increased overall susceptibility to cancer as individuals age. (ii) Individual-specific dysregulatory mechanisms, which were identified by variations in aged populations. These variations may help us understand why certain individuals have higher vulnerability to cancer (i.e., not all aged individuals develop cancer), and why those who do develop specific cancer subtypes.

Hierarchical clustering and machine learning (ML) algorithms can effectively capture the influence of DV genes. It is worth noting that even DE genes inherently exhibit variability. We suggest that by integrating the variability in aging responses, we can gain further insights that not only enable the identification of key branching points in clustering and the most relevant age-dependent biomarkers in classification but also help us understand the varying susceptibility of individuals to specific breast cancer subtypes. Indeed, we found 98% of age-dependent DE and DV genes in LEPs (580 of 589 genes in the TCGA cohort) to be differential expressed between PAM50 intrinsic subtypes (KW BH adj. p<0.05, post hoc Wilcoxon BH adj. p<0.05) [new Figure 7—source data 1]. When we further analyzed median expression of DE and DV genes across PAM50 LumA, LumB, Her2 and Basal subtypes, we found differential distribution of median expression across subtypes: (i) the largest fraction of DE genes upregulated in young LEP had the highest expression in Basal and lowest expression in LumB subtypes; (ii) the largest fraction of DE genes upregulated in old LEP had highest expression in LumA and lowest expression in Basal subtypes; and (iii) the largest fraction of DV genes with higher variance old LEP had highest expression in Basal and lowest expression in LumB subtypes (Fisher’s exact test Bonferroni adj. p<0.05); in contrast (iv) DV genes with higher variance in young LEP showed no association between subtype and median expression levels [new Figure 7—figure supplement 1A].

Of interest, in our analysis of the predictive power of DE and DV genes in identifying PAM50 subtypes [Figure 8], we observed that DV genes were strong predictors (scaled variable contribution ≥ 25%) of PAM50 LumA, LumB and Her2 subtypes and bore no strong predictive value for PAM50 Basal subtypes. Luminal subtypes are by far the most associated with aging and we hypothesized that increased variances may underly the transcriptomic architecture of luminal cancer subtypes. Thus, we speculate that inclusion of DV genes in LEP could lead to better predictors.

4. The authors should explain how siRNA-mediated knockdown can be used to reduce the level of variance of GJB6 and to constrain specific molecular noise. Did the two old MEP cells used in Figure 5G express the highest levels of GJB6 among all old MEP cells? Would knockdown of GJB6 in MEPs expressing lower GJB6 not increase ELF5 expression?

We have elaborated on the approach in the Methods section. MEP cell strains used in KD experiments were specifically selected to be the samples from older women with increased expression of *GJB6* relative to MEP from younger women – these samples had ~2-fold increase expression of *GJB6*.

Expression of *GJB6* in five old MEP was above the 75% quantile distribution of *GJB6* in young MEP, as such, our KD experiments were focused on these samples. Expression of *GJB6* in three old MEP was below the 25% quantile distribution of *GJB6* in young MEP and KD in these samples would bring *GJB6* expression outside the expression window seen in young MEP, and thus we did not do *GJB6* KD in these samples.

A key part of our variant responses hypothesis is that tuned windows of expression is essential for proper function, and deviations either up or down from this range in older individuals could lead to aging-dependent dysregulation. Thus, selective KD of upregulated genes or overexpression of downregulated genes in specific samples would be ideal; however, as tuning windows of expression are difficult to achieve experimentally by overexpression, our co-culture experiments have focused on KD of DV genes in specific old MEP samples where we see upregulated expression above the 75% quantile distribution in young MEP samples.

5. The data shown in Figure 6 do not depict what genes and expression data were used to predict breast cancer in Figures 6C and 6D. How the findings may be translated?

We have provided further details in the manuscript regarding the 589 genes identified with age dependent differential expression (DE) or differential variability (DV) in LEPs, which were also present in the TCGA cohort. Hierarchical clustering of these genes was performed and was presented in Figure 7 as a heatmap dendrogram showing their expression across the TCGA dataset. Due to the high number of genes, individual gene symbols were not displayed on the heatmap; however, a complete gene list along with their differential expression across TCGA PAM50 subtypes is now included in the new Figure 7—source data 1.

Among these genes, 536 that are pertinent to age-dependent DE and DV in LEPs were included across three machine learning (ML) datasets (TCGA, GSE81540, and GTEx). These genes were used to construct an ML model using 75% of TCGA samples, which was then validated with the remaining 25% of TCGA samples and further verified against the GSE81540 and GTEx datasets. The genes were ranked by their scaled variable importance contribution to the predictive model, with the rankings and their respective contributions to model accuracy detailed in Figure 8—source data 1.

The findings underscored the significance of age-related changes within LEPs, elucidating their impact on the tissue-level biology that could predict breast cancer subtypes. The identified changes might be indicative of age-related dysregulation that aligns with the susceptibility to or initiation of frank tumors. Variability in these age-related changes among different individuals could potentially explain why certain individuals are more prone to the onset of breast cancer and the emergence of distinct breast cancer subtypes.

6. The authors should consider analyzing, or at least discussing, the potential interaction among aging, menopause, and obesity of mammoplasty reduction donors in regulating gene expression and expression variance.

We have discussed the potential interaction between aging, menopause, and obesity in the context of reduction mammoplasty donors above. Further evaluation of obesity and parity-related gene expression patterns, as described by Burkholder et al. (BCR 2020) and Santucci-Pereira et al. (BCR 2019), was provided within the framework of our identified age-related gene expression patterns [refer to new Figure 3—figure supplement 2]. The genes displaying directional and variant responses in LEPs and MEPs were found to be distinct from those previously reported to be associated with obesity, parity, and time since full-term pregnancy (FTP) in breast tissue. Specifically, our age-dependent DE and DV genes in LEPs and MEPs showed minimal overlap with published gene sets related to obesity, parity, and time since FTP. This observation is bolstered by our scGSEA analysis of the Murrow et al. dataset, where enrichment of the aging LEP signature in luminal and basal compartments was found to be only marginally affected by parity or obesity [refer to new Figure 4], thus reinforcing our findings in bulk tissue.

Moreover, our prior publications have demonstrated that hallmark aging changes in mammary epithelia and breast tissue are expedited in young pre-menopausal women with germline mutations in BRCA1, BRCA2, or PALB2 (Shalabi et al., Nature Aging 2021; Miyano, Sayaman et al., Cancer Prevention Research 2021; Zirbes et al., JMGBN 2021). Across all three studies, we observed no correlation or discernible impact from parity, gravidity, or BMI on these aging changes.

[Editors’ note: what follows is the authors’ response to the second round of review.]

The manuscript has been improved but there are some remaining issues that need to be addressed, as outlined below:Reviewer #1:The study of Sayaman et al. used a large panel of early passage ex vivo isolates from breast tissues of young and old donors to interrogate global transcriptome underpinning of age-related reduction in lineage fidelity. This study follows on earlier findings from LaBarge lab that revealed the loss of lineage fidelity induced by aged myoepithelial cells (MEP), but the effects manifesting in luminal epithelia (LEP). In-depth bioinformatical analyses of these ex vivo cultures, followed by interrogation of public datasets and some experimental studies validate and expand earlier inferences made with a limited subset of lineage-specific markers. More importantly, these analyses enable authors to make several well-supported novel inferences of high conceptual, biological and clinical importance. This is a substantially revised manuscript, and the technical concerns raised by previous reviewers appear to be adequately addressed/resolved.Strengths:Use of a large panel of early passage LEP and MEP lineage isolates, rigorously validated to adequately reflect in situ lineage phenotypes. The large number of samples enabled authors to discover the contribution of between-individual variability, which would not be possible with a smaller number of samples.This study provides additional support for prior hypothesis that age-related increase in cancer incidence reflects loss of proper specification of LEP lineage.In-depth unbiased interrogation of transcriptional differences between young and old lineages enabled new network-level insights that substantially advance our understanding of this under-appreciated contributor to the causal link between increased cancer risk and aging.Definition of gene signatures of young and old LEP enabled the authors to infer the premature aging effects of chemotherapy exposure and BRCA mutations – supporting the author's suggestion that aged LEP lineage could be potentially used as a leading marker of physiological breast epithelia aging which could be useful for both risk stratification and deeper understanding of breast cancer etiology. On the latter, the analyses found no evidence of increased aging with parity and BMI, important breast cancer risk factors, suggesting distinct causality from that of aging, chemo and BRCA mutations.To this reviewer, the most conceptually intriguing inference is the contribution of aging-associated increase in variability between samples. Aging-related increases in variability between individuals have been noted in some prior studies, but this phenomenon appears to be underappreciated and understudied. The lack of attention likely reflects unclear sequela of the phenomenon, as the consequences of variability are not intuitively obvious. Unlike the directional changes in specific genes, the enhanced variability does not lend to a straightforward hypothesis addressable via reductionist experimental pipelines. This study sheds new light on the potential consequences of this between-individual variability. First, it makes a compelling argument that the increased individual-individual variability is a substantial contributor to age-related lineage fidelity in both LEP and MEP lineages. Second, by highlighting the observed variability ins specific cancer-related genes, the authors make a logically sound argument that this between-individual variability likely translates to between-individual differences in age-related cancer risk.Experimental documentation that normalization of expression of gap junction GJB6 gene, up-regulated in a subset of old MEPs prevents the induction of aged phenotypes in LEP suggests the possibility of a potential pharmacological rejuvenation of breast epithelia that could potentially reduce cancer risk. At this point, this is a far shot as the clear path to pharmacological intervention is lacking. Still, this is an important proof of principle that supports the utility of follow-up efforts in this direction.Importantly, while this work focuses exclusively on breast epithelia, given the universality of aging-related physiological alterations across all tissues, and the concomitant increase in cancer incidence, this study suggests the potential utility of investigating the relevance of the above findings towards other tissues.

Thank you very much, we appreciate your thoughtful commentary.

Limitations.The conclusions are primarily based on bioinformatical inferences from cultured cells. The inferences are compelling and additionally supported by analyses of non-dissociated organoids and bioinformatical analyses of publically available datasets. Still, there seems to be a missed opportunity of providing additional credence through histological analyses of gene expression in situ.

We agree that this is a current limitation of the manuscript. Due to the length and complexity of the manuscript, the paper was focused on bioinformatic characterization of lineage-specific and age-dependent gene expression biomarkers. We focused our efforts on validating results in bulk tissue gene expression datasets due to the utility of high-throughput approaches.

This is a long, complex, data and analysis-heavy manuscript. While there is no "fluff" and the writing is very thoughtful with clear description of the premises, experiments and inferences, there is some redundancy and some parts of this work are difficult to get through. Together with the massive lengths and complexity, this would probably deter some readers.

This has been among the most challenging writing projects this team has taken on. In response to this limitation, we have additionally edited the manuscript to reduce its reader-deterring characteristics.

The discussion links the reported age-related increase in between-individual variance with a more appreciated phenomenon of an increase in noise-driven cellular heterogeneity, which appears to be shared between aging and cancers. The link between noise-driven cell-cell variability the variability between samples does not appear to be intuitively clear. This is a very intriguing connection, which would warrant additional elaboration.

We have added some more details and considerations of the potential links between cell-cell variance and inter-individual variances in the discussion as discussed under “Recommendations for the authors”. As well, we cite previous work we have published on this topic: Todhunter and Sayaman et al., Curr Op in Cell Bio 2018.

Recommendations for the authors:Please define "stereotypical", this is an important descriptor that can be misconstrued.

We agree and have added a definition that appears in the second sentence of the introduction:

“Stereotypical changes appear directional to the observer and are apparent at different physiologic scales, e.g., phenotypically though wrinkling, graying hairs, and increasing frailty; cellularly through increasing organ dysfunction, and loss of bone density, muscle mass and fat pads; and molecularly through decreasing levels of androgens and estrogens, and the upregulation or downregulation of gene or protein levels.”

Differential expression inferred from the analyses of cultured samples has been validated in undissociated, non-cultured organoids. Together with the demonstrated utility of using the gene signatures, derived from the cultured isolates, in interrogating bulk transcriptomics dataset, the evidence is deemed sufficient. Still, it would have been useful to further support by in situ (by IF or IHC) validation, or at least point out to prior studies that demonstrate the differences in situ. Given the numbers of DE genes, I assume there must be ones with large differences, abundant expression and strong epitopes. If this is not feasible or if doing so would entail substantial new effort, it would be unreasonable to deter the publication of this work. Therefore, this recommendation is meant to be considered as a suggestion rather than a request.

Thank you for this suggestion. Due to the extensive cost and effort in validating the specificity of antibodies for histological and immunofluorescence analyses, we believe this effort is better suited to follow-up manuscripts that explore mechanisms of action and functional validation of our findings that are underway in our laboratory. However, we have published multiple papers showing some of the age-dependent gene expression changes that are reported also herein as part of some of the large gene sets that came out of these analyses. KRT14 and ELF5 changes in LEPs, and KRT19 changes in MEPs are cardinal examples that we reported and that others have confirmed. The histological observations from these earlier works, in part, motivated the computational genome-wide exploration in this paper.

Per the last point in the public review, the connection of between sample variability and between cell variability needs to be elaborated. It is not intuitively obvious how one would follow from another.

Increased variance appears to be a very important phenotype of aging and, as Reviewer 1 (R1) suggests, it has been ignored in many contexts including breast cancer susceptibility. We are very pleased that our message came across the page in a clear manner. As R1 states we are far away from any application that harnesses or targets increased variance genes towards a therapeutic end.

Regarding the link between the increase in heterogeneity observed in samples from older individuals and our findings here of the increase in variability between individuals as they age, we expanded the Discussion section to elaborate on the topic as suggested:

“Moreover, in our view, this age-dependent differential variability between individuals is linked to the described increase in cellular heterogeneity in older individuals in single-cell studies, where aged cells were shown to have increased transcriptional variability and loss of transcriptional coordination compared to younger cells of the same tissue (Enge et al., 2017; Kowalczyk et al., 2015; Levy et al., 2020; Martinez-Jimenez et al., 2017). We had previously proposed that this within-sample transcriptional variability which leads to the identification of cell states and cell sub-populations in scRNAseq data, in turn, leads to the observed shifts in transcriptomes at the cell population and tissue-level in RNAseq data between samples (Todhunter et al., 2018). As this gene expression shift in the cell population and tissue-level varies per individual and is dependent on the distribution of specific cell states and identified cell sub-populations in any specific individual, transcriptomic profiles of samples from some older individuals can deviate from the mean expression observed in the studied cohort. Thus, we suggest that the increase in cellular heterogeneity with age underlies the population-level increases in variances between the individuals. Indeed, our scRNA-seq analysis revealed how variations in cellular characteristics within specific cell lineages, as identified by defined cell states, were strongly linked to particular samples—underscoring the significance of differential variability between individuals.”

This concept is largely illustrated in Figure 2 of Todhunter and Sayaman et al., Curr Op in Cell Bio 2018, and we think we have shown formal evidence of this in the manuscript.

Related to the previous point, I found the description of the premise and inferences from the GJB6 down-regulation experiment very confusing. In my opinion, this part warrants a separate manuscript as it will be probably lost in the already massive amount of work presented in the manuscript. If this is unrealistic at this point and/or if the authors disagree with this suggestion, I recommend to either elaborate the rationale and conclusions more clearly or to simplify. The hypothesis presented in line 556 is not entirely clear to me, and 6C schemata is insufficient to clarify. The authors interpret the effect of GJB6 downregulation in preventing the imposition of aged phenotype on LEPs as supporting variance as the driver of stereotypic aging phenotypes (lines 582-585). Perhaps this was stated in the hypothesis illustrated by 6C, but I could not understand the connection, and the reasoning appears to be somewhat contrived. It is unclear whether the stable difference in GJB6 expression between different MEP isolates can be attributed to the effects of noise. There seems to be an implicit assumption of between-cell variability in increased GHB6 expression in aged MEP, and that higher between-sample variability somehow emerges from this cell level variance – but this is not stated explicitly, and no evidence of variability between MEP cells from the same isolate is provided. If the authors see a logically clear connection that is missed by this reviewer, there should be a way to elaborate it more clearly. Otherwise, the more on-the surface (and probably more trivial) inference is that (a) between sample variability is of biological significance, as upregulation of GHB6 ages LEPs, and (b) the result indicates a potential utility of pharmacological intervention to suppress LEP aging. The experimental part of this section could benefit from elaboration as well. Does " the subset of older MEPs with higher expression relative to younger MEPs" refers to isolates with highest expression level? If so, it can be stated more explicitly.

We agree that this needed a clearer rationale and we have endeavored to provide that. In the response to the first set of reviewers, we had explained the rationale more clearly and have now incorporated this response into the manuscript:

“A key part of our variant responses hypothesis is that tuned windows of expression is essential for proper function, and deviations either up or down from this range in older individuals could lead to aging-dependent dysregulation. Thus, selective KD of upregulated genes or overexpression of downregulated genes in specific MEP samples that show outlier expression would be ideal; however, as tuning windows of expression are difficult to achieve experimentally by overexpression, our co-culture experiments focused on KD of DV genes in specific old MEP samples where we see upregulated expression above the 75% quantile distribution in young MEP samples. Thus, we asked whether knockdown or inhibition of *GJB6* expression in the older MEPs with the highest expression, relative to younger MEPs, could restore proper signaling between LEPs and MEPs. To test this, we used our established heterochronous co-culture system and measured recovery of LEP expression of *ELF5* as a readout of biological age (Figure 6). MEP cell strains used in KD experiments were specifically selected to be the samples from older women with ~2-fold increase expression of GJB6 that were above the 75% quantile distribution of GJB6 expression in young MEP.”

Additionally, we considered adding more data to the paper that are part of our ongoing studies in support of the concept that controlling the expression levels of highly variant genes (like GJB6) in MEPs could exert an impact on LEPs, but both reviewers have remarked upon the already comprehensive nature of this paper. Thus, we provide Author response image 6 that shows a summary of a series of experiments whereby we expressed shRNAs in an old MEP strain (n=1) to silence subset of genes that showed either age dependent DE (increased in O v Y) or DV (increased in a subset of O v Y); we then performed a 10 day heterochronus culture with young LEPs as we show in the manuscript for GJB6. The trend seems to be that oMEPs modified with shRNAs against most of the DV genes positively impacted ELF5 expression levels in yLEPs, whereas silenced DE genes did not have as strong an effect. The summary of these data is provided below. Taken together with the GJB6 example it is tempting to speculate that these highly variant genes may be highly connected to pathways in MEPs that exert a biological aging effect on LEPs.

**Author response image 6. sa2fig6:** Knockdown of genes that show age-dependent upregulation or increased variability in old MEPs lead to increase expression of ELF5 in co-cultured young LEP. Genes identified as DE or DV via RNA-seq and proteins identified as DE via mass spec are knocked-down in old MEP, followed by 10 day heterochronus culture with young LEP. ELF5 expression level in co-cultured young LEP are then assessed via qPCR.

Regarding the schemata presented in Figure 6, we have clarified the text as follows:

“Because changes in MEPs were predominantly associated with DV rather than DE, we hypothesized that MEPs from different individuals could exert aging phenotypes on LEPs via different gene regulatory mechanisms that may implicate the DV genes observed in LEPs (Figure 6). As LEPs exhibited the vast majority of age-dependent DE changes, we further hypothesized that LEPs serve as integration nodes for dysregulation in MEPs where variant changes converge via common pathways that lead to directional changes in genes downstream of these pathways (Figure 6).”

Lines 274-287 confer a key inference of the insufficiency of directional changes to explain loss of lineage fidelity. The inference is well-supported by the data and reasoning, but it took substantial time and effort to digest the paragraph. It might be worth to re-write the paragraph, perhaps with adding a schemata to make it easier to digest.

Thank you for this comment. We have altered the passage with the intent of making the points less confusing. The new text reads:

“To investigate how age-related changes affect lineage fidelity in LEPs (Figure 3), we analyzed the overlap between age-dependent DE genes and genes that showed loss of lineage-specific expression. We found that only 9% of the lineage-specific DE loss was attributable to age dependent DE in either LEPs or MEPs (adj. *p*<0.05). Expanding our criteria to include genes with at least a 2-fold change in DE due to age, we found that these age-related changes accounted for only 21% of the observed loss in lineage-specific expression. This led to a notable reduction in the difference in expression levels between LEP- and MEP-specific genes in older cells (Figure 3). Further analysis showed that the age-related DE genes in LEPs and MEPs did not overlap with genes associated with obesity, parity, or time since full-term pregnancy in breast tissue (Burkholder et al., 2020; Santucci-Pereira et al., 2019) (Figure 3—figure supplement 2). Collectively, although these results indicate that age-dependent DE changes do contribute to the loss of lineage fidelity, they do not fully account for it.”

Reviewer #2:The authors precisely culture primary mammary epithelial cells from breast reduction surgeries from a set of younger (<30 years) and older (>55 years), then separate the luminal and myoepithelial cells by flow cytometry, and perform gene expression profiling and compare differences between groups. Genes that show differential expression or differential variability with aging are interpreted as indicative of a loss of cell identity. Comparisons with publically available datasets for single cell RNA sequencing of breast tissue find overlap with aging-dependent differences as well as response to chemotherapy and presence of BRCA mutations. The authors also find overlap with datasets of bulk RNA sequencing, which they interpret as indicating that the luminal/myoepithelial aging phenotype may dominate the overall aging differences in the whole tissue. They also find that their genesets can be used to create classifiers of breast cancer subtypes as defined by PAM50 classifications, and further identify specific genes that, when manipulated, suppress heterotypic induction of aging-associated gene signatures.One key strength of the study is the comprehensive and thorough evaluation of their gene sets across multiple types of datasets, which strongly supports their assertions that the age-associated differences in luminal and myoepithelial cells are the principal definers of overall aging in the breast. Another key strength is the careful and thorough evaluation of the concept that differential variability of gene expression is a key characteristic of aging. This is a very challenging concept, and the authors do an excellent job of presenting it.

Thank you very much, we appreciate the reception to the new concepts we outlined in the manuscript.

A subtle point not reinforced in the text is that since these experiments were performed on passaged epithelial cell populations, the aging-dependent gene expression differences have a degree of epigenetic stability, which further reinforces their assertions that the changes in the epithelial subtype lineage determinants are the principal drivers of the aging phenotype (and cancer susceptibility).

Thank you for your considered opinion of our work! You correctly (we think) suggest that there is a degree of epigenetic stability of the age-related phenotypes we described. Indeed, we are working on submitting a companion piece that describes a detailed analysis of genome-wide DNA methylation that recapitulates the lineage-specific and age dependent findings we present here (Sayaman and Miyano et al., bioRxiv 2021: https://www.biorxiv.org/content/10.1101/2021.02.12.430777v1).

The findings in that bioRxiv manuscript are further supported by our more recent work that showed LEP of older women also lost epigenetic suppression of retrotransposons with oncogenic potential (Senapati, et al., Genome Research 2023).

We address this point and provide further discussion of what we perceive to be the role of agedependent epigenetic alterations on oncogenic activation in the address the “Weaknesses” section below.

Weaknesses of the study include the lack of consideration of other cell types in the breast and their role in the aging process; while this is briefly mentioned in the discussion, it is certainly an essential component of aging and cancer. It may be that the loss of epithelial lineage fidelity is the underlying cause of aging, but these changes will occur in combination with reciprocal changes in stromal cell types, and these may contribute to epigenetic programming and differential variability. This possibility could be evaluated as an extension of the dataset comparisons in the study.

We agree, and we have been working for some years now to understand contribution from the stromal cell types (e.g., fibroblasts and immune cells) to aging phenotypes that we observe in epithelia. We view epithelia as the terminal recipients in breast of age dependent changes. In its entirety this is a massive problem and as such we are breaking our analysis and experiments in smaller parts.

We have expanded the original Discussion text as follows to address the stated limitations:

“Aging associated changes in the immune response were further implicated through our ligand receptor pair analysis where we identified known immune-associated ligands and receptors that exhibited loss of lineage-specific expression in breast epithelia. We had previously shown that *in situ* innate and adaptive immune cell infiltration of the breast epithelia and interstitial stroma change with age consistent with a decline in immune surveillance and increased immunosuppression (Zirbes et al., 2021). How age-dependent changes in other cell types in the breast including stromal, vascular, and immune cell populations are linked to the dysregulation of epithelial signaling with age remains to be fully elucidated. Comprehensive transcriptomic profiling of FACS isolated primary immune, stromal, and vascular cell types from normal breast tissue had identified lineage-specific expression of ligands and receptors that could contribute to dynamic and reciprocal signaling between cell types in the breast interactome (Del Toro et al., 2024). Thus, reciprocal changes in these other cell types may contribute to the differential variability observed in epithelial cell types via cell-cell and cell-ECM interactions.”

Recommendations for the authors:This study was carefully performed and beautifully presented. The results argue for a new paradigm of aging in the breast, and it is to be expected that these concepts will be challenging to parse. I have no major issues with any aspect of the study. One point that could benefit from clarification is the concept that gene expression changes precede oncogene alterations. This was briefly alluded to in the manuscript, but the interpretation of this was unclear to me: are the authors suggesting that epigenetic activation of gene expression networks may support oncogenic gene activations that will drive the same pathways? If so, this point could benefit from further discussion.

The paper itself is originally part of larger computationally-focused manuscript analyzing matched gene expression and DNA methylation (DNAm) data which is available on bioRxiv (Sayaman and Miyano et al., 2021: https://www.biorxiv.org/content/10.1101/2021.02.12.430777v1) where we provide more thorough treatment of this topic. Due the length and complexity of the subject, we decided to split the paper with this manuscript focused on gene expression; we aim to submit the companion DNAm manuscript after publication of this paper. Our DNAm manuscript tackles the concept of “epigenetic priming” that we believe precedes and contributes to oncogenic activation and we summarize key points below:

Epigenetic changes recapitulated the loss of lineage fidelity we observed in gene expression, and age-dependent differential methylation (DM) changes were overwhelmingly found in the LEP lineage.A majority of lineage-specific and age-dependent differentially expressed genes had differentially methylated (DM) regions, particularly in promoter-proximal regions, and showed negative correlation between expression and DNAm levels. This suggested that epigenetic alterations lead to the metastability of gene expression changes. DM genes likewise showed enrichment for Hallmark pathways that have been shown to play a role in cancer progression.Interestingly, there were also a large fraction of genes in LEPs with age-dependent DM of CpG sites that did not have concomitant gene expression changes with age.These same CpG sites were also DM in cancers, and the corresponding genes were highly expressed in a subtype-specific manner and relative to their matched normal tissue.Moreover, these age-dependent DM sites were also binding sites for transcription factors that are upregulated in specific breast cancer subtypes relative to normal tissue.

These findings led us to hypothesize that these demethylated CpG sites in older women are “primed” and could provide a mechanism for both aging-associated increase in cancer susceptibility and differential susceptibility between individuals to cancer initiation as they age and the development of specific cancer subtypes. We propose that gene activation may occur during cancer initiation in the subset of women with dysregulated expression of TFs that bind these age-dependent DM regions. And because some these TFs themselves are differentially variably expressed in older LEPs, it could explain why only certain individuals develop cancer and in subtype-specific manner. It is however important to note that genes with oncogenic activation had further DNAm changes outside the regions that are DM with age, which suggests that further epigenetic alterations might occur to stabilize the tumorigenic state.

We added the following brief statement to the discussion to address Reviewer 2’s comments:

“That age-dependent changes are evident in epithelial lineages isolated from primary HMEC cultures suggest a degree of epigenetic stability that allow aged epithelial cells to occupy metastable states. Indeed, our examination of DNA methylation in matched LEP and MEP samples showed age-dependent differential methylation (DM) at promoter proximal regions of DE genes and negative correlation between DNA methylation and gene expression levels in these samples (Miyano et al., 2017, Sayaman et al., 2021). DM genes in older LEPs likewise showed enrichment for signaling pathways that have been shown to play a role in cancer progression (Sayaman et al., 2021). Moreover, LEPs of older women also exhibited loss of epigenetic suppression of retrotransposons that affect regulation of genes with oncogenic potential, specifically genes associated with luminal breast cancers (Senapati et al., 2023). Thus, age-dependent activation of gene expression networks in older breast epithelia that are stabilized through concomitant age-dependent changes in the epigenetic landscape could prime aged epithelia for oncogenic gene activation.”